# Efficient Sampling on Riemannian Manifolds via Langevin MCMC

**Xiang Cheng**
Massachusetts Institute of Technology
x.cheng@berkeley.edu

**Jingzhao Zhang**
Tsinghua University jzhzhang@mit.edu

**Suvrit Sra**
Massachusetts Institute of Technology
suvrit@mit.edu

## Abstract

We study the task of efficiently sampling from a Gibbs distribution $d\pi^* = e^{-h}d\text{vol}_g$ over a Riemannian manifold $M$ via (geometric) Langevin MCMC; this algorithm involves computing exponential maps in random Gaussian directions and is efficiently implementable in practice. The key to our analysis of Langevin MCMC is a bound on the discretization error of the geometric Euler-Murayama scheme, assuming $\nabla h$ is Lipschitz and $M$ has bounded sectional curvature. Our error bound matches the error of Euclidean Euler-Murayama in terms of its stepsize dependence. Combined with a contraction guarantee for the geometric Langevin Diffusion under Kendall-Cranston coupling, we prove that the Langevin MCMC iterates lie within $\varepsilon$-Wasserstein distance of $\pi^*$ after $\tilde{O}(\varepsilon^{-2})$ steps, which matches the iteration complexity for Euclidean Langevin MCMC. Our results apply in general settings where $h$ can be nonconvex and $M$ can have negative Ricci curvature. Under additional assumptions that the Riemannian curvature tensor has bounded derivatives, and that $\pi^*$ satisfies a $CD(\cdot, \infty)$ condition, we analyze the stochastic gradient version of Langevin MCMC, and bound its iteration complexity by $\tilde{O}(\varepsilon^{-2})$ as well.

## 1 Introduction

Stochastic differential equations (SDEs) offer a powerful formalism for studying diffusion processes, Brownian motion, and algorithms for sampling and optimization. We study in particular the following *geometric stochastic differential equation*:

$$dx(t) = \beta(x(t))dt + dB_t^g, \tag{1.1}$$

that evolves on a $d$-dimensional Riemannian manifold $(M, g)$. Analogous to the Euclidean setting, the map $\beta : x \to T_xM$ denotes a drift ($T_xM$ is the tangent space at $x \in M$), and $dB_t^g$ denotes the standard Brownian motion on $M$ [Hsu, 2002, Ch. 3]. The notation in (1.1) is a shorthand; we can define (1.1) more precisely as the unique diffusion process whose generator is the operator $Lf = \langle \nabla f, \beta \rangle + \frac{1}{2}\Delta f$, where $\Delta$ is the Laplace-Beltrami operator (see [Hsu, 2002, Proposition. 3.2.1]).

When the drift $\beta(x) = -\frac{1}{2}\nabla h(x)$ (the gradient is taken w.r.t. the manifold metric), the SDE (1.1) has an invariant distribution $\pi^*$ with density $e^{-h(x)}$, with respect to the Riemannian volume measure $\text{vol}_g(x)$; for $M = \mathbb{R}^d$ with metric $g(x) = I$, this reduces to the familiar result for Euclidean Langevin diffusion. Using the *Kendall-Cranston Coupling* technique (Chapter 6.5, Hsu [2002]) together with a carefully constructed Lyapunov function by Eberle [2016], we can quantify the mixing rate for (1.1) in the 1-Wasserstein distance (w.r.t. the manifold distance) under suitable regularity conditions.

36th Conference on Neural Information Processing Systems (NeurIPS 2022).

However, the exact SDE is not implementable in practice, so we consider an MCMC algorithm using the *Geometric Euler-Murayama* [Piggott and Solo, 2016, Muniz et al., 2021] discretization of (1.1):

$$x_{k+1} = \text{Exp}_{x_k}\left(\delta\beta(x_k) + \sqrt{\delta}\zeta_k\right), \tag{1.2}$$

where $\text{Exp}_x$ denotes the exponential map for $M$, and $\zeta_k$ is a standard Gaussian with respect to *any* orthonormal basis of $T_{x_k}M$. It has long been known that the discrete-time process (1.2) converges to the SDE (1.1) in the limit as $\delta \to 0$ [Gangolli, 1964, Jørgensen, 1975, Hsu, 2002].

**Motivation for studying the SDE** (1.1)

Geometric SDEs such as (1.1) play a crucial role in the design and analysis of MCMC algorithms [Girolami and Calderhead, 2011, Patterson and Teh, 2013] that have had much success in solving Bayesian problems on statistical manifolds. Recently, it has been shown that the *mirror Langevin Algorithm* on the Hessian manifold of a function can be significantly faster than its Euclidean counterpart [Zhang et al., 2020, Chewi et al., 2020, Li et al., 2022, Gatmiry and Vempala, 2022].

These SDEs also directly relate to the tasks of sampling and optimization on manifolds, where often a Lie group structure helps capture symmetries (e.g., the Grassmann manifold, SO$(n)$, $O(n)$, etc.) [Moitra and Risteski, 2020, Piggott and Solo, 2016, Muniz et al., 2021]. Furthermore, in Lee and Vempala [2017, 2018], the authors propose fast algorithms for constrained sampling and volume estimation on polytopes by sampling from the Hessian manifold of a barrier function.

## 2 Overview of Main Contributions

**Our first contribution** in this paper is to provide a quantitative, non-asymptotic bound on the discretization error between (1.1) and (1.2). We present this bound in Lemma 1 in Section 4. In our bound, the expected-squared-distance between a single step of (1.2) and (1.1) over $\delta$ time is bounded by $O(\delta^3)$. This $\delta^3$ error scaling matches that of Euler Murayama in Euclidean space [Durmus and Moulines, 2017]. We highlight that our bound is entirely explicit, and depends polynomially on dimension, sectional curvature of $M$, and the Lipschitz parameter for $\beta$ – quantities which are intrisic to the manifold, and invariant to the choice of coordinate system.

Two recent papers, [Wang et al., 2020, Li et al., 2022] also look into a similar problem of bounding the discretization error of (1.2). Wang et al. [2020] analyze the error of (1.2), but rely on uniformly bounding certain derivatives of the densities along the sample path (see Assumption 3 [Wang et al., 2020]); even in Euclidean space, it is not clear whether such a bound exists, and whether it depends only polynomially on parameters such as dimension. Li et al. [2022] provide a quantitative bound on the bias of (1.2) on Hessian manifolds; they assume a bound on a "modified self-concordance" parameter that is not affine invariant and can be made arbitrarily large.

We discuss Lemma 1 in more detail , and sketch its proof in Section 4. The proof relies on a careful construction of geometric Langevin Diffusion as the limit of (1.2), and may be of independent interest.

**Our second contribution** is to show that after $\tilde{O}(\varepsilon^{-2})$ steps, the distribution of (1.2) is within $\varepsilon$ 1-Wasserstein distance from the stationary distribution of (1.1). We present this result in Theorem 1. This $\varepsilon$ dependence matches that of Euclidean Langevin MCMC [Durmus and Moulines, 2017].

The $\tilde{O}$ in our iteration complexity hides polynomial dependency on dimension, sectional curvature, Lipschitz-parameter of $\beta$, and $1/\alpha$, where $\alpha$ can be viewed as "mixing rate of the exact SDE." Theorem 1 requires that $\beta$ satisfy a manifold analog of the distant-dissipativity assumption (Assumption 2 with $m > L_{Ric}/2$, where $-L_{Ric}$ is a lower bound on the Ricci curvature of $M$). Assumption 2 is general enough to include cases when $\beta = -\nabla h$ for some nonconvex $h$ or when $M$ has negative Ricci curvature. The catch is that in such cases, $1/\alpha$ can become very large; this is generally unavoidable, even on Euclidean space. Distant-dissipativity assumption has often been used in the analysis of Euclidean Langevin diffusion for non-log-concave distributions [Eberle, 2016, Bou-Rabee et al., 2020, Gorham et al., 2019, Cheng et al., 2020].

**We highlight a computational sub-contribution that** (1.2) only requires computing exponential maps in the direction of $\beta$ plus a uniform Gaussian direction. In many cases, exponential maps are efficiently computable, and this Theorem 1 leads to a computationally efficient sampling algorithm. Our analysis extends naturally if one replaces (1.2) by a retraction step; if the retraction is of order $3/2$

or higher, the one-step error still scales as $O(\delta^3)$. The question of how to construct good retraction maps is well studied in literature, and is somewhat orthogonal to our main objective, so we do not provide details here, and instead refer readers to [Absil and Malick, 2012, Absil et al., 2009].

Recently, Ahn and Chewi [2021], Gatmiry and Vempala [2022], Li and Erdogdu [2020] have considered a different discretization of (1.1), where one assumes that *the endpoint of the geometric Brownian motion $dB_t^g$ can be sampled exactly* (so that only the drift $\beta$ is discretized). Sampling the exact geometric Brownian motion can be done efficiently in special settings such as the sphere [Li and Erdogdu, 2020], but on a general manifold, this can be much more expensive than computing an exponential map; in such settings the Langevin MCMC algorithm based on (1.2) may be preferable. We do highlight, however, that a number of the above results provide error bounds in KL-divergence, which is tighter than the Wasserstein bound in Theorem 1.

We discuss assumptions and consequences of Theorem 1 in more detail and sketch its proof in Section 5. The proof essentially combines our discretization analysis with a mixing result for (1.1) based on the *Kendall-Cranston* coupling and a Lyapunov function from [Eberle, 2016].

**For our third contribution,** we analyze the manifold analog of SGLD [Welling and Teh, 2011]:

$$x_{k+1} = \mathrm{Exp}_{x_k}\big(\delta\tilde{\beta}_k(x_k) + \sqrt{\delta}\zeta_k\big), \tag{2.1}$$

The the difference between (2.1) and (1.2), is that at each step, $\beta$ is replaced by a random vector field $\tilde{\beta}_k$ which satisfies: (i) $\mathbb{E}[\tilde{\beta}_k] = \beta$; and (ii) $\|\tilde{\beta}_k(x) - \beta(x)\| \leq \sigma$ almost surely. We show that after $\tilde{O}(\varepsilon^{-2})$ steps, (2.1) is within $\varepsilon$-2-Wasserstein distance from the stationary distribution of (1.1); thus the $\varepsilon$ dependency does not degrade when replacing $\beta$ by its stochastic estimate. We present this result in Theorem 2. For this analysis, we require Assumption 2, with $m > L_{\mathrm{Ric}}/2$ and $\mathcal{R} = 0$; this is a more restrictive condition than Theorem 1; when $\beta = -\frac{1}{2}\nabla h$, this restriction is equivalent to the $CD(\cdot, \infty)$ condition [Bakry et al., 2014]: $\nabla^2 h + \mathrm{Ric} \succ 0$, where $\mathrm{Ric}$ is the Ricci curvature tensor[1].

Though restrictive, the $CD(\cdot, \infty)$ class of distributions is nonetheless interesting; it is the manifold analog of the class of log-concave densities on Euclidean space, and sampling from this class has been a topic of much recent interest. In machine learning, the stochastic estimate of $\beta$ can often be computed much more quickly than the exact $\beta$, e.g., when $\beta = -2\nabla f$, where $f$ is the empirical average of some loss over a large number of observations. In such cases, (2.1) can be much faster than (1.2). In Section 6 we discuss the assumptions and consequences of Theorem 2 in greater detail.

Though Theorem 2 requires more restrictive assumptions than Theorem 1, its proof differs from the proof of Theorem 1 in a significant way: we show mixing of the discrete-time process directly (instead of relying on mixing of the exact SDE). This approach is necessary in order to take advantage of the fact that $\tilde{\beta}_k$ is an unbiased estimate $\beta$. The key intermediate result for showing mixing under (2.1) is Lemma 29. We believe this lemma to be of independent interest as it quantifies the distance evolution between two arbitrary discrete-time stochastic processes (that may be unrelated to Gaussian noise and Brownian motion). As an example, Mangoubi and Smith [2018] showed that ball walk mixes quickly on manifolds with positive sectional curvature. Using Lemma 29, one can show the more general statement "ball walk mixes quickly on compact manifolds with positive Ricci curvature."

## 3 Preliminaries: Key Assumptions and Notation

We state in this section the four key assumptions of this paper. Our first assumption involves lower bounding Ricci curvature of the manifold $M$. Let $\mathrm{Ric}$ denote the Ricci curvature tensor:

**Assumption 1.** *We assume that for all $x \in M$, $u, u \in T_x M$, $\mathrm{Ric}(u, u) \geq -L_{\mathrm{Ric}}$, for some $L_{\mathrm{Ric}} \in \mathbb{R}$.*

Intuitively, the more positive the Ricci curvature (i.e. the smaller the value of $L_{\mathrm{Ric}}$), the faster geometric Brownian motion mixes.

Our second assumption is a natural generalization of the distant-dissipativity condition in the Euclidean setting. It helps ensure that the drift traps the variable within a bounded region. For any $x, y \in M$, let $\mathrm{d}(x, y)$ denote their Riemannian distance.

**Assumption 2.** *We call a vector field $\beta$ $(m, q, \mathcal{R})$-distant-dissipative if there exist constants $m > 0$, $\mathcal{R} \geq 0$, and $q \in \mathbb{R}$ such that, for all $x, y$ satisfying $\mathrm{d}(x, y) \geq \mathcal{R}$, there exists a minimizing geodesic*

---

[1]note that $\beta = -1/2\nabla h$, therefore $m$ is $1/2$ times strong-convexity parameter of $h$.

$\gamma : [0, 1] \to M$ with $\gamma(0) = x$ and $\gamma(1) = y$, such that the inequality

$$\langle \Gamma(\beta(y); y \to x) - \beta(x), \gamma'(0) \rangle \leq -m\mathrm{d}(x, y)^2,$$

holds, where $\Gamma(\cdot; y \to x)$ denotes parallel transport from $T_y M$ to $T_x M$ along $\gamma$. In addition, for all $x, y$ satisfying $\mathrm{d}(x, y) \leq \mathcal{R}$, there exists a minimizing geodesic $\gamma : [0, 1] \to M$ with $\gamma(0) = x$ and $\gamma(1) = y$, such that we have instead the inequality

$$\langle \Gamma(\beta(y); y \to x) - \beta(x), \gamma'(0) \rangle \leq q\mathrm{d}(x, y)^2.$$

For some intuition: a strictly convex function will have $m > 0$, $\mathcal{R} = 0$, and $q$ arbitrary. Note that we do not require a unique geodesic between $x, y$.

For some intuition about Assumption 2: in the Euclidean setting, the first condition simplifies to $\langle \beta(y) - \beta(x), y - x \rangle \leq -m\|y - x\|_2^2$, and the second condition simplifies to $\langle \beta(y) - \beta(x), y - x \rangle \leq q\|y - x\|_2^2$.

We need our third and fourth assumptions for bounding the discretization error of (1.2). Assumption 3 upper bounds the Lipschitz constant of $\beta$; Assumption 4 lower bounds sectional curvature of $M$.

**Assumption 3.** *A vector field $\beta$ is $L'_\beta$-Lipschitz if, for all $x \in M$ and all $v \in T_x M$, $\|\nabla_v \beta(x)\| \leq L'_\beta \|v\|$.*

In the (flat) Euclidean setting, Assumption 3 is equivalent to saying that $\beta(x)$ being a $L'_\beta$ Lipschitz vector field.

**Assumption 4.** *Let $R$ be the Riemannian curvature tensor of the manifold $M$. We assume that there exists $L_R \in \mathbb{R}^+$ such that for all $x \in M$, and for all $u, v, w, z \in T_x M$, $\langle R(u, v)v, u \rangle \leq L_R \|u\|^2 \|v\|^2$.*

We now introduce additional notation that will be used throughout this paper. We assume some background in Riemannian geometry, and freely use standard notation; we refer the reader to [Jost, 2008, Lee, 2006, Petersen, 2006] for an in depth treatment. Readers may also find some works on Riemannian optimization useful as additional context: [Bacák, 2014, Udriste, 2013, Absil et al., 2009, Zhang and Sra, 2016, Boumal, 2022].

We use $\nabla$ to denote the Levi Civita connection. Given $x, y \in M$ and $v \in T_x M$, we use $\Gamma(v; x \to y)$ to denote parallel transport of $v$ from $x$ to $y$ along their minimizing geodesic (if such a choice is not unique we will specify); we sometimes also use the more concise alternative notation $\Gamma_x^y v := \Gamma(v; x \to y)$. Given a general curve (possibly non-geodesic) $\gamma : [0, 1] \to M$, we will also use $\tilde{\Gamma}_{\gamma(t)} v$ to denote the parallel transport of $v$ from $\gamma(0)$ to $\gamma(1)$, along $\gamma$.

A basis $F$ of the tangent space $T_x M$ at some point $x \in M$ is an ordered tuple $(F^1, \ldots, F^d)$ of vectors that span $T_x M$. We use $\Gamma(F; x \to y) := (\Gamma(F^1; x \to y)...\Gamma(F^d; x \to y))$ to denote the ordered tuple of parallel transport of the each of the basis vectors in $F$. Given $v \in \mathbb{R}^d$ and basis $F$ of some $T_x M$, we use $v \circ F$ as shorthand for $\sum_{i=1}^d v_i F^i$. A distribution that we will see frequently in this paper is the one given by $\boldsymbol{\xi} \circ E^x$, where $\boldsymbol{\xi} \sim \mathcal{N}(0, I)$ is a random vector in $\mathbb{R}^d$, and $x \in M$ and $E^x$ is an orthonormal basis of $T_x M$. We use $\mathcal{N}_x(0, I)$ to denote the distribution of $\boldsymbol{\xi} \circ E^x$. One can verify that $\mathcal{N}_x(0, I)$ does not depend on the choice of basis $E^x$.

## 3.1 An illustrative example: sampling from a sphere

To give some intuition about Assumptions 1 - 4, we present a simple example for sampling from $S^{d-1}$, the unit d-sphere in $\mathbb{R}^d$, which is a positively curved Riemannian manifold.[2] The subsequent bounds for the $S^d$ case also generalize with minor modifications to other closely-related manifolds such as $\mathrm{SO}(d)$.

Let $U(x) : \mathbb{R}^d \to Re$ be a potential function, and suppose that we wish to sample from $dp(x) \propto e^{-U(x)} dvol_g(x)$ defined over $S^{d-1}$. It is known that for any $x \in S^{d-1}, v \in T_x M$, $\mathrm{Ric}(v, v) > 0$,

---

[2]The sphere is one of the special cases when Brownian motion *can be efficiently sampled exactly*, and so if one's actual goal is to sample from a sphere, prior work such as Li and Erdogdu [2020] provide a better algorithm and analysis.

so that Assumption 1 is satisfied with $L_{Ric} = 0$. The sectional curvature is bounded by 1, so that Assumption 4 holds with $L_R = 1$. For the remainder of this example, for $x \in S^{d-1}$ we will identify $T_x M$ with $\{v \in \mathbb{R}^d : v^T x = 0\}$. Since the sphere is a submanifold of $\mathbb{R}^d$, the metric is simply given by the Euclidean dot product, i.e. $g(u, v) = u^T v$.

Next, we will verify Assumption 3. Let us assume that for all $x$ on the sphere, $U(x)$ satisfies $\|\boldsymbol{\nabla} U(x)\| \leq L_1$ and $\left\|\boldsymbol{\nabla^2} U(x)\right\|_2 \leq L_2$, where $\boldsymbol{\nabla}$ and $\boldsymbol{\nabla^2}$ are the usual Euclidean gradient and Hessian respectively (bolded to distinguish from $\nabla$, the covariant derivative). In order to sample from $dp(x)$, $\beta(x) = (I - xx^T)\boldsymbol{\nabla} U(x)$. It is known that a geodesic $\gamma(t)$ corresponds to a great arc on the sphere, and that parallel transport of a vector $v$ involves a rotation of the tangential component of $v$. To be precise, for any $x \in S^d$ and $u, v \in T_x M$ such that $\|u\| = 1$, we verify that, in Cartesian coordinates, $\nabla_u v = (uu^T)v$. Thus $\nabla \beta(x) = (uu^T)(I - xx^T)\boldsymbol{\nabla} U(x) + \boldsymbol{\nabla^2} U(x)u \leq L_1 + L_2$. Thus Assumption 3 holds with $L'_\beta = L_1 + L_2$.

Finally, we will verify Assumption 2. Since $S^{d-1}$ has diameter $\pi$, the first part of Assumption 2 is satisfied with arbitrary $m$ and $R = \pi$. The second part of Assumption 2 is satisfied with $q = L'_\beta$. This is because $\frac{d}{dt} \langle \Gamma(\beta(\gamma(t)); \gamma(t) \to \gamma(0)) - \beta(\gamma(0)), \gamma'(0) \rangle = \frac{d}{dt} \langle \beta(\gamma(t)), \gamma'(t) \rangle = \langle \nabla_{\gamma'(t)} \beta(\gamma(t)), \gamma'(t) \rangle \leq L'_\beta \|\gamma'(t)\|^2$.

# 4  Bounding the Discretization Error in Geometric Euler-Murayama

One of our main technical contributions is an error bound for the Euler-Murayama discretization. We provide an informal statement of this error bound as Lemma 1 below, and provide the formal statement as Lemma 7 in Appendix A.3.

**Lemma 1** (Informal version of Lemma 7). *Let $x(t)$ denote the solution to (1.1) initialized at some $x(0)$. Let $x^0(t)$ denote one step geometric Euler Murayama discretization: $x^0(t) := \mathrm{Exp}_{x(0)}\left(t\beta(x(0)) + \sqrt{t}\zeta\right)$, where $\zeta \sim \mathcal{N}_{x(0)}(0, I)$. Under Assumptions 3 and 4, for sufficiently small $t$, there exists a coupling between $x(t)$ and $x^0(t)$ such that*

$$\mathbb{E}[\mathrm{d}(x(t), x^0(t))^2] \leq O(t^3),$$

*where $O()$ hides polynomial dependence on the Lipschitz constant $L'_\beta$, the sectional curvature of $M$, and dimension $d$.*

**Proof Sketch for Lemma 1**

Given a time interval $T$, we construct a sequence of processes $\{x^i(t)\}_{i \geq 0}$ indexed by $i$ (exact definitinion given in (4.2) below). Marginally, each $x^i(t)$ corresponds to the linear interpolation of a sequence of Euler-Murayama steps with stepsize $\delta^i = 2^{-i}T$. More specifically, for $k \in \mathbb{Z}^+$,

$$x^i_{k+1} = \mathrm{Exp}_{x^i_k}\left(\delta^i \beta(x^i_k) + \sqrt{\delta^i}\zeta^i_k\right) \qquad \zeta^i_k \sim \mathcal{N}_{x^i_k}(0, I)$$

$$x^i(t) = \mathrm{Exp}_{x^i_k}\left(\frac{t - k\delta^i}{\delta^i}(\delta^i \beta(x^i_k) + \sqrt{\delta^i}\zeta^i_k)\right) \qquad \text{for } t \in [k\delta^i, (k+1)\delta^i]$$

Notice that for $i = 0$, $x^0(T)$ is a single step of (1.2) with stepsize $T$. On the other hand, $x(t) := \lim_{i \to \infty} x^i(t)$ is exactly the SDE in (1.1) (see Lemma 2 at the end of this section). We will soon see that $\mathbb{E}[\mathrm{d}(x^i(T), x^{i+1}(T))^2] = O(T^2\delta^i)$. We can then bound $\mathbb{E}[\mathrm{d}(x^0(T), x(T))^2] = O(T^3)$ by summing over the pairwise distance between adjacent $x^i$ and $x^{i+1}$ for $i \in \mathbb{Z}^+$ (this diminishes geometrically). This proves Lemma 1.

To bound the distance between $x^i$ and $x^{i+1}$, we introduce a crucial additional structure: adjacent pairs of processes $(x^i_k, x^{i+1}_k)$ are coupled using the **manifold analog of synchronous coupling**, together with the **discrete-time analog of "rolling without slipping"**. (see Remark 1),

The exact formula for how $x^{i+1}_k$ is to be constructed from $x^i_k$ is given in (4.1) below; it is a little dense, so we provide a pictorial illustration in Figure 1. For simplicity, we assume that $\beta = 0$ (the dominating error in Lemma 1 is due to Brownian motion). Figure 1 takes place over a period of time from $k\delta^i$ to $(k+1)\delta^i$ (equivalently, $2k\delta^{i+1}$ to $(2k+2)\delta^{i+1}$). The black squiggle denotes $(\mathbf{B}(t) - \mathbf{B}(k\delta^i)) \circ E^i_k$ for $t \in [k\delta^i, (k+1)\delta^i]$, where $\mathbf{B}(t)$ is a standard $d$-dimensional Brownian

motion, and $E_k^i$ is some basis at $x_k^i$. Let $b_k^i := (\mathbf{B}((k+1)\delta^i) - \mathbf{B}(k\delta^i)) \circ E_k^i$ (black solid arrow). Bounding the distance between $x^i$ and $x^{i+1}$ consists of two steps:

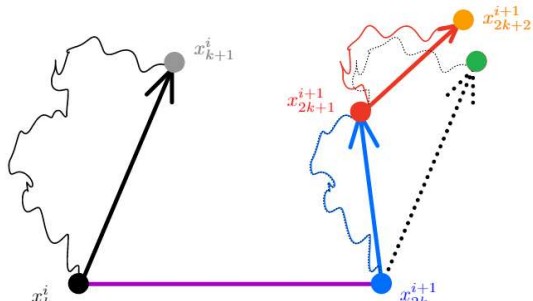

Figure 1: An illustration of the coupling between paths for $x^i$ and $x^{i+1}$ when $\beta = 0$ over the time interval $[k\delta^i, (k+1)\delta^i]$.

**Step 1: Bounding** $d(x_{k+1}^i, \mathrm{Exp}_{x_{2k}^{i+1}}(\Gamma(b_k^i; x_k^i \to x_{2k}^{i+1})))$ [Solid black arrow vs dotted black arrow]
The purple line represents the geodesic from $x_k^i$ to $x_{2k}^{i+1}$; by construction, the basis $E_{2k}^{i+1}$ is the parallel transport of $E_k^i$ along the purple geodesic. The dotted black arrow denotes the $(\mathbf{B}((k+1)\delta^i) - \mathbf{B}(k\delta^i)) \circ E_{2k}^{i+1}$; we verify that it is the paralle transport of $b_k^i$ (black arrow) along the purple geodesic. Using Lemma 28, we can bound the distance between $x_{k+1}^i = \mathrm{Exp}_{x_k^i}(b_k^i)$ (gray point) and $\mathrm{Exp}_{x_{2k}^{i+1}}(\Gamma(b_k^i; x_k^i \to x_{2k}^{i+1}))$ (green point) as

$$\mathbb{E}[d(\mathrm{Exp}_{x_k^i}(b_k^i), \mathrm{Exp}_{x_{2k}^{i+1}}(\Gamma(b_k^i; x_k^i \to x_{2k}^{i+1})))^2] \le (1 + O(L_R d\delta^i))\mathbb{E}[d(x_k^i, x_{2k}^{i+1})^2]$$

**Step 2: Bounding** $d(\mathrm{Exp}_{x_{2k}^{i+1}}(\Gamma(b_k^i; x_k^i \to x_{2k}^{i+1})), x_{2k+2}^{i+1})$ [Dotted black arrow vs blue + red arrow]
Let

$$b_{2k}^{i+1} := (\mathbf{B}((2k+1)\delta^{i+1}) - \mathbf{B}(2k\delta^{i+1})) \circ E_{2k}^{i+1}$$
$$b_{2k+1}^{i+1} := (\mathbf{B}((2k+2)\delta^{i+1}) - \mathbf{B}((2k+1)\delta^{i+1})) \circ E_{2k}^{i+1}$$

We verify that $\Gamma(b_k^i; x_k^i \to x_{2k}^{i+1})$ (black dotted arrow) is equal to $b_{2k}^{i+1} + b_{2k+1}^{i+1}$ (note that $\delta^{i+1} = 1/2\delta^i$). On the other hand, $x_{2k+2}^{i+1}$ (orange point) is obtained from taking a step in $b_{2k}^{i+1}$ direction (blue arrow) followed by a step in the $\Gamma(b_{2k+1}^{i+1}; x_{2k}^{i+1} \to x_{2k+1}^{i+1})$ direction (red arrow). Due to curvature, this is not the same as taking a single step in the $b_k^i = b_{2k}^{i+1} + b_{2k+1}^{i+1}$ direction (green point). However, we can bound the distance between the orange point and the green point using Lemma 3 of Sun et al. [2019] (restated as Lemma 38 for ease of reference):

$$\mathbb{E}[d(\mathrm{Exp}_{x_{2k}^{i+1}}(\Gamma(b_k^i; x_k^i \to x_{2k}^{i+1})), x_{2k+2}^{i+1})^2] \le O(\|b_{2k}^{i+1}\|^2 \|b_{2k+1}^{i+1}\|^2 (\|b_{2k}^{i+1}\| + \|b_{2k+1}^{i+1}\|)^2) = O((\delta^i)^3)$$

Summing the bounds of Step 1 and Step 2 gives $\mathbb{E}[d(x_{k+1}^i, x_{2k+2}^{i+1})^2] \le e^{O(L_R d\delta^i)}\mathbb{E}[d(x_k^i, x_{2k}^{i+1})^2] + O((\delta^i)^3)$. Recursing over $k = 0...2^i$, and assuming that $T \le \frac{1}{L_R d}$, we get $\mathbb{E}[d(x_{2^i}^i, x_{2^{i+1}}^{i+1})^2] = \mathbb{E}[d(x^i(T), x^{i+1}(T))^2] \le O(T^2 \delta^i)$. Summing over $i = 0...\infty$ gives $\mathbb{E}[d(x^i(T), x(T))^2] = O(T^3)$.

**Details for Construction of Brownian Motion**

To bound the discretization error, we first characterize the manifold SDE (1.1) as the limit of a family of random processes. Let $x_0 \in M$ be an initial point and $E = \{E^1, \ldots, E^d\}$ be an orthonormal basis of $T_{x^0}$. Let $\mathbf{B}(t)$ denote a standard Brownian Motion in $\mathbb{R}^d$. Let $T \in \mathbb{R}^+$. Define

$$x_0^0 = x_0, \qquad E_0^0 = E,$$
$$x_1^0 = \mathrm{Exp}_{x_0^0}(T\beta(x_0^0) + (\mathbf{B}(T) - \mathbf{B}(0)) \circ E_0^0).$$

For any $i \in \mathbb{Z}^+$, let $\delta^i := 2^{-i}T$. We will now define points $x_k^i \in M$ and orthonormal basis $E_k^i$ of $T_{x_k^i}$ for all $i$ and all $k \in \{0, \ldots, T/\delta^i\}$. Our construction is inductive: Suppose we have already defined $x_k^i$ and $E_k^i$ for some $i$ and for all $k \in \{0, \ldots, T/\delta^i\}$. Then, we construct $x_k^{i+1}$, for all $k = \{0, \ldots, T/\delta^{i+1}\}$, as follows:

$$x_0^{i+1} := x_0, \qquad E_0^{i+1} := E,$$
$$x_{2k+1}^{i+1} := \mathrm{Exp}_{x_{2k}^{i+1}} \left( \delta^{i+1}\beta(x_{2k}^{i+1}) + (\mathbf{B}((2k+1)\delta^{i+1}) - \mathbf{B}(2k\delta^{i+1})) \circ E_{2k}^{i+1} \right),$$
$$E_{2k+1}^{i+1} := \Gamma(E_{2k}^{i+1}; x_{2k}^{i+1} \to x_{2k+1}^{i+1}),$$
$$x_{2k+2}^{i+1} := \mathrm{Exp}_{x_{2k+1}^{i+1}} \left( \delta^{i+1}\beta(x_{2k+1}^{i+1}) + (\mathbf{B}((2k+2)\delta^{i+1}) - \mathbf{B}((2k+1)\delta^{i+1})) \circ E_{2k+1}^{i+1} \right),$$
$$E_{2k+2}^{i+1} := \Gamma(E_{k+1}^i; x_{k+1}^i \to x_{2k+2}^{i+1}). \tag{4.1}$$

The above display defines points $x_k^{i+1}$ for all $k = \{0, \ldots, T/\delta^{i+1}\}$. For parallel transport, if the minimizing geodesic is not unique, any arbitrary choice will do. We verify that for any $i$, for all $k$, $x_k^i$ is indeed of the form (1.2), i.e. $x_{k+1}^i = \mathrm{Exp}_{x_k^i}(\delta^i\beta(x_k^i) + \sqrt{\delta^i}\zeta_k^i)$, where $\zeta_k^i := \frac{1}{\sqrt{\delta^i}}(\mathbf{B}((k+1)\delta^i) - \mathbf{B}(k\delta^i) \circ E_k^i) \sim \mathcal{N}_{x_k^i}(0, I)$. Finally, for any $i$, any $k$, and any $t \in [k\delta^i, (k+1)\delta^i)$, we define $x^i(t)$ to be the "linear interpolation" of $x_k^i$ and $x_{k+1}^i$, i.e.,

$$x^i(t) := \mathrm{Exp}_{x_k^i}\left( \tfrac{t-k\delta^i}{\delta^i}(\delta^i\beta(x_k^i) + (\mathbf{B}((k+1)\delta^i) - \mathbf{B}(k\delta^i))) \circ E_k^i \right). \tag{4.2}$$

**Remark 1.** The choice of basis $E_k^i$ in (4.1) can be seen as a combination of "synchronous coupling" and "rolling without slipping" (see Chapter 2 of [Hsu, 2002]). In particular, $E_{2k}^{i+1} := \Gamma(E_k^i; x_k^i \to x_{2k}^{i+1})$ corresponds to "synchronous coupling"—the step from $x_{2k}^{i+1}$ to $x_{2k+1}^{i+1}$ is (roughly) parallel to the step from $x_k^i$ to $x_{k+1}^i$. On the other hand, $E_{2k+1}^{i+1} := \Gamma(E_{2k}^{i+1}; x_{2k}^{i+1} \to x_{2k+1}^{i+1})$ corresponds to "rolling without slipping"—the step from $x_{2k+1}^{i+1}$ to $x_{2k+2}^{i+1}$ is with respect to an orthonormal basis that is parallel-transported from $x_{2k}^{i+1}$ to $x_{2k+1}^{i+1}$.

We verify in the following Lemma that the limit, $i \to \infty$, of $x^i(t)$ is the SDE (1.1):

**Lemma 2.** *For any $T$, for $t \in [0, T]$, let $x(t) := \lim_{i \to \infty} x^i(t)$. This limit exists uniformly almost-surely, and $x(t)$ is a diffusion process generated by the operator $L$ whose action on any smooth function $f$ is given by $Lf = \langle \nabla f, \beta \rangle + \frac{1}{2}\Delta(f)$, where $\Delta$ denotes the Laplace Beltrami operator. Thus $x(t)$ is equal to (1.1) in distribution.*

Lemma 2 follows immediately from [Gangolli, 1964, Jørgensen, 1975], but we provide a proof in Appendix A.1 for completeness.

## 5 Langevin MCMC on Riemannian Manifolds

With the one-step discretization error in Lemma 1, we can bound the iteration complexity of Langevin MCMC (1.2):

**Theorem 1** (Convergence of Langevin MCMC on Riemannian Manifold). *Assume the manifold $M$ satisfies Assumptions 1 and 4. Assume in addition that there exists a constant $L_R'$ such that for all $x \in M$, $u, v, w, z, a \in T_x M$, $\langle (\nabla_a R)(u, v)w, z \rangle \leq L_R' \|u\|\|v\|\|w\|\|z\|\|a\|$ (this last assumption is for analytical convenience; $L_R'$ does not show up in the quantitative bounds). Let $\beta$ be a vector field satisfying Assumptions 2 and 3; assume in addition that $m > L_{\mathrm{Ric}}/2$ and that $q + L_{\mathrm{Ric}}/2 \geq 0$. Let $x^*$ be some point with $\beta(x^*) = 0$.*

*Let $y(t)$ denote the exact geometric SDE given in (1.1) initialized at some $y(0)$. Let $K \in \mathbb{Z}^+$ be some iteration number and $\delta$ be some stepsize. Let $x_k$ denote the Euler Murayama discretization of (1.1), defined by $x_{k+1} = \mathrm{Exp}_{x_k}(\delta\beta(x_k) + \sqrt{\delta}\zeta_k)$, where $\zeta_k \sim \mathcal{N}_{x_k}(0, I)$, initialized at some $x_0$ satisfying $\mathrm{d}(x_0, x^*) \leq 2\mathcal{R}$.*

*Then, there exists a constant $\mathcal{C}_0 = poly(L_\beta', d, L_R, \mathcal{R}, \frac{1}{m - L_{\mathrm{Ric}}/2}, \log K)$, such that if $\delta \leq \frac{1}{\mathcal{C}_0}$, then there is a coupling between $x_K$ and $y(K\delta)$ satisfying the distance bound*

$$\mathbb{E}[\mathrm{d}(y(K\delta), x_K)] \leq e^{-\alpha K\delta + (q + L_{\mathrm{Ric}}/2)\mathcal{R}^2/2}\mathbb{E}[\mathrm{d}(y(0), x_0)] + \exp\left((q + L_{\mathrm{Ric}}/2)\mathcal{R}^2\right) \cdot \tilde{O}(\delta^{1/2}).$$

$\tilde{O}$ hides polynomial dependence on $L'_\beta, d, L_R, \mathcal{R}, \frac{1}{m-L_{\text{Ric}}/2}, \log K, \log \frac{1}{\delta}$, and
$$\alpha := \min\left\{\frac{m-L_{\text{Ric}}/2}{16}, \frac{1}{2\mathcal{R}^2}\right\} \cdot e^{-\frac{1}{2}(q+L_{\text{Ric}}/2)\mathcal{R}^2}.$$

We defer the proof of Theorem 1 to Appendix A.4, where we state the explicit expressions for $\mathcal{C}_0$.

**Discussion of Theorem 1**

**To sample from** $d\pi^*(x) = e^{-h(x)}d\text{vol}_g(x)$, we let $\beta(x) := -\frac{1}{2}\nabla h(x)$; under this choice of $\beta$, $\pi^*(x)$ is invariant under the SDE for $y(t)$. Picking $y(0) \sim \pi^*(x)$, we thus ensure that $\text{Law}(y(t)) = \pi^*$ for all $t$. **Therefore, the** $W_1$ **distance between** $\text{Law}(x_K)$ **and** $\pi^*$ **is upper bounded by** $\mathbb{E}[\text{d}(y(K\delta), x_K)]$, which is in turn upper bounded in Theorem 1.

The distance bound consists of two terms: the first term is exponentially small in $K\delta$, so it goes to 0 as the number of steps tends to infinity; the second term is proportional to $\delta^{1/2}$. We first pick $\delta = \tilde{O}(\varepsilon^2)$ so that the second term is bounded by $\varepsilon$. It then suffices to let $K = \frac{1}{\alpha\delta}\log\left(\frac{\mathbb{E}[\text{d}(y(0),x_0)]}{\varepsilon}\right) + (q + L_{\text{Ric}}/2)\mathcal{R}^2 = \tilde{O}(\varepsilon^{-2})$ in order for the first term to be bounded by $\varepsilon$. Note that by our assumptions and by Lemma 18 $\mathbb{E}[\text{d}(y(0), x_0)] \leq poly(L_R, L'_\beta, d, \mathcal{R}, \frac{1}{m})$.

**We now discuss a few specific cases for Assumption 2.** The easiest setting is when $m > L_{\text{Ric}}/2$ and $\mathcal{R} = 0$ (in this case the value of $q$ does not matter); this occurs, for example, if $\beta = -\frac{1}{2}\nabla h$ for some $c$-strongly-convex $h$, and $M$ has positive Ricci curvature. The mixing rate is $\alpha = \frac{c-L_{\text{Ric}}}{32}$, and this closely relates to a well known result by Bakry and Emery on the Log-Sobolev Inequality of $CD(\cdot, \infty)$ distributions:

***Theorem** ([Bakry et al., 2014]) Let Ric denote the Ricci curvature tensor. If $d\pi^*(x) = e^{-h(x)}d\text{vol}_g(x)$, and $\nabla^2 h + \text{Ric} \succ \rho I$, then $\pi^*$ satisfies the Log-Sobolev Inequality (LSI) with parameter $\rho$.*

When $\pi^*$ satisfies the $LSI(\rho)$, it has been shown in [Bakry et al., 2014] that for any initialization, the KL divergence of $\text{Law}(x(t))$ of (1.1) with respect to its stationary distribution converges to 0 with rate $\rho$. In our example in the preceding paragraph, $d\pi^* = e^{-h}d\text{vol}_g$ satisfies LSI with $\rho = c - L_{\text{Ric}}$, which is, up to a constant factor, equal to our mixing rate.

The requirement $q + L_{\text{Ric}}/2 \geq 0$ is without loss of generality. If $q + L_{\text{Ric}}/2 < 0$, we can take $q' = -L_{\text{Ric}}/2$ and verify that $\beta$ satisfies Assumption 2 with $(m, q', \mathcal{R})$. Since $q' + L_{\text{Ric}}/2 = 0$, the mixing rate is then $\alpha = \min\left\{\frac{m-L_{\text{Ric}}/2}{16}, \frac{1}{2\mathcal{R}^2}\right\}$, which corresponds to the easy "$CD(0, \infty)$" setting.

**A harder, but more general setting** is when $\beta = \frac{1}{2}\nabla h$ for some non-convex $h$, and $L_{\text{Ric}} > 0$, i.e. the manifold can have negative Ricci curvature. We will still need to assume that $\nabla h(x)$ is contractive for points which are further away than some radius $\mathcal{R}$; otherwise, the SDE (1.1) may drift off to infinity and $e^{-h}$ may not be integrable. Under Assumption 3, $\beta$ satisfies Assumption 2 with $m, L'_\beta, \mathcal{R}$. The mixing rate $\alpha$ is then proportional to $e^{-(L'_\beta+L_{\text{Ric}}/2)\mathcal{R}^2}$; this can be very small if $h$ is nonconvex and highly nonsmooth, and the manifold $M$ has large negative Ricci curvature, leading to very slow mixing. This is generally unavoidable, even when $M$ is the flat Euclidean space.

Readers familiar with the Holly-Stroock perturbation may find the $\exp\left((q + L_{Ric}/2)\mathcal{R}^2\right)$ term familiar: Let $M$ have diameter $\mathcal{R}$ and let $U$ have $2q$-Lipschitz gradient, so that $\beta = \frac{1}{2}\nabla U$ satisfies Assumption 2 with $q$. Then we can decompose $U = U_1 + U_2$, where $U_1$ is $-L_{\text{Ric}}$-strongly-convex and $U_2$ is a "perturbation" with magnitude $(2q + L_{\text{Ric}})\mathcal{R}^2$. $e^{2U_1}d\text{vol}_g$ satisfies $CD(0, \infty)$ and thus $e^{2U}d\text{vol}_g$ has Log-Sobolev constant scaling with $\exp\left((2q + L_{Ric})\mathcal{R}^2\right)$.

**Finally, on compact manifolds without boundary,** one may take $\mathcal{R}$ to be the diameter of the manifold and upper bound the mixing rate $\alpha$ by $\alpha = \frac{1}{2\mathcal{R}^2}e^{-\frac{1}{2}(q+L_{\text{Ric}}/2)\mathcal{R}^2}$.

## 5.1 Proof Sketch of Theorem 1

The proof of Theorem 1 consists of two steps. The first step is bounding the discretization error of (1.2); we have already done this in Lemma 1 in the previous section.

The second step is showing that two paths of (1.1) converge. To do so, we consider $f(\text{d}(x(t), y(t)))$, where $f$ is a Lyapunov function taken from [Eberle, 2016] (its exact form is given in Definition 2 in

the Appendix). Two key property of $f$ are that (i) $\frac{1}{2}\exp\left(-(q+L_{\mathrm{Ric}}/2)\mathcal{R}^2/2\right)r \le f(r) \le r$; and (ii) $f'(r) \le 1$, so that convergence in $f(\mathrm{d}(x(t),y(t)))$ implies convergence in $\mathrm{d}(x(t),y(t))$.

Under the *Kendall-Cranston Coupling* [3] of $x(t)$ and $y(t)$, one can show that $f(\mathrm{d}(x(t),y(t)))$ contracts with rate $\alpha$. We quantify the contraction rate in Lemma 3 below. We stress that Lemma 3 is not new; it was first presented by Eberle [2016] (with minor variation), and its proof combines Theorem 6.6.2 of [Hsu, 2002] with the Lyapunov function analysis of [Eberle, 2016]. For completeness, we provide a proof of Lemma 3 in Appendix B.3.

**Lemma 3.** *Assume $\beta$ is $(m,q,\mathcal{R})$-distant dissipative as per Assumption 2, and that is also satisfies Assumption 3. Further assume that $m > L_{\mathrm{Ric}}/2$ and $q+L_{\mathrm{Ric}}/2 \ge 0$. Let $x(t)$ and $y(t)$ denote solutions to (1.1). Then there exists a Lyapunov function $f$ satisfying 1. $f(r) \ge \frac{1}{2}\exp\left(-(q+L_{\mathrm{Ric}}/2)\mathcal{R}^2/2\right)r$ and 2. $|f'(r)| \le 1$, and a coupling between $x(t)$ and $y(t)$, such that for all time $T$,*

$$\mathbb{E}[f(\mathrm{d}(x(T),y(T)))] \le \exp\left(-\alpha T\right)f(\mathrm{d}(x_0,y_0)),$$

*where $\alpha := \min\left\{\frac{m-L_{\mathrm{Ric}}/2}{16}, \frac{1}{2\mathcal{R}^2}\right\} \cdot \exp\left(-\frac{1}{2}(q+L_{\mathrm{Ric}}/2)\mathcal{R}^2\right)$.*

**Given these two steps**, we can now sketch the proof of Theorem 1. Consider an arbitrary step $k$ of (1.2). For $t \in [k\delta,(k+1)\delta)$, let $\bar{x}(t)$ denote the solution to (1.1), initialized at $\bar{x}(k\delta)$. Then by Lemma 3, $\mathbb{E}[f(\mathrm{d}(y((k+1)\delta),\bar{x}((k+1)\delta)))] \le e^{-\alpha\delta}\mathbb{E}[f(\mathrm{d}(y(k\delta),x_k))]$. On the other hand, by Lemma 1 $\mathbb{E}[\mathrm{d}(\bar{x}((k+1)\delta),x_{k+1})] \le O(\delta^{3/2})$. Summing these two bounds, applying triangle inequality, and using the fact that $f'(r) \le 1$, we get

$$\mathbb{E}[f(\mathrm{d}(y((k+1)\delta),x_{k+1}))] \le e^{-\alpha\delta}\mathbb{E}[f(\mathrm{d}(y(k\delta),x_k))] + O(\delta^{3/2}) \tag{5.1}$$

Applying the above recursively for $k = 0...K$, we get

$$\mathbb{E}[f(\mathrm{d}(x_K,y(K\delta)))] \le e^{-\alpha K\delta}\mathbb{E}[f(\mathrm{d}(x_0,y(0)))] + \frac{1}{\alpha}O(\delta^{1/2})$$

Theorem 1 thus follows from the bounds $\frac{1}{2}\exp\left(-(q+L_{\mathrm{Ric}}/2)\mathcal{R}^2/2\right)r \le f(r) \le r$.

Lastly, we briefly mention a complication in the full proof that we omitted from the above sketch: the discretization error between $\bar{x}((k+1)\delta)$ and $x_{k+1}$ grows with $\|\beta(x_k)\|$, which does not have a global upper bound. In practice, we can verify that $\mathrm{d}(x_k,x^*)$ is sub-Gaussian, so that with high probability $\mathrm{d}(x_k,x^*) \le O(\log(k\delta))$; this in turn allows us to bound $\|\beta(x_k)\|$ via Assumption 3. This is the reason for the dependency on $\log K$ and $\log(1/\delta)$ in the iteration complexity of Theorem 1.

# 6  Stochastic Gradient Langevin MCMC

Finally, we bound the iteration complexity of process (2.1), which takes the Euler-Murayama scheme (1.2), and replaces $\beta(x_k)$ at step $k$ by a stochastic estimate $\tilde{\beta}_k(x_k)$.

**Theorem 2** (Convergence of SGLD on Riemannian Manifold). *Assume the manifold $M$ satisfies Assumptions 1 and 4. Assume in addition that there exists a constant $L_R'$ such that for all $x \in M$, $u,v,w,z,a \in T_xM$, $\langle(\nabla_a R)(u,v)w,z\rangle \le L_R'\|u\|\|v\|\|w\|\|z\|\|a\|$. Let $\tilde{\beta}$ be a vector field satisfying Assumptions 2 and 3; assume in addition that $m > L_{\mathrm{Ric}}/2$ and that $\mathcal{R} = 0$.*

*Let $y(t)$ denote the exact geometric SDE given in (1.1). Let $x_k$ denote the stochastic Euler Murayama discretization (2.1), where $\tilde{\beta}_k$ denote independent random vector fields, with $\mathbb{E}[\tilde{\beta}(x)] = \beta(x)$ for all $x$ and $\|\beta(x) - \tilde{\beta}_k(x)\| \le \sigma$ with probability 1.*

*Let $K \in \mathbb{Z}^+$ be some iteration number and $\delta$ be some stepsize. There exists a constant $\mathcal{C}_1 = poly(L_\beta', d, L_R, L_R', \frac{1}{m}, \log K)$, such that if $\delta \le \frac{1}{\mathcal{C}_1}$, then there is a coupling between $x_K$ and $y(K\delta)$ satisfying the distance bound*

$$\mathbb{E}[\mathrm{d}(y(K\delta),x_K)^2] \le e^{-\frac{1}{8}(m-L_{\mathrm{Ric}}/2)K\delta}\mathbb{E}[\mathrm{d}(y(0),x_0)^2] + \tilde{O}(\delta).$$

*$\tilde{O}$ hides polynomial dependence on $L_\beta', d, L_R, L_R', \sigma, \frac{1}{m-L_{\mathrm{Ric}}/2}, \log K, \log\frac{1}{\delta}$.*

We defer the proof of Theorem 2 to Appendix A.5, where we state the explicit expressions for $\mathcal{C}_1$.

---

[3]This is the manifold analog of reflection coupling of Euclidean Brownian motions; see [Hsu, 2002, Ch. 6.5]

**Discussion of Theorem 2**

**To sample from** $d\pi^*(x) = e^{-h(x)}d\mathrm{vol}_g(x)$, we let $\beta(x) := -\frac{1}{2}\nabla h(x)$ and let $y(0) \sim \pi^*(x)$. The $W_2$ distance between $\mathrm{Law}(x_K)$ and $\pi^*$ is upper bounded by $\sqrt{\mathbb{E}[\mathrm{d}(y(K\delta), x_K)^2]}$. To achieve $\varepsilon$ error in $W_2$, Theorem 2 requires $\tilde{O}(\varepsilon^{-2})$ steps. The reasoning is very similar to Theorem 1.

As already discussed in Section 2, **the Assumption on Theorem 2 is considerably more restrictive than Theorem 1.** We also highlight that the error bound in Theorem 2 *does depend on $L'_R$, the derivative of the Riemannain curvature tensor*. This is in contrast to Theorem 1, where $L'_R$ does not appear in any of the bounds.

Though the iteration complexity for Theorem 2 can be larger than Theorem 1 as it depends on additional parameters $\sigma$ and $L'_R$, (2.1) **may nonetheless be faster than** (1.2). For example, let $h(x) = \frac{1}{N}\sum_{i=1}^{N}h_i(x)$ and assume $\|\nabla h_i - \nabla h_j\| \leq \sigma$ for all $i, j$, one step of (2.1) can be performed with a single gradient computation for a single uniformly sampled $h_i$, whereas (1.2) would require $N$ gradient computations.

**Proof Sketch of Theorem 2 and Theoretical Highlights**

For any step $k$, let us define $\bar{y}_{k+1} := \mathrm{Exp}_{y(k\delta)}\left(\delta\beta(y(k\delta)) + \sqrt{\delta}\bar{\zeta}_k\right)$, where $\bar{\zeta}_k \sim \mathcal{N}_{y_k}(0, I)$. We show, in Lemma 8, that $\mathbb{E}[\mathrm{d}(x_{k+1}, \bar{y}_{k+1})^2] \leq (1 - \frac{\delta}{4}(m - L_{\mathrm{Ric}}/2))\mathbb{E}[\mathrm{d}(x_k, y(k\delta))^2] + 16\delta^2\sigma^2$. On the other hand, once again applying Lemma 1, we can verify that $\mathbb{E}[\mathrm{d}(y((k+1)\delta), \bar{y}_{k+1})^2] \leq O(\delta^3)$. By Young's inequality and triangle Inequality,

$$\mathbb{E}[\mathrm{d}(x_{k+1}, y((k+1)\delta))^2] \leq (1 - \frac{\delta}{8}(m - L_{\mathrm{Ric}}/2))\mathbb{E}[\mathrm{d}(x_k, y(k\delta))^2] + O(\delta^2)$$

The bound in Theorem 2 follows immediately from applying the above recursively for $k = 1...K$. We note that The contraction in Lemma 8 is in turn derived from Lemma 29, which may be of independent interest as it quantifies the distance evolution between two general discrete-time stochastic processes (which do not have to be diffusions or related to the Gaussian noise).

It is elucidative to compare the proof structure of Theorem 2 with that of Theorem 1 above. On a high level, at step $k$, Theorem 1 first approximates the discrete MCMC step ($x_{k+1}$) by an exact SDE initialized at $x_k$ ($\bar{x}((k+1)\delta)$). It then uses two facts: 1. $f(\mathrm{d}(\bar{x}(t), y(t)))$ *contracts under the **exact SDE***, and 2. the approximation error between $x_{k+1}$ and $\tilde{x}((k+1)\delta)$ is small. In contrast, at step $k$, Theorem 2 first approximates the *exact SDE* ($y((k+1)\delta)$) by a Euler-Murayama step ($\bar{y}_{k+1}$). It then uses two facts: 1. $\mathrm{d}(x_{k+1}, \bar{y}((k+1)\delta))$ *contracts under the **stochastic Euler-Murayama step***, and 2. the approximation error between $\bar{y}_{k+1}$ and $y((k+1)\delta)$ is small. The reason for this change is to make use of the fact that $\mathbb{E}[\tilde{\beta}_k] = \beta$, *conditioned on the randomness up to time $k\delta$*. Suppose we had followed the proof of Theorem 1 and defined $\bar{x}(t)$ to be the solution to the exact SDE, with drift $\tilde{\beta}_k$, then showing contraction of the exact SDE becomes very tricky as $\mathbb{E}[\tilde{\beta}_k]$ *is not equal to $\beta$* when conditioned on $\bar{x}(t)$ for any $t > k\delta$.

# 7 Acknowledgements

Xiang Cheng acknowledge support from NSF BIGDATA grant (1741341) and NSF CCF-2112665 (TILOS AI Research Institute). Jingzhao Zhang acknowledges support by Tsinghua University Initiative Scientific Research Program.

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
