# Contents of the Appendices

# A Manifold SDE

An outline of this section is as follows:

1. In Section A.1, we prove Lemma 2, which guarantees that $x^i(t)$ defined in (A.2) has a limit $x(t)$ that equals the solution of the exact Langevin Diffusion in (1.1).

2. In Section A.3, we prove Lemma 7, which bounds the distance between $x^0(t)$ from (A.2) and the limit $x(t)$. This is equivalent to bounding the distance between the Euler Murayama discretization (1.2) and the exact Langevin Diffusion (1.1).

3. In Section A.4, we prove Theorem 1.

4. In Section A.5, we prove Theorem 2.

We also list below the key lemmas which are used to prove the results above.

1. Theorem 1 relies on Lemma 3 (contraction of Lyapunov function under exact SDE) and Lemma 7 (bound on Euler Murayama discretization error).
   (a) Lemma 7 essentially sums the bound from Lemma 4.
   (b) Lemma 4 relies on Lemma 28 and Lemma 38.
   (c) Lemma 3 relies on Lemma 29

2. Theorem 2 relies on Lemma 8 (contraction of Lyapunov function under stochastic gradient Euler-Murayama step) and Lemma 7 (bound on Euler Murayama discretization error).
   (a) Lemma 8 relies on Lemma 29.

## A.1 SDE Construction

In this section, we state and prove key lemmas related to our construction in Section 4, which we reproduce below for ease of reference: Let $x_0 \in M$ be an initial point and $E = \{E^1, \ldots, E^d\}$ be an orthonormal basis of $T_{x^0}$. Let $\mathbf{B}(t)$ denote a standard Brownian Motion in $\mathbb{R}^d$. Let $T \in \mathbb{R}^+$. Define

$$x_0^0 = x_0, \qquad E_0^0 = E,$$
$$x_1^0 = \mathrm{Exp}_{x_0^0}(T\beta(x_0^0) + (\mathbf{B}(T) - \mathbf{B}(0)) \circ E_0^0).$$

For any $i \in \mathbb{Z}^+$, let $\delta^i := 2^{-i}T$. We will now define points $x_k^i \in M$ and orthonormal basis $E_k^i$ of $T_{x_k^i}$ for all $i$ and all $k \in \{0, \ldots, T/\delta^i\}$. Our construction is inductive: Suppose we have already defined $x_k^i$ and $E_k^i$ for some $i$ and for all $k \in \{0, \ldots, T/\delta^i\}$. Then, we construct $x_k^{i+1}$, for all $k = \{0, \ldots, T/\delta^{i+1}\}$, as follows:

$$x_0^{i+1} := x_0, \qquad E_0^{i+1} := E,$$
$$x_{2k+1}^{i+1} := \mathrm{Exp}_{x_{2k}^{i+1}}(\delta^{i+1}\beta(x_{2k}^{i+1}) + (\mathbf{B}((2k+1)\delta^{i+1}) - \mathbf{B}(2k\delta^{i+1})) \circ E_{2k}^{i+1}),$$
$$E_{2k+1}^{i+1} := \Gamma(E_{2k}^{i+1}; x_{2k}^{i+1} \to x_{2k+1}^{i+1}),$$
$$x_{2k+2}^{i+1} := \mathrm{Exp}_{x_{2k+1}^{i+1}}(\delta^{i+1}\beta(x_{2k+1}^{i+1}) + (\mathbf{B}((2k+2)\delta^{i+1}) - \mathbf{B}((2k+1)\delta^{i+1})) \circ E_{2k+1}^{i+1}),$$
$$E_{2k+2}^{i+1} := \Gamma(E_{k+1}^i; x_{k+1}^i \to x_{2k+2}^{i+1}). \tag{A.1}$$

The above display defines points $x_k^{i+1}$ for all $k = \{0, \ldots, T/\delta^{i+1}\}$. For any $i$, any $k$, and any $t \in [k\delta^i, (k+1)\delta^i)$, we define $x^i(t)$ to be the "linear interpolation" of $x_k^i$ and $x_{k+1}^i$, i.e.,

$$x^i(t) := \mathrm{Exp}_{x_k^i}\left(\tfrac{t-k\delta^i}{\delta^i}(\delta^i\beta(x_k^i) + (\mathbf{B}((k+1)\delta^i) - \mathbf{B}(k\delta^i)) \circ E_k^i)\right). \tag{A.2}$$

Let us define two convenient notation that we will use throughout our proofs in this Appendix. First, let

$$\overline{\Phi}(t; x, E, \beta, \mathbf{B}, i) \tag{A.3}$$

denote the solution to the interpolated process in (4.2) (reproduced in (A.2)), initialized at $x_0 = x$, i.e. $\overline{\Phi}(t; x_0, E, \beta, \mathbf{B}, i) = x^i(t)$ as defined in (A.2). (This notation becomes convenient later on when

we need to refer to (4.2) but with different initial points, or with drift vector fields other than $\beta$, or with specific choices of $\mathbf{B}$.)

We also let

$$\Phi(t; x, E, \beta, \mathbf{B}) := \lim_{i \to \infty} \overline{\Phi}(t; x, E, \beta, \mathbf{B}, i). \tag{A.4}$$

Below, we prove Lemma 2 (stated at the end of Section 4), which guarantees that $x(t) = \Phi(t; x, E, \beta, \mathbf{B})$ exists, and that $x(t)$ is a solution to the exact Langevin diffusion SDE in (1.1).

*Proof of Lemma 2.* The existence of the almost-sure, uniform limit $x(t)$ is proven in Lemma 5. For the rest of this proof, we verify that $x(t)$ has the generator $Lf = \langle \nabla f, \beta \rangle + \frac{1}{2}\Delta(f)$, where $\Delta$ denotes the Laplace Beltrami operator. By Proposition 3.2.1 of [Hsu, 2002], this implies that $x(t)$ is the solution to (1.1).

Let $\mathcal{F}_t$ denote the sigma field generated by $\mathbf{B}(s) : s \in [0, t]$.

Consider any $f : M \to \mathbb{R}$ with $\|f'\| \le C$, $\|f''\| \le C$, $\|f'''\| \le C$ globally. Let $x^i(t)$ be as defined in (4.2). We will verify that $f(x(t)) - f(x(0)) - \int_0^t Lf(x(t))dt$ is a martingale.

To begin, let $s, t \in [0, T]$ be such that $s = j\delta^a$ and $t = j'\delta^a$ for some positive integers $j \le j'$, and $a$, (recall that $\delta^i = T/2^i$). We will show that conditioned on $x(s)$, $f(x(t)) - f(x(s)) - \int_s^t Lf(x(t))dt$ is a martingale. Let us define

$$u_k^i = \delta^i \beta(x_k^i) + (\mathbf{B}((k+1)\delta^i) - \mathbf{B}(k\delta^i)) \circ E_k^i$$

so that $x^i(t) = \mathrm{Exp}_{x_k^i}(\frac{t - k\delta^i}{\delta^i} u_k^i)$, where $t \in [k\delta^i, (k+1)\delta^i]$.

Consider an arbitrary $\ell \ge a$. Consider the sum

$$\sum_{k=s/\delta^i}^{t/\delta^i - 1} f(x^\ell((k+1)\delta^\ell)) - f(x^\ell(k\delta^\ell)) - \langle u_k^\ell, \nabla f(x^\ell(k\delta^\ell)) \rangle - \nabla^2 f(x_k^\ell)[u_k^\ell, u_k^\ell]. \tag{A.5}$$

By Taylor's theorem,

$$\left| f(x^\ell((k+1)\delta^\ell)) - f(x^\ell(k\delta^\ell)) - \langle u_k^\ell, \nabla f(x^\ell(k\delta^\ell)) \rangle - \nabla^2 f(x_k^\ell)[u_k^\ell, u_k^\ell] \right|$$
$$\le C\|u_k^\ell\|^3$$
$$\le 8C^3 \delta^{\ell 3} \|\beta(x_k^\ell)\|^3 + 8C^3 \|\mathbf{B}((k+1)\delta^\ell) - \mathbf{B}(k\delta^\ell)\|_2^3,$$

where $C\|u_k^\ell\|^3$ captures the third-and-higher order Taylor terms.

The first order Taylor term can be decomposed as

$$\langle u_k^\ell, \nabla f(x^\ell(k\delta^\ell)) \rangle = \delta^\ell \underbrace{\langle \beta(x^\ell), \nabla f(x^\ell(k\delta^\ell)) \rangle}_{\|\cdot\|^2 \le \delta^{\ell 2} C^2 \|\beta(x^\ell(k\delta^\ell))\|} + \underbrace{\langle \nabla f(x^\ell(k\delta^\ell)), (\mathbf{B}((k+1)\delta^\ell) - \mathbf{B}(k\delta^\ell)) \circ E_k^\ell \rangle}_{\mathbb{E}_{\mathcal{F}_{k\delta^\ell}}[\cdot] = 0}.$$

We now simplify the second order Taylor term. Let $v := (\mathbf{B}((k+1)\delta^i) - \mathbf{B}(k\delta^i)) \circ E_k^i$. We verify that $\mathbb{E}_{\mathcal{F}_{k\delta^\ell}}\left[ \nabla^2 f(x^\ell(k\delta^\ell))[v, v] - \delta^\ell \Delta f(x^\ell(k\delta^\ell)) \right] = 0$, because $(\mathbf{B}((k+1)\delta^i) - \mathbf{B}(k\delta^i)) \circ E_k^i$ has identity covariance, and the Laplace Beltrami operator is the trace of the Hessian. We can also bound, using Young;s inequality,

$$\mathbb{E}\left[ \left| \nabla^2 f(x^\ell(k\delta^\ell))[u_k^\ell, u_k^\ell] - \delta^\ell \Delta f(x^\ell(k\delta^\ell)) \right|^2 \right]$$
$$\le \delta^{\ell 4} C^2 \|\beta(x^\ell(k\delta^\ell))\|^4 + \delta^{\ell 2} \|\mathbf{B}((k+1)\delta^\ell) - \mathbf{B}(k\delta^\ell)\|_2^4 + \delta^{\ell 2} C^2 d$$

Finally, note that there exists a constant $C'$, which depends on $T, d, L'_\beta \|\beta(x_0)\|$, such that for all $\ell$, for all $t \in [0, T]$, $\mathbb{E}\left[ \|\beta(x^\ell(t)\|^6 \right] \le C'$. The proof is similar to Lemma 16 and we omit it here.

Plugging into (A.5) and taking expectation conditioned on the Brownian motion $\mathbf{B}(t) : t \in [0, s]$, we get that

$$\mathbb{E}_{\mathcal{F}_s}\left[\left|\left|f(x^\ell(t)) - f(x^\ell(s)) + \sum_{k=s/\delta^\ell}^{t/\delta^\ell - 1} -\delta^\ell \left\langle \beta(x^\ell(k\delta^\ell)), \nabla f(x^\ell(k\delta^\ell)) \right\rangle - \frac{\delta^\ell}{2}\Delta f(x^\ell(k\delta^\ell))\right|\right|^2\right]$$

$$=\mathbb{E}_{\mathcal{F}_s}\left[\left|\sum_{k=s/\delta^\ell}^{t/\delta^\ell - 1} f(x^\ell((k+1)\delta^\ell)) - f(x^\ell(k\delta^\ell)) - \delta^\ell \left\langle \beta(x^\ell(k\delta^\ell)), \nabla f(x^\ell(k\delta^\ell)) \right\rangle - \frac{\delta^\ell}{2}\Delta f(x^\ell(k\delta^\ell))\right|^2\right]$$

$$\leq \text{poly}(C, C', d) \sum_{k=s/\delta^\ell}^{t/\delta^\ell - 1} \delta^{\ell 2}$$

$$\leq \text{poly}(C, C', d, T)\delta^\ell.$$

where the first line is because $f(x^\ell(t)) - f(x^\ell(s)) = \sum_{k=s/\delta^\ell}^{t/\delta^\ell - 1} f(x^\ell((k+1)\delta^\ell)) - f(x^\ell(k\delta^\ell))$, noting that $t, s$ are multiples of $\delta^\ell$ by definition, and the second line uses our first and second Taylor approximation bounds above.

Next, define $g_k^\ell := \delta^\ell \left\langle \beta(x^\ell(k\delta^\ell)), \nabla f(x^\ell(k\delta^\ell)) \right\rangle + \frac{\delta^\ell}{2}\Delta f(x^\ell(k\delta^\ell))$. Using the smoothness of $\beta$ and $f$, and the fact that $\text{d}(x^\ell(t), x(k\delta^\ell)) \leq \delta^\ell \|\beta(x(k\delta^\ell))\| + \|\mathbf{B}((k+1)\delta^\ell) - \mathbf{B}(k\delta^\ell)\|_2$, we verify that for any $k$,

$$\mathbb{E}_{\mathcal{F}_s}\left[\left|g_k^\ell - \int_{k\delta^\ell}^{(k+1)\delta^\ell} \left\langle \beta(x(r)), \nabla f(x(r)) \right\rangle + \frac{1}{2}\Delta f(x(r))dr\right|\right] \leq \text{poly}(C, C', d)\delta^{\ell 3/2}$$

Putting everything together, we obtain the bound

$$\mathbb{E}_{\mathcal{F}_s}\left[\left|f(x^\ell(t)) - f(x^\ell(s)) + \int_s^t -\left\langle \beta(x^\ell(r)), \nabla f(x^\ell(r)) \right\rangle - \frac{1}{2}\Delta f(x^\ell(r))dr\right|\right] \leq \text{poly}(C, C', d, T)\delta^{\ell 1/2}.$$

By Lemma 5, $\sup_{t\in[0,T]} \text{d}(x^\ell(t), x(t))$ converges to 0 almost surely as $\ell \to \infty$. By Dominated Convergence Theorem, and by smoothness of $f$ and $\beta$,

$$\mathbb{E}_{\mathcal{F}_s}\left[\left|f(x(t)) - f(x(s)) + \int_s^t -\left\langle \beta(x(r)), \nabla f(x(r)) \right\rangle - \frac{1}{2}\Delta f(x(r))dr\right|\right]$$

$$= \lim_{\ell\to\infty} \mathbb{E}_{\mathcal{F}_s}\left[\left|f(x^\ell(t)) - f(x^\ell(s)) + \int_s^t -\left\langle \beta(x^\ell(r)), \nabla f(x^\ell(r)) \right\rangle - \frac{1}{2}\Delta f(x^\ell(r))dr\right|\right]$$

$$= 0.$$

The last equality holds because $\lim_{\ell\to\infty} \text{poly}(C, C', d, T)\delta^{\ell 1/2} = 0$.

Recall that we assumed that $s$ and $t$ are integral multiples of $T/2^a$ for some positive integer $a$. To extend to general $s, t$, we note that the set of dyadic points (i.e. multiples of $T/2^a$, for some integer $a$) is uniformly dense on the real line.

$\square$

## A.2 Existence of Limit

We present below Lemma 4, which bounds the distance between two adjacent trajectories $x^i(t)$ and $x^{i+1}(t)$ as defined in (4.2) (or equivalently (A.2)). The proof of Lemma 4 works by combining Lemma 28 (which bounds distance evolution under "synchronous coupling"), and Lemma 38 (which bounds distance evolution under "rolling without slipping"; Lemma 38 is taken from [Sun et al., 2019]). The proof of Lemma 4 corresponds to Step 1 and Step 2 of the proof sketch of Lemma 1 in Section 4.

Lemma 4 plays a key role bounding the Euler Murayama discretization error in Lemma 7 in Section A.3. The proof of Lemma 7 essentially involves summing the bound from Lemma 4, for $i = 0...\infty$.

Another application of Lemma 4 is to verify the existence of $x(t) = \lim_{i \to \infty} x^i(t)$ as defined in (A.4) in Lemma 5.

**Lemma 4.** *Let $T$ be any positive constant. Let $x^i(t)$ be the (interpolation) of the Euler Murayama discretization with stepsize $\delta^i = T/2^i$ as defined in (4.2) (or equivalently (A.2)). Let $K := 2^i$ so that $T = K\delta^i$.*

*Assume that there is are constants $L_\beta, L'_\beta$ such that for all $x, y \in M$, $\|\beta(x)\| \le L_\beta$ and $\left\|\beta(x) - \Gamma_y^x \beta(y)\right\| \le L'_\beta \|x - y\|$. Then*

$$
\mathbb{E}\left[\sup_{t \in [0, K\delta^i]} \mathrm{d}(x^i(t), x^{i+1}(t))^2\right]
$$
$$
\le 2^{10} \cdot e^{40K\delta^{i^2} L_R L_\beta^2 + 2K\delta^i L_R d + K\delta^i L'_\beta} (K\delta^i)^2 (\delta^{i^4} L_R^2 L_\beta^6 + \delta^i L_R^2 d^3 + \delta^{i^2} L'^2_\beta L_\beta^2 + \delta^i L'^2_\beta d),
$$

*and*

$$
\mathbb{P}(\sup_{t \in [0, T]} \mathrm{d}(x^i(t), x^{i+1}(t)) \ge 2^{-\frac{i}{4} - 2})
$$
$$
\le e^{(2^{6-i} T L_R L_\beta^2 + 2L_R d + L'_\beta) T} T^2 \cdot (\delta^{i^3} L_R^2 L_\beta^6 + L_R^2 d^3 + \delta^i L'^2_\beta L_\beta^2 + L'^2_\beta d) \cdot 2^{-i/2 + 14}.
$$

**Remark:** Lemma 4 is usually applied with $T$ being the step-size of a a single Euler-Murayama discretization step (i.e. $\delta$ in (1.2)). Therefore, by taking $T$ to be sufficiently small, the exponential term can be made small, e.g. $\le 2$.

*Proof.* Recall that $x^i(t)$ is the linear interpolation of $x_k^i$ as defined in (4.1).

Let us define

$$
a_k := \delta^{i+1} \beta(x_{2k}^{i+1}) + (\mathbf{B}((2k+1)\delta^{i+1}) - \mathbf{B}(2k\delta^{i+1})) \circ E_{2k}^{i+1}
$$
$$
b_k := \delta^{i+1} \beta(x_{2k}^{i+1}) + (\mathbf{B}((2k+2)\delta^{i+1}) - \mathbf{B}((2k+1)\delta^{i+1})) \circ E_{2k}^{i+1} \tag{A.6}
$$

Our proof breaks down the bound of $\mathrm{d}(x_k^i, x_{2k+2}^{i+1})$ into two parts: by Young's inequality,

$$
\mathrm{d}(x_{k+1}^i, x_{2k+2}^{i+1})^2 \le (\mathrm{d}(x_{k+1}^i, \mathrm{Exp}_{x_{2k}^{i+1}}(a_k + b_k)) + \mathrm{d}(\mathrm{Exp}_{x_{2k}^{i+1}}(a_k + b_k), x_{2k+2}^{i+1}))^2
$$
$$
\le (1 + \frac{1}{2K})\mathrm{d}(x_{k+1}^i, \mathrm{Exp}_{x_{2k}^{i+1}}(a_k + b_k))^2 + K\mathrm{d}(\mathrm{Exp}_{x_{2k}^{i+1}}(a_k + b_k), x_{2k+2}^{i+1})^2
$$
$$\tag{A.7}
$$

We now bound the first term of (A.7). From definition in (4.1) and (A.6),

$$
x_{k+1}^i = \mathrm{Exp}_{x_k^i}(\delta^i \beta(x_k^i) + (\mathbf{B}((k+1)\delta^i) - \mathbf{B}(k\delta^i)) \circ E_k^i)
$$
$$
\mathrm{Exp}_{x_{2k}^{i+1}}(a_k + b_k) = \mathrm{Exp}_{x_k^i}(\delta^i \beta(x_{2k}^{i+1}) + (\mathbf{B}((2k+2)\delta^{i+1}) - \mathbf{B}(2k\delta^{i+1})) \circ E_{2k}^{i+1})
$$

We thus apply Lemma 28, with $x := x_k^i$, $y := x_{2k}^{i+1}$, $u := \delta^i \beta(x_k^i) + (\mathbf{B}((k+1)\delta^i) - \mathbf{B}(k\delta^i)) \circ E_k^i$, $v := \delta^i \beta(x_{2k}^{i+1}) + (\mathbf{B}((2k+2)\delta^{i+1}) - \mathbf{B}(2k\delta^{i+1})) \circ E_k^{i+1}$. Let $\gamma(t), u(t), v(t)$ be as defined in Lemma 28. Then Lemma 28 bounds

$$
\mathrm{d}(\mathrm{Exp}_x(u), \mathrm{Exp}_y(v))^2 \le (1 + 4\mathcal{C}_k^2 e^{4\mathcal{C}_k})\mathrm{d}(x, y)^2 + 32 e^{\mathcal{C}_k} \|v(0) - u(0)\|^2 + 2\langle \gamma'(0), v(0) - u(0)\rangle
$$
$$\tag{A.8}
$$

where $\mathcal{C}_k := \sqrt{L_R}(\|u\| + \|v\|) \le 2\sqrt{L_R}(\delta^i L_\beta + \left\|\mathbf{B}((k+1)\delta^i) - \mathbf{B}(k\delta^i)\right\|_2)$.

Some of the terms above can be simplified. We begin by bounding the $\|u(0) - v(0)\|$ term. By assumption that $\beta$ is Lipschitz, $\left\|\delta^i \beta(x_k^i) - \Gamma_{x_{2k}^{i+1}}^{x_k^i} \delta^i \beta(x_{2k}^{i+1})\right\| \le \delta^i L'_\beta \mathrm{d}(x_k^i, x_{2k}^i)$. By definition of

$E_{2k}^{i+1}$ from (4.1),

$$\Gamma_{x_{2k}^{i+1}}^{x_k^i}((\mathbf{B}((2k+2)\delta^{i+1}) - \mathbf{B}(2k\delta^{i+1})) \circ E_{2k}^{i+1})$$

$$=(\mathbf{B}((2k+2)\delta^{i+1}) - \mathbf{B}(2k\delta^{i+1})) \circ (\Gamma_{x_{2k}^{i+1}}^{x_k^i} E_{2k}^{i+1})$$

$$=(\mathbf{B}((k+1)\delta^i) - \mathbf{B}(k\delta^i)) \circ E_k^i$$

where the last line is because $\delta^i = 2\delta^{i+1}$ and because $E_{2k}^{i+1} := \Gamma_{x_k^i}^{x_{2k}^{i+1}}(E_k^i)$ from (4.1).

Thus

$$(\mathbf{B}((k+1)\delta^i) - \mathbf{B}(k\delta^i)) \circ E_k^i - \Gamma_{x_{2k}^{i+1}}^{x_k^i}((\mathbf{B}((2k+2)\delta^{i+1}) - \mathbf{B}(2k\delta^{i+1})) \circ E_{2k}^{i+1})$$

$$=(\mathbf{B}((k+1)\delta^i) - \mathbf{B}(k\delta^i)) \circ E_k^i - (\mathbf{B}((k+1)\delta^i) - \mathbf{B}(k\delta^i)) \circ \Gamma_{x_{2k}^{i+1}}^{x_k^i} E_k^i$$

$$=0$$

We can thus bound via Young's Inequality:

$$\|u(0) - v(0)\|^2 \leq 2\delta^{i^2} L_\beta'^2 \mathrm{d}(x_k^i, x_{2k}^{i+1})^2$$

Finally, noting that $\|\gamma'(0)\| = i\mathrm{d}(x_k^i, x_{2k}^{i+1})$,

$$2\langle \gamma'(0), v(0) - u(0) \rangle = 2\left\langle \gamma'(0), \Gamma_{x_{2k}^{i+1}}^{x_k^i} \delta^i \beta(x_{2k}^{i+1}) - \delta^i \beta(x_k^i) \right\rangle \leq 2\delta^i L_\beta' \mathrm{d}(x_k^i, x_{2k}^{i+1})^2$$

Plugging into A.8

$$\mathrm{d}(x_{k+1}^i, \mathrm{Exp}_{x_{2k}^{i+1}}(a_k + b_k))^2 \leq (1 + 4\mathcal{C}_k^2 e^{4\mathcal{C}_k} + 64 e^{\mathcal{C}_k} \delta^i L_\beta') \mathrm{d}(x_k^i, x_{2k}^{i+1})^2$$

We now bound the second term of (A.7). Let us introduce two more convenient definitions:

$$b_k' := \delta^i \beta(x_{2k+1}^{i+1}) + (\mathbf{B}((2k+2)\delta^{i+1}) - \mathbf{B}((2k+1)\delta^{i+1})) \circ E_{2k+1}^{i+1}$$

$$z := \mathrm{Exp}_{x_{2k}^{i+1}}(a_k)$$

It follows from definition that

$$x_{2k+2}^{i+1} = \mathrm{Exp}_z(b_k')$$

We break the bound on $\mathrm{d}(\mathrm{Exp}_{x_{2k}^{i+1}}(a_k + b_k), x_{2k+2}^{i+1})$ into two terms:

$$\mathrm{d}(\mathrm{Exp}_{x_{2k}^{i+1}}(a_k + b_k), x_{2k+2}^{i+1}) \leq \mathrm{d}(\mathrm{Exp}_{x_{2k}^{i+1}}(a_k + b_k), \mathrm{Exp}_z(\Gamma_{x_{2k}^{i+1}}^{x_{2k+1}^{i+1}} b_k)) + \mathrm{d}(\mathrm{Exp}_z(\Gamma_{x_{2k}^{i+1}}^{x_{2k+1}^{i+1}} b_k), x_{2k+2}^{i+1})$$

$$= \mathrm{d}(\mathrm{Exp}_{x_{2k}^{i+1}}(a_k + b_k), \mathrm{Exp}_z(\Gamma_{x_{2k}^{i+1}}^{x_{2k+1}^{i+1}} b_k)) + \mathrm{d}(\mathrm{Exp}_z(\Gamma_{x_{2k}^{i+1}}^{x_{2k+1}^{i+1}} b_k), \mathrm{Exp}_z(b_k'))$$

To bound the first term, we apply Lemma 38 (from [Sun et al., 2019]) with $x = x_{2k}^i$, $a = a_k$, $y = b_k$, to get

$$\mathrm{d}(\mathrm{Exp}_{x_{2k}^{i+1}}(a_k + b_k), \mathrm{Exp}_z(\Gamma_{x_{2k}^{i+1}}^{x_{2k+1}^{i+1}} b_k))$$

$$\leq L_R \|a_k\| \|b_k\| (\|a_k\| + \|b_k\|) e^{\sqrt{L_R}(\|a_k\| + \|b_k\|)}$$

To bound the second term, we apply Lemma 28 (with $x = y = z$), so that

$$\mathrm{d}(\mathrm{Exp}_z(\Gamma_{x_{2k}^{i+1}}^{x_{2k+1}^{i+1}} b_k), \mathrm{Exp}_z(b_k'))^2$$

$$\leq 32 e^{\mathcal{C}_k'} \left\| \Gamma_{x_{2k}^{i+1}}^{x_{2k+1}^{i+1}} b_k - b_k' \right\|^2$$

$$\leq 64 e^{\mathcal{C}_k'} \delta^{i+1^2} L_\beta'^2 \mathrm{d}(x_{2k}^{i+1}, x_{2k+1}^{i+1})^2$$

$$\leq 128 e^{\mathcal{C}_k'} \delta^{i+1^2} L_\beta'^2 \|a_k\|_2^2$$

where we define $\mathcal{C}'_k := \sqrt{L_R}(\|b_k\| + \|b'_k\|)$.

Plugging everything into (A.7),

$$
\begin{aligned}
\mathrm{d}(x^i_{k+1}, x^{i+1}_{2k+2})^2 \leq &(1 + \frac{1}{2K})(1 + 4\mathcal{C}^2_k e^{4\mathcal{C}_k} + 64e^{\mathcal{C}_k}\delta^i L'_\beta)\mathrm{d}(x^i_k, x^{i+1}_{2k})^2 \\
&+ 32KL^2_R(\|a_k\|^6 + \|b_k\|^6)e^{2\sqrt{L_R}(\|a_k\|+\|b_k\|)} + 256Ke^{\mathcal{C}'_k}\delta^{i+1^2}L'^2_\beta\|a_k\|^2
\end{aligned}
$$

In fact, if we consider any $t \in [k\delta^i, (k+1)\delta^i)$, and using the definition of $x^i(t)$ from (A.3) as the linear interpolation between $x^i_k$ and $x^i_{k+1}$, we can extend the bound to

$$
\sup_{t \in [k\delta^i, (k+1)\delta^i)} \mathrm{d}(x^i(t), x^{i+1}(t))^2
$$

$$
\begin{aligned}
\leq &(1 + \frac{1}{2K})(1 + 4\mathcal{C}^2_k e^{4\mathcal{C}_k} + 64e^{\mathcal{C}_k}\delta^i L'_\beta)\mathrm{d}(x^i_k, x^{i+1}_{2k})^2 \\
&+ 32KL^2_R(\|a_k\|^6 + \|b_k\|^6)e^{2\sqrt{L_R}(\|a_k\|+\|b_k\|)} + 256Ke^{\mathcal{C}'_k}\delta^{i+1^2}L'^2_\beta\|a_k\|^2 \qquad \text{(A.9)}
\end{aligned}
$$

Let us define

$$
r_0 = 0
$$

$$
\begin{aligned}
r^2_{k+1} := &(1 + \frac{1}{2K})(1 + 4\mathcal{C}^2_k e^{4\mathcal{C}_k} + 64e^{\mathcal{C}_k}\delta^i L'_\beta)r^2_k \\
&+ 32KL^2_R(\|a_k\|^6 + \|b_k\|^6)e^{2\sqrt{L_R}(\|a_k\|+\|b_k\|)} + 256Ke^{\mathcal{C}'_k}\delta^{i+1^2}L'^2_\beta\|a_k\|^2
\end{aligned}
$$

It follows from (A.9) that $r_k \geq \sup_{t \in [(k-1)\delta^i, k\delta^i)} \mathrm{d}(x^i(t), x^{i+1}(t))$ and that $r_{k+1} \geq r_k$ with probability 1, for all $k$, so that $\sup_{t \leq T} \mathrm{d}(x^i(t), x^{i+1}(t)) \leq r_K$. We will now bound $\mathbb{E}\left[r^2_K\right]$, and then apply Markov's Inequality. Let us define $\mathcal{F}_k$ to be the $\sigma$-field generated by $\mathbf{B}(t)$ for $t \in [0, k\delta^i)$. Then

$$
\begin{aligned}
\mathbb{E}_{\mathcal{F}_k}\left[r^2_{k+1}\right] \leq &\mathbb{E}_{\mathcal{F}_k}\left[(1 + \frac{1}{2K})(1 + 4\mathcal{C}^2_k e^{4\mathcal{C}_k} + 64e^{\mathcal{C}_k}\delta^i L'_\beta)\right]r^2_k \\
&+ \mathbb{E}_{\mathcal{F}_k}\left[32KL^2_R(\|a_k\|^6 + \|b_k\|^6)e^{2\sqrt{L_R}(\|a_k\|+\|b_k\|)} + 256Ke^{\mathcal{C}'_k}\delta^{i+1^2}L'^2_\beta\|a_k\|^2\right]
\end{aligned}
$$

We will bound the terms above one by one. First, note from definition that $\mathcal{C}_k \leq \sqrt{L_R}(2\delta^i L_\beta + 2\|\mathbf{B}((k+1)\delta^i) - \mathbf{B}(k\delta^i)\|_2)$. Let $\eta^i_k := \mathbf{B}((k+1)\delta^i) - \mathbf{B}(k\delta^i)$.

For sufficiently large $i$, $\delta^i \leq \sqrt{L_R}L_\beta/8$. Simplifying,

$$
\mathbb{E}_{\mathcal{F}_k}\left[(1 + \frac{1}{2K})(1 + 4\mathcal{C}^2_k e^{4\mathcal{C}_k} + 64e^{\mathcal{C}_k}\delta^i L'_\beta)\right]
$$

$$
\begin{aligned}
\leq &1 + \frac{1}{2K} + 16\delta^{i^2}L_R L^2_\beta + 16L_R\mathbb{E}\left[\|\eta^i_k\|^2\right] + 16\delta^{i^2}L_R L^2_\beta\mathbb{E}\left[e^{2\sqrt{L_R}\|\eta^i_k\|}\right] \\
&+ \frac{8L_R}{\delta^i d}\mathbb{E}\left[\|\eta^i_k\|^4\right] + 8L_R\delta^i d\mathbb{E}\left[e^{4\sqrt{L_R}\|\eta^i_k\|}\right] + 128\delta^i L'_\beta\mathbb{E}\left[e^{2\sqrt{L_R}\|\eta^i_k\|}\right] \\
\leq &1 + \frac{1}{2K} + 16L_R(\delta^{i^2}L^2_\beta + \mathbb{E}\left[\|\eta^i_k\|^2\right] + \frac{1}{\delta^i d}\mathbb{E}\left[\|\eta^i_k\|^4\right]) + 8\delta^i(\delta^i L_R L^2_\beta + L_R d + 16L'_\beta)\mathbb{E}\left[e^{4\sqrt{L_R}\|\eta^i_k\|}\right] \\
\leq &1 + \frac{1}{2K} + 40(\delta^{i^2}L_R L^2_\beta + 2\delta^i L_R d + \delta^i L'_\beta)
\end{aligned}
$$

where we use

$$
\mathbb{E}\left[\|\eta^i_k\|^2\right] = \delta^i d
$$

$$
\mathbb{E}\left[\|\eta^i_k\|^2\right] \leq 2\delta^{i^2}d^2
$$

$$
\mathbb{E}\left[e^{4\sqrt{L_R}\|\eta^i_k\|}\right] \leq 2\mathbb{E}\left[e^{8L_R\|\eta^i_k\|^2}\right] \leq 2e^{16L_R\delta^i d} \leq 4
$$

where we use Lemma 21, and the fact that $\delta^i \leq \frac{1}{32L_R d}$ for sufficiently large $i$.

Next, we bound $\mathbb{E}_{\mathcal{F}_k}\left[32KL_R^2(\|a_k\|^6 + \|b_k\|^6)e^{2\sqrt{L_R}(\|a_k\|+\|b_k\|)}\right]$. Note that $\|a_k\| \leq \frac{\delta^i}{2}L_\beta + \|\eta_{2k}^{i+1}\|$ and $\|b_k\| \leq \frac{\delta^i}{2}L_\beta + \|\eta_{2k+1}^{i+1}\|$. By similar argument as above,

$$\mathbb{E}_{\mathcal{F}_k}\left[32KL_R^2(\|a_k\|^6 + \|b_k\|^6)e^{2\sqrt{L_R}(\|a_k\|+\|b_k\|)}\right]$$
$$\leq 2048KL_R^2 e^{2\sqrt{L_R}\delta^i L_\beta}\mathbb{E}_{\mathcal{F}_k}\left[(2^{-5}\delta^{i6}L_\beta^6 + \|\eta_{2k}^{i+1}\|^6 + \|\eta_{2k}^{i+1}\|^6) \cdot e^{2\sqrt{L_R}(\|\eta_{2k}^{i+1}\|+\|\eta_{2k+1}^{i+1}\|)}\right]$$
$$\leq KL_R^2(512\delta^{i6}L_\beta^6 + 2048\delta^{i3}d^3)$$

Finally, note that $\mathcal{C}_k' \leq 2\sqrt{L_R}(L_\beta + \|\eta_{2k+1}^{i+1}\|)$, so that

$$\mathbb{E}_{\mathcal{F}_k}\left[256Ke^{\mathcal{C}_k'}\delta^{i+1^2}L_\beta'^2\|a_k\|^2\right] \leq 256K\delta^{i2}L_\beta'^2(\delta^{i2}L_\beta^2 + \delta^i d)$$

Put together,

$$\mathbb{E}_{\mathcal{F}_k}\left[r_{k+1}^2\right]$$
$$\leq (1 + \frac{1}{2K} + 40(\delta^{i2}L_RL_\beta^2 + 2\delta^i L_R d + \delta^i L_\beta'))(KL_R^2(512\delta^{i6}L_\beta^6 + 2048\delta^{i3}d^3) + 256K\delta^{i2}L_\beta'^2(\delta^{i2}L_\beta^2 + \delta^i d))$$
$$\tag{A.10}$$

Applying the above recursively and simplifying,

$$\mathbb{E}\left[r_K^2\right] \leq e^{40K\delta^{i2}L_RL_\beta^2 + 2K\delta^i L_R d + K\delta^i L_\beta'}(K\delta^i)^2(\delta^{i4}L_R^2L_\beta^6 + \delta^i L_R^2 d^3 + \delta^{i2}L_\beta'^2L_\beta^2 + \delta^i L_\beta'^2 d) \cdot 2^{10}.$$
$$\tag{A.11}$$

Recall that $r_k \geq \sup_{t \in [(k-1)\delta^i, k\delta^i)} \mathrm{d}(x^i(t), x^{i+1}(t))$ (see (A.9)), and that $r_{k+1} \geq r_k$ with probability 1, for all $k$, so that $\sup_{t \leq T} \mathrm{d}(x^i(t), x^{i+1}(t)) \leq r_K$. This proves the first claim of the lemma.

By Markov's Inequality, and recalling that $r_k$ is w.p. 1 non-decreasing and $\sup_{t \leq T} \mathrm{d}(x^i(t), x^{i+1}(t)) \leq R_K$,

$$\mathbb{P}(\sup_{t \in [0,T]} \mathrm{d}(x^i(t), x^{i+1}(t))^2 \geq 2^{-i/2-4})$$
$$\leq \mathbb{E}\left[r_K^2\right] \cdot 2^{i/2+4}$$
$$\leq e^{(40\delta^i L_RL_\beta^2 + 2L_R d + L_\beta')T}T^2 \cdot (\delta^{i3}L_R^2L_\beta^6 + L_R^2 d^3 + \delta^i L_\beta'^2 L_\beta^2 + L_\beta'^2 d) \cdot 2^{-i/2+14} \tag{A.12}$$

This proves the second claim of the lemma. $\qquad\square$

**Lemma 5.** *Let $x \in M$ be some initial point and $E$ an orthonormal basis of $T_xM$. Let $\mathbf{B}(t)$ be a Brownian motion in $\mathbb{R}^d$; and $\beta(x)$ a vector field satisfying Assumption 3. Let $T \in \mathbb{R}^+$, for $t \in [0, T]$ and let $x^i(t)$ be constructed as per (4.2). Then with probability 1, there is a limit $x(t)$ such that for all $\varepsilon$, there exists an integer $N$ such that for all $i \geq N$,*

$$\sup_{t \in [0,T]} \mathrm{d}(x^i(t), x(t)) \leq \varepsilon.$$

*Proof of Lemma 5.* Let us define $L_0 := \|\beta(x(0))\|$.

**Step 1: Bounding the probability of deviation between $x^i$ and $x^{i+1}$**

We would like to apply Lemma 4. However, note that Lemma 4 assumes that $\|\beta(x)\| \leq L_\beta$ globally, which we do not assume here. We must therefore approximate $\beta$ by a sequence of Lipschitz vector fields.

Let us define

$$\beta^j(x) := \begin{cases} \beta(x) & \text{for } \|\beta(x)\| \leq 2^{j/2} \\ \beta(x) \cdot \frac{2^{j/2}}{\|\beta(x)\|} & \text{for } \|\beta(x)\| > 2^{j/2} \end{cases}$$

Let us denote by $L_{\beta^j} := 2^{j/2}$. We verify that for all $x, y \in M$, $\left\|\beta^j(x)\right\| \leq L_{\beta^j}$ and $\left\|\beta^j(x) - \Gamma_y^x \beta^j(y)\right\| \leq L'_\beta \|x - y\|$.

Finally, for any let $\tilde{x}^{i,j}(t)$ be as defined in (A.3), with $\beta$ replaced by $\beta^j$. Lemma 4 immediately implies that, for all $i \geq \mathcal{C}$ (where $\mathcal{C}$ is some constant depending on $L_R, T, d$),

$$\mathbb{P}(\sup_{t \in [0,T]} \mathrm{d}(\tilde{x}^{i,i}(t), \tilde{x}^{i+1,i}(t)) \geq 2^{-\frac{i}{4}-2})$$
$$\leq e^{(40TL_R + 2L_R d + L'_\beta)T} T^2 \cdot (T^3 L_R^2 + L_R^2 d^3 + T L'^2_\beta + L'^2_\beta d) \cdot 2^{-i/2+14}$$

where we use the fact that $\delta^i L_\beta^{i\,2} = T$ by definition.

Recalling that $\beta^j(x) = \beta(x)$ unless $\|\beta(x)\| \geq 2^{j/2}$,

$$\mathbb{P}(\exists_{t \in [0,T]} x^i(t) \neq \tilde{x}^{i,i}(t)) = \mathbb{P}(\exists_{k \in \{0\ldots2^i\}} x_k^i \neq \tilde{x}_k^{i,i}) \leq \mathbb{P}(\sup_{k \in \{0\ldots2^i\}} \|\beta(x_k^i)\| \geq 2^{i/2})$$

We can bound $\|\beta(x)\| \leq L_0 + L'_\beta \mathrm{d}(x, x_0)$, so that

$$\mathbb{P}(\sup_{k \in \{0\ldots2^i\}} \|\beta(x_k^i)\| \geq 2^{i/2})$$
$$\leq \mathbb{P}(\sup_{k \in \{0\ldots2^i\}} \mathrm{d}(x_k^i) \geq \frac{2^{i/2} - L_0}{L'_\beta})$$
$$\leq \exp(2 + 8TL'_\beta + TL_R d + TL_R L_0^2) \cdot (2Td + 4T^2 L_0^2) \cdot L'^2_\beta \cdot 2^{-i+2}$$

where we use Lemma 15, with $K = 2^i$, and assume that $i$ satisfies $2^{i/2} \geq L_0$ and $2^i \geq T$.

Using identical steps, we can also bound

$$\mathbb{P}(\exists_{t \in [0,T]} x^{i+1}(t) \neq \tilde{x}^{i+1,i}(t)) \leq \exp(2 + 8TL'_\beta + TL_R d + TL_R L_0^2) \cdot (2Td + 4T^2 L_0^2) \cdot L'^2_\beta \cdot 2^{-i+2}$$

Put together,

$$\mathbb{P}(\sup_{t \in [0,T]} \mathrm{d}(x^i(t), x^{i+1}(t)) \geq 2^{-\frac{i}{4}-2})$$
$$\leq \mathbb{P}(\sup_{t \in [0,T]} \mathrm{d}(\tilde{x}^{i,i}(t), \tilde{x}^{i+1,i}(t)) \geq 2^{-\frac{i}{4}-2}) + \mathbb{P}(\exists_{t \in [0,T]} x^i(t) \neq \tilde{x}^{i,i}(t)) + \mathbb{P}(\exists_{t \in [0,T]} x^{i+1}(t) \neq \tilde{x}^{i+1,i}(t))$$
$$\leq \mathcal{C}_2 \cdot 2^{-i/2}$$

where $\mathcal{C}_2$ is a constant that depends on $T, L_R, L'_\beta, L_0, d$, but does not depend on $i$.

**Step 2: Apply Borel-Cantelli to show uniformly-Cauchy sequence with probability 1**
Thus

$$\sum_{i=\mathcal{C}_1}^{\infty} \mathbb{P}(\sup_t \mathrm{d}(x^i(t), x^{i+1}(t)) \geq 2^{-\frac{i}{4}}) < \infty$$

By the Borel-Cantelli Lemma,

$$\mathbb{P}(\sup_t \mathrm{d}(x^i(t), x^{i+1}(t)) \geq 2^{-\frac{i}{4}} \text{ for infinitely many } i) = 0$$

Equivalently, with probability 1, for all $\varepsilon$, there exists a $N$ such that for all $i \geq N$, $\sup_t \mathrm{d}(x^i(t), x^{i+1}(t)) \leq 2^{-\frac{i}{4}}$. For any $j \geq i \geq N$, it then follows that

$$\sup_t \mathrm{d}(x^i(t), x^j(t)) \leq \sum_{\ell=i}^{j} \mathrm{d}(x^\ell(t), x^{\ell+1}(t))$$
$$\leq \sum_{\ell=i}^{j} 2^{-\frac{\ell}{4}}$$
$$\leq 6 \cdot 2^{-i/4}$$

**Step 3: Uniform-Cauchy sequence implies uniform convergence to limit using standard arguments** Therefore, with probability 1, $x^i(t)$ is a uniformly Cauchy sequence. Let $x(t)$ be the point-wise limit of $x^i(t)$, as $i \to \infty$. It follows [4] that with probability 1, for any $\varepsilon$, there exists a $N$ such that for all $i \geq N$,

$$\sup_{t \in [0,T]} \mathrm{d}(x^i(t), x(t)) \leq \varepsilon$$

$\square$

**Lemma 6.** *Let $\beta(\cdot)$ be a vector field satisfying Assumption 3. Assume also that there exists $L_\beta$ such that $\|\beta(x)\| \leq L_\beta$ for all $x$. Consider arbitrary $x_0 \in M$ and let $E$ be an orthonormal basis of $T_{x_0}M$. Let $\mathbf{B}$ be a standard Brownian motion in $\mathbb{R}^d$. Let $x^i(t) = \overline{\Phi}(t; x, E, \beta, \mathbf{B}, i)$ and $x(t) = \Phi(t; x, E, \beta, \mathbf{B})$ as defined in (A.3) and (A.4) respectively. (Existence of $x(t)$ follows from Lemma 5).*

*Then for any non-negative integer $\ell$,*

$$\mathbb{E}\left[\sup_{t \in [0,T]} \mathrm{d}(x^\ell(t), x(t))^2\right] \leq 2^{14} e^{40 T \delta^\ell L_R L_\beta^2 + 2 T L_R d + T L_\beta'} T^3 (\delta^{\ell^3} L_R^2 L_\beta^6 + L_R^2 d^3 + \delta^\ell {L_\beta'}^2 L_\beta^2 + {L_\beta'}^2 d) \cdot 2^{-\ell}$$

*where $\delta^i := 2^{-i}T$*

*Proof.* Consider any fixed $i$, let $\delta^i := T/2^i$ and let $K := T/\delta^i = 2^i$ as in (4.1).

By the first claim of Lemma 4, we can bound

$$\mathbb{E}\left[\sup_{t \in [0,T]} \mathrm{d}(x^i(t), x^{i+1}(t))^2\right]$$

$$\leq 2^{10} \cdot e^{40 K {\delta^i}^2 L_R L_\beta^2 + 2 K \delta^i L_R d + K \delta^i L_\beta'} (K \delta^i)^2 ({\delta^i}^4 L_R^2 L_\beta^6 + \delta^i L_R^2 d^3 + {\delta^i}^2 {L_\beta'}^2 L_\beta^2 + \delta^i {L_\beta'}^2 d)$$

$$= \underbrace{2^{10} e^{40 T \delta^i L_R L_\beta^2 + 2 T L_R d + T L_\beta'} T^3 ({\delta^i}^3 L_R^2 L_\beta^6 + L_R^2 d^3 + \delta^i {L_\beta'}^2 L_\beta^2 + {L_\beta'}^2 d)}_{:=s_i} \cdot 2^{-i}$$

where we use the fact that $K\delta^i = T$ by definition.

By repeated application of Young's Inequality, we can bound, for any $\ell$ and any $j \geq \ell$,

$$\mathbb{E}\left[\sup_{t \in [0,T]} \mathrm{d}(x^\ell(t), x^j(t))^2\right] \leq \sum_{i=\ell}^{j-1} 3\left(\frac{3}{2}\right)^{i-\ell} \mathbb{E}\left[\sup_{t \in [0,T]} \mathrm{d}(x^i(t), x^{i+1}(t))^2\right]$$

$$\leq \sum_{i=\ell}^{j-1} 3\left(\frac{3}{2}\right)^{i-\ell} \cdot 2^{-i} s_i$$

$$\leq 12 \cdot 2^{-\ell} \cdot s_\ell$$

Since the above holds for any $j$, we can take the limit of $j \to \infty$ and

$$\mathbb{E}\left[\sup_{t \in [0,T]} \mathrm{d}(x^\ell(t), x(t))^2\right] \leq 12 \cdot 2^{-\ell} \cdot s_\ell$$

where we use the fact that $\mathrm{d}(x_{2^j}^j, x(T))$ converges almost surely to 0, from Lemma 5. $\square$

### A.3 Discretization Error of Euler Murayama

Given the results of the previous section, we are now ready to prove Lemma 7, which is informally stated as Lemma 1 in the Section 4. The proof of Lemma 7 works by summing, for all $i$, the distance

---

[4]A nice clean proof can be seen at https://math.stackexchange.com/questions/1287669/uniformly-cauchy-sequences

between $x^i(t)$ and $x^{i+1}(t)$ (which is bounded in Lemma 4). Extra care must be taken to ensure that iterates do not stray too far from the initial error.

The crucial analysis corresponding to Step 1 (synchronous coupling) and Step 2 (rolling without slipping) discussed in the proof sketch of Lemma 1 in Section 4 can be found in the proof of Lemma 4 in Section A.2 above.

**Lemma 7.** *Let $M$ satisfy Assumption 4. Let $\beta(\cdot)$ be a vector field satisfying Assumption 3. Consider arbitrary $x(0) \in M$. Let $L_1$ be any constant such that $L_1 \geq \|\beta(x(0))\|$ and let $T$ be a step-size satisfying $T \leq \min\left\{\frac{1}{16L'_\beta}, \frac{1}{16L_R d}, \frac{1}{16\sqrt{L_R}L_1}\right\}$.*

*Let $x(t)$ denote the solution to (1.2), initialized at $x(0)$. Let $x^0(t) := \mathrm{Exp}_{x(0)}\left(t\beta(x(0)) + \sqrt{t}\zeta\right)$, where $\zeta \sim \mathcal{N}_{x(0)}(0, I)$. Then there exists a coupling between $x^0(T)$ and $x(T)$ such that*

$$\mathbb{E}\left[\mathrm{d}(x^0(T), x(T))^2\right] \leq 2^{20}(T^4 L_1^4 (1 + L_R) + T^4 L'^4_\beta + T^3(d^3(L_R + L'^2_\beta/L_1^2) + L'^2_\beta d))$$

*Proof of Lemma 7.* Let $E$ be an orthonormal basis of $T_{x(0)}M$. Following the definition of $\overline{\Phi}$ in (A.3), we verify that $x^0(T) = \overline{\Phi}(T; x(0), E, \beta, \mathbf{B}, 0)$, where $\mathbf{B}$ is some Brownian motion, and equality is in the sense of distribution. On the other hand, by Lemma 2, $x(t) = \Phi(t; x(0), E, \beta, \mathbf{B})$, where $\Phi$ is the limit of $\overline{\Phi}$ as defined in (A.4).

**Step 1: Bounding the distance between $x^i(T)$ and $x^{i+1}(T)$**
Let us consider some fixed $i$. Let $\delta^i = T/2^i$ denote the stepsize, and let $K := 2^i$ so that $T = K\delta^i$. Let $x^i(t) = \overline{\Phi}(T; x(0), E, \beta, \mathbf{B}, i)$ be as defined in (A.3). Recall that $x^i(t)$ is by definition the linear interpolation of $x^i_k$ (defined in (4.1) or equivalently (A.2)), which are marginally Euler-Murayama sequences with stepsize $\delta^i$.

Our goal is to bound $\mathbb{E}\left[\mathrm{d}(x^i(T), x^{i+1}(T))^2\right]$. Lemma 4 almost gives us what we want; the problem is that Lemma 4 assumes that for all $x \in M$, $\|\beta(x)\| \leq L_\beta$, but we do not make that assumption in this lemma. In order to get around this issue, **we use an argument based on truncating $\beta$ at larger and larger norms**.

Let us define $L_0 := \|\beta(x(0))\|$ and

$$\beta^j(y) := \begin{cases} \beta(y) & \text{for } \|\beta(y)\| \leq L_1 2^{j/2+1} \\ \beta(y) \cdot \frac{L_1 2^{j/2+1}}{\|\beta(y)\|} & \text{for } \|\beta(y)\| > L_1 2^{j/2+1} \end{cases}, \tag{A.13}$$

i.e. $\beta^j$ is the truncated version of $\beta$, so that the norm of $\beta^j$ is globally upper bounded by $L_1 2^{j/2+1}$.

Given this definition of $\beta^j$, we now define, for all $j \in \mathbb{Z}^+$, $\tilde{x}^{i,j}(t) := \overline{\Phi}(t; x(0), E, \beta^j, \mathbf{B}, i)$, which is the (interpolated) Euler-Murayama discretization from (A.3), with step-size $\delta^i$, and drift $\beta^j$. In other words, $x^i(t)$ and $\tilde{x}^{i,j}(t)$ are both Euler Murayama discretizations with stepsize $\delta^i$, but the former has drift $\beta$ whereas the latter has drift $\beta^j$. We also let $\tilde{x}^{\cdot,j}(t) := \Phi(t; x(0), E, \beta^i, \mathbf{B})$ denote the limit, as $i \to \infty$, of $\tilde{x}^{i,j}(t)$ (see definition in (A.4)).

By Young's Inequality,

$$\mathrm{d}(x^i(t), x^{i+1}(t))^2 \leq 8\mathrm{d}(x^i(t), \tilde{x}^{i,i}(t))^2 + 8\mathrm{d}(x^{i+1}(t), \tilde{x}^{i+1,i}(t))^2 + 8\mathrm{d}(\tilde{x}^{i,i}(t), \tilde{x}^{i+1,i}(t))^2 \tag{A.14}$$

Before proceeding, we briefly explain the intuition behind the decomposition in (A.14). Notice that we let $j = i$ in $\tilde{x}^{i,i}$, i.e. as $i$ increases, two this happen: $\delta^i$ becomes smaller, and $\beta^i$ becomes truncated at a larger norm (see (A.13)), and is thus closer to the true un-truncated $\beta$.

The first and second term on the right hand side of (A.14) correspond to error due to truncating $\beta$ to $\beta^i$. These two terms would equal 0 if $\sup_t \mathrm{d}(x^i(t), \tilde{x}^{i,i}(t)) \leq \frac{L_1 2^{i/2}}{L'_\beta}$, as that implies (via Assumption 3) that $\sup_t \left\|\beta(x^i(t))\right\| \leq L_1 2^{i/2+1}$ (so along the entire path, $\beta$ was never large enough to require truncation). As $i$ increases, $\frac{L_1 2^{i/2}}{L'_\beta} \to \infty$, and these two truncation errors are 0 with increasing probability.

The last term in (A.14) corresponds to error due to different discretization stepsize ($\delta^i$ vs $\delta^{i+1}$).

**Step 1.1: Bounding distance of truncated-drift sequences $\tilde{x}^{i,i}$ and $\tilde{x}^{i+1,i}$**

We first bound the last term of (A.14). By the first claim of Lemma 4, we can bound

$$
\mathbb{E}\left[\mathrm{d}(\tilde{x}^{i,i}(T), \tilde{x}^{i+1,i}(T))^2\right]
$$

$$
\leq 2^{10} e^{40 T^2 L_R L_1^2 + 2T L_R d + T L_\beta'} T^3 (T^3 L_R^2 L_1^6 + L_R^2 d^3 + T L_\beta'^2 L_1^2 + L_\beta'^2 d) \cdot 2^{-i}
$$

$$
\leq 2^{14-i} T^3 (T^3 L_R^2 L_1^6 + L_R^2 d^3 + T L_\beta'^2 L_1^2 + L_\beta'^2 d),
$$

where the first inequality uses the fact that $\delta^i = T/2^i \leq T$ and that $\delta^i L_1 2^{i+1} = 2T L_1$, and second inequality is by our assumed upper bound on $T$. Here, we crucially use the fact that, the effect of $\delta^i$ halving with $i$ "cancels out" the effect of $\left\|\beta^i(x)\right\|$ becoming larger with $i$.

**Step 1.2: Bounding error due to truncation**

We now bound the first two terms of (A.14). Once again recall from the definition in (4.2) and (A.3) that $x^i(t)$ (resp $\tilde{x}^{i,i}(t)$) are linear interpolations of the discrete sequence $x_k^i$ (resp $\tilde{x}_k^{i,i}$) as defined in (4.1). Under the event $\sup_{k \in \{0...2^i\}} \mathrm{d}(x_k^i, x(0)) \leq \frac{2^{i/2} L_1}{L_\beta'}$, we verify that $x_k^i = \tilde{x}_k^{i,i}$ for all $k \in \{0...2^i\}$ – this is because we can then bound, for all $k$, $\left\|\beta(x_k^i)\right\| \leq L_\beta' \mathrm{d}(x_k^i, x(0)) + L_0 \leq 2^{i/2+1} L_1$, which in turn implies that for all $k$, $\beta(x_k^i)$ equals the truncated version $\beta^i(x_k^i)$, which in turn implies that $\mathrm{d}(x^i(T), \tilde{x}^{i,i}(T)) = 0$. Therefore,

$$
\mathbb{E}\left[\mathrm{d}(x^i(T), \tilde{x}^{i,i}(T))^2\right]
$$

$$
= \mathbb{E}\left[\mathbb{1}\left\{\sup_{k \in \{0..2^i\}} \mathrm{d}(x_k^i, x(0)) > \frac{2^{i/2} L_1}{L_\beta'}\right\} \mathrm{d}(x^i(T), \tilde{x}^{i,i}(T))^2\right]
$$

$$
\leq 2\sqrt{\mathbb{P}\left(\sup_{k \in \{0..2^i\}} \mathrm{d}(x_k^i, x(0)) > \frac{2^{i/2} L_1}{L_\beta'}\right)} \cdot \left(\sqrt{\mathbb{E}\left[\mathrm{d}(x^i(T), x(0))^4\right]} + \sqrt{\mathbb{E}\left[\mathrm{d}(\tilde{x}^{i,i}(T), x(0))^4\right]}\right).
$$

$$
\tag{A.15}
$$

where the second line follows from Young's inequality and Cauchy Schwarz.

From Lemma 16, and our assumed bound on $T$,

$$
\sqrt{\mathbb{P}\left(\sup_{k \in \{0...2^i\}} \mathrm{d}(x_k^i, x(0)) > \frac{2^{i/2} L_1}{L_\beta'}\right)} \leq \frac{L_\beta'^2}{L_1^2 2^i} \exp\left(1 + 8T L_\beta' + 2T L_R d + 2T \delta^i L_R L_0^2\right)(3Td + 8T^2 L_0^2)
$$

$$
\leq \frac{L_\beta'^2 (Td + T^2 L_0^2)}{L_1^2} \cdot 2^{4-i}.
$$

Also from Lemma 16, for any $k$,

$$
\mathbb{E}\left[\mathrm{d}(x_k, x(0))^4\right] \leq \exp\left(2 + 16T L_\beta' + 4T L_R d + 3T^2 L_R L_0^2\right)(T^2 d^2 + 64T^4 L_0^4)
$$

$$
\leq 4(T^2 d^2 + 64T^4 L_0^4).
$$

The same upper bound also applies to $\mathbb{E}\left[\mathrm{d}(\tilde{x}^{i,i}(T), x(0))^4\right]$. Plugging into (A.15),

$$
\mathbb{E}\left[\mathrm{d}(x^i(T), \tilde{x}^{i,i}(T))^2\right] \leq 2^{12-i} \frac{L_\beta'^2 (Td + T^2 L_0^2)^3}{L_1^2}.
$$

Note that the bound in Lemma 16 is strictly stronger for $x_k^{i+1}$ compared to $x_k^i$. Thus by exactly identical steps, we can also upper bound

$$
\mathbb{E}\left[\mathrm{d}(x^{i+1}(T), \tilde{x}^{i+1,i}(T))^2\right] \leq 2^{12-i} \frac{L_\beta'^2 (Td + T^2 L_0^2)^3}{L_1^2}.
$$

Plugging into (A.14), and applying Young's Inequality, we finally have our bound between the Euler-Murayama sequences $x^i(T)$ and $x^{i+1}(T)$ (without truncation of $\beta$).

$$
\begin{aligned}
\mathbb{E}\left[\mathrm{d}(x^i(T), x^{i+1}(T))^2\right] &\leq 2^{14-i}T^3(T^3L_R^2L_1^6 + L_R^2d^3 + TL_\beta'^2L_1^2 + L_\beta'^2d) + 2^{13-i}\frac{L_\beta'^2(Td + T^2L_0^2)^3}{L_1^2} \\
&\leq 2^{16-i}(T^4L_1^4(1 + L_R) + T^4L_\beta'^4 + T^3(d^3(L_R + L_\beta'^2/L_1^2) + L_\beta'^2d)),
\end{aligned}
\tag{A.16}
$$

where we use the assumed upper bound on $T$ in the lemma statement.

**Step 2: Summing over $i$**
For any $\ell \in \mathbb{Z}^+$, we can summing (A.16) for $i \in \{0, 1, 2...\ell\}$ to bound

$$
\begin{aligned}
\mathbb{E}\left[\mathrm{d}(x^0(T), x^\ell(T))^2\right] &\leq \sum_{i=0}^\ell 3 \cdot \left(\frac{3}{2}\right)^i 2^{16-i}(T^4L_1^4(1 + L_R) + T^4L_\beta'^4 + T^3(d^3(L_R + L_\beta'^2/L_1^2) + L_\beta'^2d)) \\
&\leq 2^{20}(T^4L_1^4(1 + L_R) + T^4L_\beta'^4 + T^3(d^3(L_R + L_\beta'^2/L_1^2) + L_\beta'^2d)).
\end{aligned}
\tag{A.17}
$$

The first inequality uses triangle inequality and Young's inequality recursively: for any $i$,

$$
\mathrm{d}(x^i(T), x^\ell(T))^2 \leq \frac{3}{2}\mathrm{d}(x^{i+1}(T), x^\ell(T))^2 + 3\mathrm{d}(x^i(T), x^{i+1}(T))^2.
$$

Since (A.17) holds for all $\ell$, we take the limit of $\ell \to \infty$. By dominated convergence together with Lemma 5,

$$
\mathbb{E}\left[\mathrm{d}(x^0(T), x(T))^2\right] \leq 2^{20}(T^4L_1^4(1 + L_R) + T^4L_\beta'^4 + T^3(d^3(L_R + L_\beta'^2/L_1^2) + L_\beta'^2d)).
$$

$\square$

## A.4 Proof of Theorem 1

Below, we provide the full proof of Theorem 1, which was sketched in Section 5. The main results used are Lemma 3 (contraction of Lyapunov function under exact SDE) and Lemma 7 (bound on Euler Murayama discretization error).

*Proof of Theorem 1.*
**Step 0: Defining some Key Constants**
In this step, we define a radius $r$, an event $A_k$ based on $r$, and an upper bound on $\delta$.

$$
\begin{aligned}
c_0 &:= \log\left(\frac{L_\beta'}{m}\right) + \log\left(\frac{(1 + L_R)d}{m}\right) + \log\mathcal{R} + \log(K), \\
r_0 &= 32\sqrt{\frac{L_\beta'^2\mathcal{R}^2}{m} + \frac{L_Rd^2}{m^2} + \frac{d}{m} \cdot c_0}, \\
r &= r_0 + 32\sqrt{\frac{d}{m} \cdot \log(1/\delta)}.
\end{aligned}
\tag{A.18}
$$

Let $A_k$ denote the event $\max_{i \leq k} \mathrm{d}(x_i, x^*) \leq r$. The value of $r$ and $A_k$ are chosen so that (A.23) holds in Step 1 below. Note that $r$ depends on $\log(1/\delta)$ on the last line.

We now define a suitable upperbound on the stepsize $\delta$. To do so, we first define the constants $c_1, c_5, c_6, c_7$:

$$
c_1 := \min\left\{\frac{1}{16L_\beta'}, \frac{1}{16L_Rd}, \frac{m}{64d\sqrt{L_R}L_\beta'r_0}, \frac{m}{64d\sqrt{L_R}L_\beta'\sqrt{\log(d\sqrt{L_R}L_\beta'/m)}}\right\},
\tag{A.19}
$$

$$c_5 := \frac{1}{64} \min \left\{ \frac{m}{L_\beta'^2}, \frac{d}{m} \right\},$$

$$c_6 := \frac{1}{64} \min \left\{ \frac{m}{L_\beta'^2 \sqrt{L_R} r_0}, \frac{d}{m\sqrt{L_R} r_0}, \frac{d^2}{m^2 r_0^2} \right\},$$

$$c_7 := \frac{1}{64} \min \left\{ \sqrt{\frac{m^3}{dL_\beta'^4 L_R^2 \log\left(dL_\beta'^4 L_R^2/m^3\right)}}, \sqrt{\frac{d}{mL_R \log\left(mL_R/d\right)}}, \frac{d}{m \log\left(d/m\right)} \right\}. \quad \text{(A.20)}$$

For our proof, we require that $\delta$ satisfies

$$\delta \leq \min \{c_1, c_5, c_6, c_7\} \quad \text{(A.21)}$$

Thus $\mathcal{C}_0$ from the theorem statement is explicitly $\mathcal{C}_0 = \frac{1}{\min\{c_1, c_5, c_6, c_7\}}$. The motivation for this upper bound on $\delta$ is so that $\delta$ satisfies the conditions in Lemma 7 and Lemma 20 which are used in Step 1 and Step 3. We provide details in Step 4 below. Note that the upper bound on $\delta$ depends only on $r_0$ but not $r$, so the definition is not circular.

**Step 1: Tail Bound:**
We now show that with high probability, the discretization sequence $x_k$ never steps outside the ball of radius $r$ centered at $x^*$ (this is exactly the event $A_k$ that we defined in the previous step).

By Lemma 20 with $\sigma = 0$ and $\tilde{\beta} = \beta$, for any $(\delta, r)$ satisfying
$\delta \leq \min \left\{ \frac{m}{16L_\beta'^2(1+\sqrt{L_R}r)}, \frac{d}{m(1+\sqrt{L_R}r)}, \frac{32d^2}{m^2 r^2} \right\}$, we can bound the probability of $x_k$ stepping outside the ball of radius $r$ centered at $x^*$, for any $k \leq K$, as

$$\mathbb{P}(A_k^c) = \mathbb{P}(\max_{k\leq K} \mathrm{d}(x_k, x^*) \geq r) \leq 32K\delta m \exp\left( \frac{2L_\beta'^2 \mathcal{R}^2}{d} + \frac{64L_R d}{m} - \frac{mr^2}{256d} \right). \quad \text{(A.22)}$$

Furthermore, we can bound the fourth-moment of the distance between $x_K$ and $x^*$: by Lemma 17 with $\xi_k(x_k) = \zeta_k$ and $\sigma_\xi = 2\sqrt{d}$, we can bound

$$\mathbb{E}\left[ \mathrm{d}(x_K, x^*)^2 \right] \leq \frac{2^{13} L_R L_\beta'^4 d^2}{m^6} + \frac{16 L_\beta' \mathcal{R}^2}{m} + \frac{16d}{m}$$

Similarly, we can use Lemma 18 to bound

$$\mathbb{E}\left[ \mathrm{d}(y(K\delta), x^*)^2 \right] \leq \frac{2^{13} L_R L_\beta'^4 d^2}{m^6} + \frac{16 L_\beta' \mathcal{R}^2}{m} + \frac{32d}{m}.$$

Together, we can bound the expected distance between $y(K\delta)$ and $x_K$ under the low probability event $A_K^c$. By choosing $r$ to be sufficiently large in (A.18), we make (A.22) sufficiently small. We then apply triangle inequality to bound $\mathrm{d}(y(K\delta), x_K) \leq \mathrm{d}(y(K\delta), x^*) + \mathrm{d}(x_K, x^*)$ followed by Cauchy Schwarz, and verify that

$$\mathbb{E}\left[ \mathbb{1}\{A_K^c\} \mathrm{d}(y(K\delta), x_K) \right] \leq \sqrt{\delta} \quad \text{(A.23)}$$

**Step 2: Continuous Time Contraction**
Having established a bound in the distance under the event $A_K^c$, we now turn our attention to the high-probability event $A_K$.

Consider some fixed but arbitrary $k \leq K$. Let us define a continuous time SDE $\bar{x}^k(t)$, for $t \in [k\delta, (k+1)\delta]$, as the solution to the exact Langevin diffusion (1.2) initialized at $\bar{x}^k(k\delta) := x_k$.

The goal of this step is to bound $\mathbb{E}\left[ \mathbb{1}\{A_k\} \mathrm{d}(y((k+1)\delta), \bar{x}^k((k+1)\delta)) \right]$ in terms of $\mathbb{E}\left[ \mathbb{1}\{A_k\} \mathrm{d}(y(k\delta), \bar{x}^k(k\delta)) \right] := \mathbb{E}\left[ \mathbb{1}\{A_k\} \mathrm{d}(y(k\delta), x_k) \right]$. We apply Lemma 3, which guarantees that there exists a coupling between the two exact SDE processes $y$ and $\bar{x}^k$, such that $\mathbb{E}\left[ f(\mathrm{d}(y(t), \bar{x}^k(t))) \right]$ contracts with rate $\alpha$, i.e.

$$\mathbb{E}\left[ \mathbb{1}\{A_k\} f(\mathrm{d}(y((k+1)\delta), \bar{x}^k((k+1)\delta))) \right] \leq e^{-\alpha\delta} \mathbb{E}\left[ \mathbb{1}\{A_k\} f(\mathrm{d}(y(k\delta), x_k)) \right], \quad \text{(A.24)}$$

where $f$ is a Lyapunov function satisfying $f(r) \geq \frac{1}{2} \exp\left(-(q + L_{\text{Ric}}/2)\mathcal{R}^2/2\right)r$ and $|f'(r)| \leq 1$, and $\alpha := \min\left\{\frac{m - L_{Ric}/2}{16}, \frac{1}{2\mathcal{R}^2}\right\} \cdot \exp\left(-\frac{1}{2}(q + L_{\text{Ric}}/2)\mathcal{R}^2\right)$ are as defined in Lemma 3, and are consistent with the definition in our theorem statement.

**Step 3: Euler Murayama Error**
Next, we bound the distance between $x_{k+1}$ and $\bar{x}^k((k+1)\delta)$. This represents the discretization error between a single Euler-Murayama step with stepsize $\delta$, and the exact Langevin diffusion over $\delta$ time.

We will apply Lemma 7 with $L_1 = L'_\beta r$. We verify that under the event $A_k$ and by Assumption 3, $\|\beta(x_k)\|$ is indeed bounded by $L_1$. Thus by Lemma 7,

$$
\mathbb{E}\left[\mathbb{1}\{A_k\}\mathrm{d}(x_{k+1}, \bar{x}^k((k+1)\delta))\right]
$$

$$
\leq \sqrt{2^{20}(\delta^4 L_1^4(1 + L_R) + \delta^4 L'^4_\beta + \delta^3(d^3(L_R + L'^2_\beta/L_1^2) + L'^2_\beta d))}
$$

$$
\leq \sqrt{2^{20}(\delta^4 L'^4_\beta r^4(1 + L_R) + \delta^4 L'^4_\beta + \delta^3(d^3(L_R + 1/\mathcal{R}^2) + L'^2_\beta d))}
$$

$$
\leq \tilde{O}(\delta^{3/2}) \tag{A.25}
$$

where $\tilde{O}$ hides polynomial dependency on $L'_\beta, d, L_R, \mathcal{R}, \log K, \log(1/\delta)$.

Combining (A.24) and (A.25) and using triangle inequality and the fact that $|f'| \leq 1$, along with the fact that $A_{k+1} \subset A_k$, we can bound

$$
\mathbb{E}\left[\mathbb{1}\{A_{k+1}\}f(\mathrm{d}(y((k+1)\delta), x_{k+1}))\right] \leq e^{-\alpha\delta}\mathbb{E}\left[\mathbb{1}\{A_k\}f(\mathrm{d}(y(k\delta), x_k))\right] + \tilde{O}(\delta^{3/2}), \tag{A.26}
$$

where $\tilde{O}$ hides polynomial dependency on $L'_\beta, d, L_R, \mathcal{R}, \log K, \log(1/\delta)$. This shows that, in one $\delta$-time step, the Lyapunov function of the distance contracts with rate $\alpha$, plus a discretization error of order $\delta^{3/2}$. Applying (A.26) recursively, we can bound

$$
\mathbb{E}\left[\mathbb{1}\{A_K\}f(\mathrm{d}(y(K\delta), x_K))\right] \leq e^{-\alpha K\delta}\mathbb{E}\left[f(\mathrm{d}(y(0), x_0)^2)\right] + \frac{1}{\alpha} \cdot \tilde{O}(\delta^{1/2}). \tag{A.27}
$$

Combining (A.23) and (A.27), and using the fact that $f(r) \leq r$, gives

$$
\mathbb{E}\left[f(\mathrm{d}(y(K\delta), x_K))\right] \leq e^{-\alpha K\delta}\mathbb{E}\left[f(\mathrm{d}(y(0), x_0)^2)\right] + \frac{1}{\alpha} \cdot \tilde{O}(\delta).
$$

Using the fact that $\frac{1}{2}\exp\left(-(q + L_{\text{Ric}}/2)\mathcal{R}^2/2\right)r \leq f(r) \leq r$, we have

$$
\mathbb{E}\left[\mathrm{d}(y(K\delta), x_K)\right] \leq e^{-\alpha K\delta + (q + L_{Ric}/2)\mathcal{R}^2/2}\mathbb{E}\left[\mathrm{d}(y(0), x_0)\right] + \frac{\exp\left((q + L_{Ric}/2)\mathcal{R}^2/2\right)}{\alpha} \cdot \tilde{O}(\delta^{1/2})
$$

$$
= e^{-\alpha K\delta + (q + L_{Ric}/2)\mathcal{R}^2/2}\mathbb{E}\left[\mathrm{d}(y(0), x_0)\right] + \exp\left((q + L_{Ric}/2)\mathcal{R}^2\right) \cdot \tilde{O}(\delta^{1/2}),
$$

where $\tilde{O}$ hides polynomial dependency on $L'_\beta, d, L_R, \mathcal{R}, \frac{1}{m - L_{Ric}/2}, \log K, \log\frac{1}{\delta}$. This concludes the proof of Theorem 1.

**Step 4: Verifying Conditions on $\delta$**
In the proof above, we applied Lemma 7 and Lemma 20. Each of these requires certain bounds on $\delta$. In this step, we verify that the conditions on $\delta$ for each of these lemmas is satisfied by (A.21).

**Lemma 7 with $L_1 = L'_\beta r$ requires**

$$
\delta \leq \frac{1}{16}\min\left\{\frac{1}{L'_\beta}, \frac{1}{L_R d}, \frac{1}{\sqrt{L_R}L'_\beta r}\right\}
$$

We verify that this follows from (A.19). We specifically verify that $\delta \leq \frac{1}{16\sqrt{L_R}L'_\beta r}$ is satisfied due to the last two terms in (A.19), and Lemma 37.

**Lemma 20 requires** requires $\delta \leq \min\left\{\frac{m}{16L'^2_\beta(1 + \sqrt{L_R}r)}, \frac{d + \sigma^2}{m(1 + \sqrt{L_R}r)}, \frac{32(d^2 + \sigma^4)}{m^2 r^2}\right\}$. This is satisfied by (A.20) and Lemma 37.

$\square$

## A.5 Proof of Theorem 2

Below, we provide the full proof of Theorem 2, which was sketched in Section 6. The main results used are Lemma 8 (contraction of distance under Euler Murayama step) and Lemma 7 (bound on Euler Murayama discretization error).

*Proof of Theorem 2.*
**Step 0: Defining some Key Constants**
In this step, we define a radius $r$, an event $A_k$ based on $r$, and an upper bound on $\delta$.

$$c_0 := \log\left(\frac{L'_\beta}{m}\right) + \log\left(\frac{(1+L_R)(d+\sigma^2)}{m}\right) + \log \mathcal{R} + \log(K),$$

$$r_0 = 32\sqrt{\frac{L'^2_\beta \mathcal{R}^2}{m} + \frac{L_R(d^2+\sigma^4)}{m^2} + \frac{d}{m} \cdot c_0},$$

$$r = r_0 + 32\sqrt{\frac{d}{m} \cdot \log(1/\delta)}. \tag{A.28}$$

Let $A_k$ denote the event $\max_{i \le k} \max \{d(x_i, x^*), d(y(i\delta), x^*)\} \le r$. The value of $r$ and $A_k$ are chosen so that (A.36) holds in Step 1 below. Note that $r$ depends on $\log(1/\delta)$ on the last line.

We now define a suitable upperbound on the stepsize $\delta$. To do so, we first define the constants $c_1, c_2, c_3, c_4, c_5, c_6, c_7$:

$$c_1 := \min\left\{\frac{1}{16L'_\beta}, \frac{1}{16L_R d}, \frac{m}{64d\sqrt{L_R}L'_\beta r_0}, \frac{m}{64d\sqrt{L_R}L'_\beta\sqrt{\log(d\sqrt{L_R}L'_\beta/m)}}\right\}, \tag{A.29}$$

$$c_2 := \min\left\{\frac{m-L_{Ric}/2}{128L'^2_\beta}, \frac{m-L_{Ric}/2}{32L_R\sigma^2}, \frac{m-L_{Ric}/2}{2^{13}L_RL'_\beta d}, \frac{(m-L_{Ric}/2)^2}{2^{24}L'^2_R d^3}, \frac{m-L_{Ric}/2}{2^{17}d^2 L^2_R}\right\}, \tag{A.30}$$

$$c_3 := \min\left\{\frac{m-L_{Ric}/2}{32L_R L'^2_\beta r_0^2}, \sqrt{\frac{m-L_{Ric}/2}{2^{14}(L'^3_\beta r_0^3)L'_R}}\right\},$$

$$c_4 := \frac{1}{128}\min\left\{\frac{m(m-L_{Ric}/2)}{32dL_R L'^2_\beta \log\left(dL_R L'^2_\beta/m(m-L_{Ric}/2)\right)}, \frac{(m-L_{Ric}/2^{1/2})m^{3/4}}{L'^{1/2}_\beta L'^{1/2}_R d^{3/4}\log\left(L'^{1/2}_\beta L'^{1/2}_R d^{3/4}/((m-L_{Ric}/2)^{1/2}m^{3/4})\right)}\right\}, \tag{A.31}$$

$$c_5 := \frac{1}{64}\min\left\{\frac{m}{L'^2_\beta}, \frac{d}{m}\right\},$$

$$c_6 := \frac{1}{64}\min\left\{\frac{m}{L'^2_\beta\sqrt{L_R}r_0}, \frac{d}{m\sqrt{L_R}r_0}, \frac{d^2}{m^2 r_0^2}\right\},$$

$$c_7 := \frac{1}{64}\min\left\{\sqrt{\frac{m^3}{dL'^4_\beta L^2_R \log\left(dL'^4_\beta L^2_R/m^3\right)}}, \sqrt{\frac{d}{mL_R\log\left(mL_R/d\right)}}, \frac{d}{m\log\left(d/m\right)}\right\}. \tag{A.32}$$

For our proof, we require that $\delta$ satisfies

$$\delta \le \min\{c_1, c_2, c_3, c_4, c_5, c_6, c_7\}. \tag{A.33}$$

Thus $\mathcal{C}_1$ from the theorem statement is explicitly $\mathcal{C}_1 = \frac{1}{\min\{c_1,c_2,c_3,c_4,c_5,c_6\}}$. The motivation for this upper bound on $\delta$ is so that $\delta$ satisfies the conditions in Lemma 7, Lemma 8 and Lemma 20 which are used in Steps 1-3. We provide details in Step 4 below. Note that the upper bound on $\delta$ depends only on $r_0$ but not $r$, so the definition is not circular.

**Step 1: Tail Bound:**
We now show that with high probability, the discretization sequence $x_k$ and the exact SDE $y(t)$ never step outside the ball of radius $r$ centered at $x^*$ (this is exactly the event $A_k$ that we defined in the previous step).

By Lemma 20, for any $(\delta, r)$ satisfying $\delta \leq \min \left\{ \frac{m}{16 L_\beta'^2 (1 + \sqrt{L_R} r)}, \frac{d + \sigma^2}{m(1 + \sqrt{L_R} r)}, \frac{32(d^2 + \sigma^4)}{m^2 r^2} \right\}$, we can bound the probability of $x_k$ stepping outside the ball of radius $r$ centered at $x^*$, for any $k \leq K$, as

$$\mathbb{P}(\max_{k \leq K} d(x_k, x^*) \geq r) \leq 32 K \delta m \exp \left( \frac{2 L_\beta'^2 \mathcal{R}^2}{d + \sigma^2} + \frac{64 L_R (d + \sigma^2)}{m} - \frac{mr^2}{256(d + \sigma^2)} \right). \quad \text{(A.34)}$$

On the other hand, by Lemma 2, for $T = K\delta$, there is a family of discrete sequences, $y_k^i$, corresponding to (1.2) with stepsize $\delta^i = T/2^i$, whose linear interpolation $y^i(t)$ converges to $y(t)$ uniformly almost surely. By Lemma 20 with $\tilde{\beta} = \beta$, and $\sigma = 0$, and stepsize $\delta^i$, and iteration number $K^i := 2^i$ (so that $T = K^i \delta^i$), for $i$ sufficiently large, we can bound $\mathbb{P}(\max_{k \leq K^i} d(y_k^i, x^*) \geq r) \leq 32 T m \exp \left( \frac{2 L_\beta'^2 \mathcal{R}^2}{d} + \frac{64 L_R d}{m} - \frac{mr^2}{256 d} \right)$. Taking the limit of $i \to \infty$, we can bound

$$\mathbb{P}(\max_{k \leq K} d(y(k\delta), x^*) \geq r) \leq 32 T m \exp \left( \frac{2 L_\beta'^2 \mathcal{R}^2}{d} + \frac{32 L_R d}{m} - \frac{mr^2}{256 d} \right). \quad \text{(A.35)}$$

Combining (A.34) and (A.35), we can bound

$$\mathbb{E}\left[ \mathbb{1}\{A_k^c\} \right] = \mathbb{P}(A_k^c) \leq 64 K \delta m \exp \left( \frac{2 L_\beta'^2 \mathcal{R}^2}{d} + \frac{64 L_R (d + \sigma^2)}{m} - \frac{mr^2}{256 d} \right).$$

Furthermore, we can bound the fourth-moment of the distance between $x_K$ and $x^*$: by Lemma 17 with $\xi_k(x_k) = \sqrt{\delta}(\tilde{\beta}_k(x_k) - \beta(x_k)) + \zeta_k$ and $\sigma_\xi = 2(\sigma + \sqrt{d})$, we can bound

$$\mathbb{E}\left[ d(x_K, x^*)^4 \right] \leq \frac{2^{26} L_R^2 L_\beta'^8 (\sigma^8 + d^4)}{m^{12}} + \frac{128 L_\beta'^2 \mathcal{R}^4}{m^2} + \frac{2^{10}(d^2 + \sigma^4)}{m^2}.$$

Similarly, we can use Lemma 18 to bound

$$\mathbb{E}\left[ d(y(K\delta), x^*)^4 \right] \leq \frac{2^{26} L_R^2 L_\beta'^8 d^4}{m^{12}} + \frac{128 L_\beta'^2 \mathcal{R}^4}{m^2} + \frac{512 d^2}{m^2}.$$

Combining the above, we can bound the expected squared distance between $y(K\delta)$ and $x_K$ under the low probability event $A_K^c$. By choosing $r$ to be sufficiently large in (A.28), we make (A.35) sufficiently small. We then apply Young's inequality to bound $d(y(K\delta), x_K)^2 \leq 2 d(y(K\delta), x^*)^2 + 2 d(x_K, x^*)^2$ followed by Cauchy Schwarz, to verify that

$$\mathbb{E}\left[ \mathbb{1}\{A_K^c\} d(y(K\delta), x_K)^2 \right] \leq \sqrt{\delta}. \quad \text{(A.36)}$$

**Step 2: Discrete Contraction**
Having established a bound in the distance under the event $A_K^c$, we now turn our attention to the high-probability event $A_K$.

Consider some fixed but arbitrary $k \leq K$. We define a useful intermediate variable, representing a single Euler-Murayama step initialized at $y(k\delta)$ (recall that $y(t)$ corresponds to the exact SDE):

$$\bar{y}_{k+1} := \text{Exp}_{y(k\delta)} \left( \delta \beta(y(k\delta)) + \sqrt{\delta} \bar{\zeta}_k \right).$$

where $\bar{\zeta}_k \sim \mathcal{N}_{y(k\delta)}(0, I)$. Note that $\bar{y}_{k+1}$ evolves according to the Euler-Murayama step with *exact drift* $\beta(y(k\delta))$, whereas $x_{k+1}$ evolves according to the Euler-Murayama step with *stochastic drift* $\tilde{\beta}_k(x_k)$.

Under the strong-convexity-like condition due to Assumption 2 with $\mathcal{R} = 0$, we can show that a single discrete Euler Murayama step leads to contraction in distance between $\bar{y}_{k+1}$ and $x_{k+1}$. From Lemma 8, there exists a coupling such that

$$\mathbb{E}\left[ \mathbb{1}\{A_k\} d(\bar{y}_{k+1}, x_{k+1})^2 \right] \leq (1 - \delta(m - L_{Ric}/2)) \mathbb{E}\left[ \mathbb{1}\{A_k\} d(y(k\delta), x_k)^2 \right] + 16 \delta^2 \sigma^2. \quad \text{(A.37)}$$

**Step 3: Euler Murayama Error**
Next, we bound the distance between $\bar{y}_{k+1}$ and $y((k+1)\delta)$. This represents the discretization error between a single Euler-Murayama step with stepsize $\delta$, and the exact Langevin diffusion over $\delta$ time.

We will apply Lemma 7 with $L_1 = L'_\beta r$. We verify that under the event $A_k$, $\|\beta(y(k\delta))\|$ is indeed bounded by $L_1$. Thus by Lemma 7,

$$
\begin{aligned}
\mathbb{E}\left[\mathbb{1}\left\{A_k\right\}\mathrm{d}(y((k+1)\delta),\bar{y}_{k+1})^2\right] &\leq 2^{20}(\delta^4 L_1^4(1+L_R)+\delta^4 {L'_\beta}^4+\delta^3(d^3(L_R+{L'_\beta}^2/L_1^2)+{L'_\beta}^2 d))\\
&\leq 2^{20}(\delta^4 {L'_\beta}^4 r^4(1+L_R)+\delta^4 {L'_\beta}^4+\delta^3(d^3(L_R+1/\mathcal{R}^2)+{L'_\beta}^2 d))\\
&\leq \tilde{O}(\delta^3) \qquad\qquad\qquad\qquad\qquad\qquad\qquad\qquad\qquad (A.38)
\end{aligned}
$$

where $\tilde{O}$ hides polynomial dependency on $L'_\beta, d, L_R, \mathcal{R}, \log K, \log(1/\delta)$. This shows that, in one $\delta$-time step, the distance contracts with rate $m - L_{Ric}/2$, plus a discretization error of order $\delta^2$.

Combining (A.37) and (A.38) and using Young's inequality inequality, together with the fact that $A_{k+1} \subset A_k$.

$$
\mathbb{E}\left[\mathbb{1}\left\{A_{k+1}\right\}\mathrm{d}(y((k+1)\delta),x_{k+1})^2\right] \leq e^{-\delta(m-L_{Ric}/2)}\mathbb{E}\left[\mathbb{1}\left\{A_k\right\}\mathrm{d}(y(k\delta),x_k)^2\right] + \tilde{O}(\delta^2),
$$

where $\tilde{O}$ hides polynomial dependency on $L'_\beta, d, L_R, \mathcal{R}, \sigma, \log K, \log(1/\delta)$.

Applying the above recursively, we can bound

$$
\mathbb{E}\left[\mathbb{1}\left\{A_K\right\}\mathrm{d}(y(K\delta),x_K)^2\right] \leq e^{-K\delta(m-L_{Ric}/2)}\mathbb{E}\left[\mathrm{d}(y_0,x_0)^2\right] + \frac{1}{m-L_{Ric}/2}\cdot\tilde{O}(\delta). \quad (A.39)
$$

Combining (A.36) with (A.39) gives

$$
\mathbb{E}\left[\mathrm{d}(y(K\delta),x_K)^2\right] \leq e^{-K\delta(m-L_{Ric}/2)}\mathbb{E}\left[\mathrm{d}(y_0,x_0)^2\right] + \frac{1}{m-L_{Ric}/2}\cdot\tilde{O}(\delta),
$$

where $\tilde{O}$ hides polynomial dependency on $L'_\beta, d, L_R, \mathcal{R}, \sigma, \frac{1}{m-L_{Ric}/2}, \log K, \log\frac{1}{\delta}$. This concludes the proof of Theorem 2.

**Step 4: Verifying Conditions on $\delta$**
In the proof above, we applied Lemma 7, Lemma 8 and Lemma 20. Each of these requires certain bounds on $\delta$. In this step, we verify that the conditions on $\delta$ for each of these lemmas is satisfied by (A.33).

**Lemma 7 with $L_1 = L'_\beta r$ requires**

$$
\delta \leq \frac{1}{16}\min\left\{\frac{1}{L'_\beta},\frac{1}{L_R d},\frac{1}{\sqrt{L_R}L'_\beta r}\right\}.
$$

We verify that this follows from (A.29). We specifically verify that $\delta \leq \frac{1}{16\sqrt{L_R}L'_\beta r}$ is satisfied due to the last two terms in (A.29), and Lemma 37.

**Lemma 8 with $L_1 = rL'_\beta$ requires**

$$
\begin{aligned}
\delta\\
\leq \min&\left\{\frac{m-L_{Ric}/2}{128{L'_\beta}^2},\frac{m-L_{Ric}/2}{32L_R\sigma^2},\frac{m-L_{Ric}/2}{2^{13}L_R L'_\beta d},\frac{(m-L_{Ric}/2)^2}{2^{24}{L'_R}^2 d^3},\frac{m-L_{Ric}/2}{2^{17}d^2 L_R^2},\frac{m-L_{Ric}/2}{32L_R L_\beta^2},\sqrt{\frac{m-L_{Ric}/2}{2^{14}(L_\beta^3+\sigma^3)L'_R}}\right\}\\
= \min&\left\{\frac{m-L_{Ric}/2}{128{L'_\beta}^2},\frac{m-L_{Ric}/2}{32L_R\sigma^2},\frac{m-L_{Ric}/2}{2^{13}L_R L'_\beta d},\frac{(m-L_{Ric}/2)^2}{2^{24}{L'_R}^2 d^3},\frac{m-L_{Ric}/2}{2^{17}d^2 L_R^2},\frac{m-L_{Ric}/2}{32L_R {L'_\beta}^2 r^2},\sqrt{\frac{m-L_{Ric}/2}{2^{14}({L'_\beta}^3 r^3+\sigma^3)L'_R}}\right\}
\end{aligned}
$$
$$(A.40)$$

The first 5 bounds in (A.40) follow from (A.30). The last two bounds in (A.40) follow from (A.31) and Lemma 37.

**Lemma 20 requires** requires $\delta \leq \min\left\{\frac{m}{16{L'_\beta}^2(1+\sqrt{L_R}r)},\frac{d+\sigma^2}{m(1+\sqrt{L_R}r)},\frac{32(d^2+\sigma^4)}{m^2 r^2}\right\}$. This is satisfied by (A.32) and Lemma 37.

$\square$

The following lemma shows that, for any two initial points $x$ and $y$, if $x$ undergoes an *exact* Euler Murayama step with drift $\beta$, and $y$ undergooes a *stochastic* Euler Murayama step with drift $\tilde{\beta}$, then their expected squared distance contracts, with rate $m - L_{Ric}/2$, plus an additional error of $\delta^2\sigma^2$, where $\sigma = \left\|\tilde{\beta} - \beta\right\|$. This lemma is somewhat analogous to Lemma 3 which shows contraction under the exact SDE, though Lemma 8 also requires a fair amount of additional discretization analysis.

The key result used in the proof of Lemma 8 is Lemma 29.

**Lemma 8.** *Let $M$ satisfy Assumption 1, Assumption 4. Assume in addition that there exists a constant $L'_R$ such that for all $x \in M$, $u, v, w, z, a \in T_x M$, $\langle(\nabla_a R)(u, v)w, z\rangle \leq L'_R\|u\|\|v\|\|w\|\|z\|\|a\|$. Let $\beta$ be a deterministic vector field satisfying Assumption 3 and Assumption 2, with $\mathcal{R} = 0$. Let $x, y \in M$ be arbitrary. Let $\tilde{\beta}$ be a random vector field such that $\mathbb{E}\left[\tilde{\beta}\right] = \beta$. Assume that there exists $\sigma \in \mathbb{R}^+$ such that $\left\|\tilde{\beta}(y) - \beta(y)\right\| \leq \sigma$. Let $L_\beta := \max\{\|\beta(x)\|, \|\beta(y)\|\}$. Let $x' = \mathrm{Exp}_x\left(\delta\beta(x) + \sqrt{\delta}\zeta\right)$. Let $y' = \mathrm{Exp}_y\left(\delta\tilde{\beta}(y) + \sqrt{\delta}\tilde{\zeta}\right)$, where $\zeta \sim \mathcal{N}_x(0, I)$ and $\tilde{\zeta} \sim \mathcal{N}_y(0, I)$.*

*Assume that*

$$\delta \leq \min\left\{\frac{m - L_{Ric}/2}{128L'_\beta{}^2}, \frac{m - L_{Ric}/2}{32L_R L_\beta^2}, \frac{m - L_{Ric}/2}{32L_R\sigma^2}, \frac{m - L_{Ric}/2}{2^{13}L_R L'_\beta d}, \frac{(m - L_{Ric}/2)^2}{2^{24}L'_R{}^2 d^3}, \sqrt{\frac{m - L_{Ric}/2}{2^{14}(L_\beta^3 + \sigma^3)L'_R}}, \frac{m - L_{Ric}/2}{2^{17}d^2 L_R^2}\right\}.$$

*Then there is a coupling (synchronous coupling) between $\zeta$ and $\tilde{\zeta}$ such that*

$$\mathbb{E}\left[\mathrm{d}(x', y')^2\right] \leq (1 - \delta(m - L_{Ric}/2))\mathrm{d}(x, y)^2 + 16\delta^2\sigma^2$$

**Note:** elsewhere in this paper, we have used $L_\beta$ do denote a Lipschitz constant for $\beta$; the use of $L_\beta$ in Lemma 8 is different (but related).

*Proof.* Let $\gamma(s) : [0, 1] \to M$ be a minimizing geodesic between $x$ and $y$ with $\gamma(0) = x$ and $\gamma(1) = y$, such that $\langle\Gamma_y^x\beta(y) - \beta(x), \gamma'(0)\rangle \leq -m\mathrm{d}(x, y)^2$. (Assumption 2 guarantees the existence of such a $\gamma$.)

**Step 1: Synchronous Coupling of $\zeta$ and $\tilde{\zeta}$**
We will now define a coupling between $\zeta$ and $\tilde{\zeta}$. Let $E$ be an orthonormal basis at $T_x M$, and let $F$ be the parallel transport of $E$ along $\gamma$, i.e. $F$ is an orthonormal basis for $T_y M$. Let $\boldsymbol{\zeta} \sim \mathcal{N}(0, I)$ be a standard Gaussian random variable in $\mathbb{R}^d$, and define $\zeta := \boldsymbol{\zeta} \circ E$, and it follows by definition that $\zeta$ so defined has distribution $\mathcal{N}_x(0, I)$. Let $\tilde{\zeta} := \tilde{\boldsymbol{\zeta}} \circ F$, it follows by definition that $\tilde{\zeta}$ has distribution $\mathcal{N}_y(0, I)$.

**Step 2: Applying Lemma 29 and Simplifications**
We will apply Lemma 29 with $u = \delta\beta(x) + \sqrt{\delta}\zeta$ and $v = \delta\tilde{\beta}(y) + \sqrt{\delta}\tilde{\zeta}$. Then

$$\mathrm{d}(\mathrm{Exp}_x(u), \mathrm{Exp}_y(v))^2 - \mathrm{d}(x, y)^2$$
$$\leq \underbrace{2\langle\gamma'(0), v(0) - u(0)\rangle}_{①} + \underbrace{\|v(0) - u(0)\|^2}_{②}$$
$$\underbrace{- \int_0^1 \langle R(\gamma'(s), (1-s)u(s) + sv(s))(1-s)u(s) + sv(s), \gamma'(s)\rangle \, ds}_{③}$$
$$+ \underbrace{(2\mathcal{C}^2 e^{\mathcal{C}} + 18\mathcal{C}^4 e^{2\mathcal{C}})\|v(0) - u(0)\|^2 + (18\mathcal{C}^4 e^{2\mathcal{C}} + 4\mathcal{C}')\mathrm{d}(x, y)^2 + 4\mathcal{C}^2 e^{2\mathcal{C}}\mathrm{d}(x, y)\|v(0) - u(0)\|}_{④}$$

(A.41)

where $\mathcal{C} := \sqrt{L_R}(\|u\| + \|v\|)$ and $\mathcal{C}' := L'_R(\|u\| + \|v\|)^3$, and $u(t)$ and $v(t)$ are parallel trannsport of $u$ and $v$ along $\gamma$, as defined in Lemma 29. Some notes on notation:
1. We will use $\Gamma_x^y$ and $\Gamma_{\gamma(s)}^{\gamma(t)}$ to denote parallel transport along $\gamma$.
2. In subsequent parts, for $i = 1, 2...$, we will use

1. $\tau_i$ to denote terms which depend super-linearly on $\delta$.

2. $\xi_i$, to denote terms which have $0$ expectation.

3. $\theta_i$ to denote terms which depend linearly on $\delta$, and have non-zero expectation (i.e. the important terms).

**Step 2.1: Simplifying ①**

By definition, $v(0) - u(0) = \delta(\Gamma_y^x \tilde{\beta}(y) - \beta(x))$, thus

$$① = \underbrace{2\delta \left\langle \Gamma_y^x \beta(y) - \beta(x), \gamma'(0) \right\rangle}_{:=\theta_1} + \underbrace{2\delta \left\langle \Gamma_y^x \tilde{\beta}(y) - \Gamma_y^x \beta(y), \gamma'(0) \right\rangle}_{:=\xi_1} \tag{A.42}$$

**Step 2.2: Simplifying ②**

By similar algebra as Step 2.1,

$$② = \underbrace{\delta^2 \left\| \Gamma_y^x \tilde{\beta}(y) - \beta(x) \right\|^2}_{:=\tau_1} \leq \delta^2 {L'_\beta}^2 \mathrm{d}(x,y)^2 + \delta^2 \sigma^2 \tag{A.43}$$

**Step 2.3: Simplifying ③**

$$③ = -\int_0^1 \left\langle R(\gamma'(s), (1-s)u(s) + sv(s))(1-s)u(s) + sv(s), \gamma'(s) \right\rangle ds$$

$$= \underbrace{-\delta \int_0^1 \left\langle R(\gamma'(s), \Gamma_x^{\gamma(s)} \zeta) \Gamma_x^{\gamma(s)} \zeta, \gamma'(s) \right\rangle ds}_{:=\theta_2}$$

$$\underbrace{-\delta^2 \int_0^1 \left\langle R(\gamma'(s), \Gamma_x^{\gamma(s)} \beta(x)) \Gamma_x^{\gamma(s)} \beta(x), \gamma'(s) \right\rangle ds}_{:=\tau_2}$$

$$\underbrace{-2\delta^{3/2} \int_0^1 \left\langle R(\gamma'(s), \Gamma_x^{\gamma(s)} \zeta) \Gamma_x^{\gamma(s)} \beta(x), \gamma'(s) \right\rangle ds}_{:=\xi_2}$$

$$\underbrace{-2\delta^2 \int_0^1 s \left\langle R(\gamma'(s), \Gamma_x^{\gamma(s)} \beta(x)) \Gamma_y^{\gamma(s)} \tilde{\beta}(y) - \Gamma_x^{\gamma(s)} \beta(x), \gamma'(s) \right\rangle ds}_{:=\tau_3}$$

$$\underbrace{-2\delta^{3/2} \int_0^1 s \left\langle R(\gamma'(s), \Gamma_x^{\gamma(s)} \zeta) \Gamma_y^{\gamma(s)} \tilde{\beta}(y) - \Gamma_x^{\gamma(s)} \beta(x), \gamma'(s) \right\rangle ds}_{:=\xi_3}. \tag{A.44}$$

We will now bound $\tau_2$ and $\tau_3$. By Assumption 3,

$$|\tau_2| \leq 2\delta^2 L_R L_\beta^2 \mathrm{d}(x,y)^2$$

$$|\tau_3| \leq 4\delta^2 L_R (L_\beta^2 + \sigma^2) \mathrm{d}(x,y)^2$$

**Step 2.4: Simplifying ④**

Since ④ has quite a few terms, we will bound them one by one:

$$④ = \underbrace{2\mathcal{C}^2 e^{\mathcal{C}} \|v(0) - u(0)\|^2}_{:=\tau_5} + \underbrace{18\mathcal{C}^4 e^{2\mathcal{C}} \|v(0) - u(0)\|^2}_{:=\tau_6} + \underbrace{18\mathcal{C}^4 e^{2\mathcal{C}} \mathrm{d}(x,y)^2}_{:=\tau_7}$$

$$+ \underbrace{4\mathcal{C}' \mathrm{d}(x,y)^2}_{:=\tau_8} + \underbrace{4\mathcal{C}^2 e^{2\mathcal{C}} \mathrm{d}(x,y) \|v(0) - u(0)\|}_{:=\tau_9}.$$

Recall that $\mathcal{C} := \sqrt{L_R}(\|u\| + \|v\|)$ and $\mathcal{C}' := L_R'(\|u\| + \|v\|)^3$. Following previous calculations, we can bound $\|v(0) - u(0)\| \leq \delta\sigma + \delta L_\beta' \mathrm{d}(x, y)$. We can also bound $\|u\| \leq \delta L_\beta + \sqrt{\delta}\|\zeta\|$ and $\|v\| \leq \delta L_\beta + \delta\sigma + \sqrt{\delta}\|\zeta\|$. Thus

$$
\begin{aligned}
\tau_5 =&\, 2\mathcal{C}^2 e^{\mathcal{C}}\|v(0) - u(0)\|^2 \\
\leq&\, 4L_R(\|u\|^2 + \|v\|^2)e^{\sqrt{L_R}(\|u\| + \|v\|)}\|v(0) - u(0)\|^2 \\
\leq&\, 16L_R(\delta^2(2L_\beta^2 + \sigma^2) + \delta\|\zeta\|^2) \cdot \exp\left(\sqrt{L_R}(\delta(2L_\beta + \sigma) + 2\sqrt{\delta}\|\zeta\|)\right) \\
&\, \cdot \left(\delta L_\beta + \sqrt{\delta}\|\zeta\|\right)^2 \\
\leq&\, 128 L_R e^{2\sqrt{\delta}\|\zeta\|} \cdot (\delta^2\|\zeta\|^4 + \delta^4(L_\beta^4 + \sigma^4)), \\
\mathbb{E}\left[\tau_5^4\right]^{1/4} \leq&\, 512 L_R \cdot (\delta d + \delta^2(L_\beta^2 + \sigma^2))(\delta^2\sigma^2 + \delta^2 L_\beta'^2 \mathrm{d}(x, y)^2),
\end{aligned}
$$

where for the third line, we use the fact that our bound on $\delta$ implies that $\delta \leq \min\left\{\frac{1}{32\sqrt{L_R}L_\beta}, \frac{1}{32\sqrt{L_R}\sigma}\right\}$.

By similar algebra, we verify that

$$
\begin{aligned}
\tau_6 \quad &= 18\mathcal{C}^4 e^{2\mathcal{C}}\|v(0) - u(0)\|^2 \\
&\leq 2048 L_R^2 e^{4\sqrt{\delta}\|\zeta\|} \cdot (\delta^2\|\zeta\|^4 + \delta^4(L_\beta^4 + \sigma^4))(\delta^2\sigma^2 + \delta^2 L_\beta'^2 \mathrm{d}(x, y)^2), \\
\tau_7 \quad &= 18\mathcal{C}^4 e^{2\mathcal{C}}\mathrm{d}(x, y)^2 \\
&\leq 2048 L_R^2 e^{4\sqrt{\delta}\|\zeta\|} \cdot (\delta^2\|\zeta\|^4 + \delta^4(L_\beta^4 + \sigma^4)) \cdot \mathrm{d}(x, y)^2, \\
\tau_8 \quad &= 4\mathcal{C}'\mathrm{d}(x, y)^2 \\
&\leq 128 L_R' e^{4\sqrt{\delta}\|\zeta\|} \cdot (\delta^{3/2}\|\zeta\|^3 + \delta^3(L_\beta^3 + \sigma^3)) \cdot \mathrm{d}(x, y)^2, \\
\tau_9 \quad &= 4\mathcal{C}^2 e^{2\mathcal{C}}\mathrm{d}(x, y)\|v(0) - u(0)\| \\
&\leq 128 L_R e^{4\sqrt{\delta}\|\zeta\|} \cdot (\delta\|\zeta\|^2 + \delta^2(L_\beta^2 + \sigma^2)) \cdot (\delta L_\beta'\mathrm{d}(x, y)^2 + \delta\sigma\mathrm{d}(x, y)), \\
\mathbb{E}\left[\tau_6^4\right]^{1/4} &\leq 2^{14}L_R^2 \cdot (\delta^2 d^2 + \delta^4(L_\beta^4 + \sigma^4))(\delta^2\sigma^2 + \delta^2 L_\beta'^2 \mathrm{d}(x, y)^2), \\
\mathbb{E}\left[\tau_7^4\right]^{1/4} &\leq 2^{14}L_R^2 \cdot (\delta^2 d^2 + \delta^4(L_\beta^4 + \sigma^4)) \cdot \mathrm{d}(x, y)^2, \\
\mathbb{E}\left[\tau_8^4\right]^{1/4} &\leq 512 L_R' \cdot (\delta^{3/2}d^{3/2} + \delta^3(L_\beta^3 + \sigma^3)) \cdot \mathrm{d}(x, y)^2, \\
\mathbb{E}\left[\tau_9^4\right]^{1/4} &\leq 512 L_R \cdot (\delta d + \delta^2(L_\beta^2 + \sigma^2)) \cdot (\delta L_\beta'\mathrm{d}(x, y)^2 + \delta\sigma\mathrm{d}(x, y)), \quad\quad (A.45)
\end{aligned}
$$

where we use Lemma 21 and the fact that our assumption on $\delta$ implies that $\delta \leq \frac{1}{2^{14}L_R d}$.

**Step 3: Putting Things Together**

Let $\mathbb{E}[\cdot]$ denote expectation wrt the $\zeta$ and $\tilde{\beta}$ (which is a random function). By Assumption 2,

$$\mathbb{E}[\theta_1] \leq -2\delta m \mathrm{d}(x, y)^2.$$

By definition of Ricci curvature and by Assumption 1,

$$\mathbb{E}[\theta_2] \leq \delta L_{Ric}.$$

Using our assumed bound on $\delta$, we verify that

$$\mathbb{E}[\tau_1 + \tau_2 + \tau_3 + \tau_5 + \tau_6 + \tau_7 + \tau_8 + \tau_9] \leq \frac{\delta(m - L_{Ric}/2)}{2}\mathrm{d}(x, y)^2 + 16\delta^2\sigma^2.$$

We omit the proof for this fact, since it follows by basic but long algebra, but note that we need to apply Young's inequality at several points. We also verify that

$$\mathbb{E}[\xi_1] = \mathbb{E}[\xi_2] = \mathbb{E}[\xi_3] = 0.$$

Plugging everything into (A.41), we get

$$\mathbb{E}\left[\mathrm{d}(x', y')^2\right] \leq (1 - \delta(m - L_{Ric}/2))\mathrm{d}(x, y)^2 + 16\delta^2\sigma^2.$$

$\square$

# B   Distance Contraction under Kendall Cranston Coupling

In this section, we prove Lemma 3, which is the main tool for proving mixing of manifold diffusion processes under the distant dissipativity assumption. We note again that the proof is entirely based on existing results from [Eberle, 2016, Hsu, 2002], and is only included for completeness.

## B.1   The Kendall Cranston Coupling

**Lemma 9.** *Let $T \in \mathbb{R}^+$ be some fixed time. Assume that there is are constants $L_\beta, L'_\beta$ such that for all $x, y \in M$, $\|\beta(x)\| \leq L_\beta$ and $\left\|\beta(x) - \Gamma^x_y \beta(y)\right\| \leq L'_\beta \|x - y\|$. Let $i$ be some integer satisfying $i \geq \max \left\{ \log_2 \left( 32 T \sqrt{L_R} L_\beta \right), \log_2 \left( 32 T d \right), \log_2 \left( 32 L_\beta T \right) \right\}$.*

*For any $x, y$, let $\Lambda(x, y)$ denote the set of minimizing geodesics from $x$ to $y$, i.e. for any $\gamma \in \Lambda(x, y)$, $\gamma(0) = x$, $\gamma(1) = y$, $\forall t, \nabla_{\gamma'(t)} \gamma'(t) = 0$ and $d(x, y) = \|\gamma'(0)\|$. Let $\kappa(r) := \frac{1}{r^2} \sup_{d(x,y) = r} \inf_{\gamma \in \Lambda(x,y)} \left\langle \Gamma^x_y \beta(y) - \beta(x), \gamma'(0) \right\rangle$.*

*Let $x, y \in M$ and let $E^x$ be an arbitrary orthonormal basis of $T_x M$ and let $E^y$ be an arbitrary orthonormal basis of $T_y$. let $x^i(t) := \overline{\Phi}(t; x, E^x, \beta, \mathbf{B}^x, i)$ and $y^i(t) := \overline{\Phi}(t; y, E^y, \beta, \mathbf{B}^y, i)$ where $\mathbf{B}^x$ and $\mathbf{B}^y$ are standard Brownian motion in $\mathbb{R}^d$, and where $\overline{\Phi}$ is as defined in (A.3).*

*For any $\varepsilon$, there exists a coupling between $\mathbf{B}^x$ and $\mathbf{B}^y$, and Brownian motion $\mathbf{W}^i$ over $\mathbb{R}$, such that for all $k \in \left\{ 0 \ldots 2^i \right\}$,*

$$
\begin{aligned}
d(x^i_{k+1}, y^i_{k+1})^2 \leq &(1 + \delta^i (2\kappa(d(x^i_k, y^i_k)) + L_{Ric})) d(x^i_k, y^i_k)^2 \\
&+ \mathbb{1}\left\{ d(x^i_k, y^i_k) > \varepsilon \right\} (4\delta^i - 4d(x^i_k, y^i_k)(\mathbf{W}^i((k+1)\delta^i) - \mathbf{W}^i(k\delta^i))) \\
&+ \tau^i_k
\end{aligned}
$$

*where $\tau^i_k$ satisfies*

$$
\begin{aligned}
\mathbb{E}_{\mathcal{F}_k} \left[ \left| \tau^i_k \right| \right] &\leq \mathcal{C}_1 \delta^{i\,3/2} (1 + L^4_\beta)(1 + d(x^i_k, y^i_k)^2) \\
\mathbb{E}_{\mathcal{F}_k} \left[ \tau^{i\,2}_k \right] &\leq \mathcal{C}_1 (1 + d(x^i_k, y^i_k)^4) \delta^{i\,2}
\end{aligned}
$$

*where $\mathcal{C}_1$ is a constant depending on $L_R, L'_R, d, T$.*

*Proof.* We set up some notation: throughout this proof, consider a fixed $i$. Recall that $\delta^i := T/2^i$, and assume $i$ is large enough such that $\delta^i \leq \frac{1}{32\sqrt{L_R} L_\beta}$. Let $x^i_k$ be as defined in (4.1) so that $x^i_k = x^i(k\delta^i)$. Let us also define $K := 2^i$, so that $T = K\delta^i$.

**Step 1: defining the coupling** By definition, for any $k \in \{0 \ldots K\}$,

$$
\begin{aligned}
x^i_{k+1} &:= \mathrm{Exp}_{x^i_k} \left( \delta^i \beta(x^i_k) + (\mathbf{B}^x((k+1)\delta^i) - \mathbf{B}^x(k\delta^i)) \circ E^i_k \right) \\
y^i_{k+1} &:= \mathrm{Exp}_{y^i_k} \left( \delta^i \beta(y^i_k) + (\mathbf{B}^y((k+1)\delta^i) - \mathbf{B}^y(k\delta^i)) \circ \tilde{E}^i_k \right)
\end{aligned}
$$

Let $\gamma^i_k : [0, 1] \to M$ denote a minimizing geodesic from $x^i_k$ to $y^i_k$.

Let $F^i_k$ be an orthonormal basis at $T_{y^i_k} M$, obtained from the parallel transport of $E^i_k$ along $\gamma^i_k$, i.e. for all $j = 1 \ldots d$,

$$
F^{i,j}_k = \Gamma_{\gamma^i_k} E^{i,j}_k
$$

Let us define $\mathbf{M}^i_k \in \mathbb{R}^{d \times d}$ as matrix whose $a, b$ entry is

$$
\left[ \mathbf{M}^i_k \right]_{a,b} = \left\langle F^{i,a}_k, \tilde{E}^{i,b}_k \right\rangle
$$

one can verify that $\mathbf{M}^i_k$ is an orthogonal matrix, and that for all $\mathbf{v} \in \mathbb{R}^d$, $\mathbf{v} \circ F^i_k = \mathbf{M}\mathbf{v} \circ \tilde{E}^i_k$.

Let us define $\bar{\boldsymbol{\nu}}^i_k$ denote the unique coordinates of $\frac{\gamma^i_k{}'(1)}{\|\gamma^i_k{}'(1)\|}$ wrt $F^i_k$ (equivalently the coordinates of $\frac{\gamma^i_k{}'(0)}{\|\gamma^i_k{}'(0)\|}$ wrt $E^i_k$). We define $\boldsymbol{\nu}^i_k := \mathbb{1}\left\{ d(x^i_k, y^i_k) > \varepsilon \right\} \bar{\boldsymbol{\nu}}^i_k$.

We now define a coupling between $\mathbf{B}^x(t)$ and $\mathbf{B}^y(t)$ as follows:

$$\mathbf{B}^y(t) := \int_0^T \mathbb{1}\left\{t \in [k\delta^i, (k+1)\delta^i)\right\}\mathbf{M}_k^i(I - 2\boldsymbol{\nu}_k^i\boldsymbol{\nu}_k^{i\,T})d\mathbf{B}^x(t)$$

For this to be a valid coupling, it suffices to verify that
$\int_0^T \mathbb{1}\left\{t \in [k\delta^i, (k+1)\delta^i)\right\}\mathbf{M}_k^i(I - 2\boldsymbol{\nu}_k^i\boldsymbol{\nu}_k^{i\,T})d\mathbf{B}^x(t)$ is indeed a standard Brownian motion. This can be done by verifying that the definition satisfies Levy's characterization of Brownian motion. We omit the proof, but highlight two important facts: 1. $\int_0^T \mathbb{1}\left\{t \in [k\delta^i, (k+1)\delta^i)\right\}\mathbf{M}_k^i(I - 2\boldsymbol{\nu}_k^i\boldsymbol{\nu}_k^{i\,T})$ is adapted to the natural filtration of $\mathbf{B}^x(t)$, and 2. $\mathbf{M}_k^i(I - 2\boldsymbol{\nu}_k^i\boldsymbol{\nu}_k^{i\,T})$ is an orthogonal matrix. We have thus defined a coupling between $\mathbf{B}^x$ and $\mathbf{B}^y$, and consequently, a coupling between $x^i(t)$ and $y^i(t)$ for all $t$.

### Step 2: Applying Lemma 29

Having defined a coupling between $x_k^i$ and $y_k^i$, we bound $\mathbb{E}\left[\mathrm{d}(x_K^i, y_K^i)^2\right]$ for $K := T/\delta^i = 2^i$ by applying Lemma 29 , with $x = x_k^i$, $y = y_k^i$, $u = \delta^i\beta(x_k^i) + (\mathbf{B}((k+1)\delta^i) - \mathbf{B}(k\delta^i)) \circ E_k^i$, $v = \delta^i\beta(y_k^i) + (\tilde{\mathbf{B}}((k+1)\delta^i) - \tilde{\mathbf{B}}(k\delta^i)) \circ \tilde{E}_k^i$ and $\gamma := \gamma_k^i$.

Following the notation in Lemma 29, let $u(t)$ and $v(t)$ be the parallel transport of $u$ and $v$ along $\gamma(t)$. We verify that $u(s) = \delta^i\Gamma_{x_k^i}^{\gamma_k^i(s)}\beta(x_k^i) + (\mathbf{B}((k+1)\delta^i) - \mathbf{B}(k\delta^i)) \circ \Gamma_{x_k^i}^{\gamma_k^i(s)}E_k^i$ and that

$$
\begin{aligned}
v(s) =& \delta^i\Gamma_{y_k^i}^{\gamma_k^i(s)}\beta(y_k^i) + \mathbf{M}_k^i(I - 2\boldsymbol{\nu}_k^i\boldsymbol{\nu}_k^{i\,T})(\mathbf{B}((k+1)\delta^i) - \mathbf{B}(k\delta^i)) \circ \Gamma_{x_k^i}^{\gamma_k^i(s)}\tilde{E}_k^i \\
=& \delta^i\Gamma_{y_k^i}^{\gamma_k^i(s)}\beta(y_k^i) + (I - 2\boldsymbol{\nu}_k^i\boldsymbol{\nu}_k^{i\,T})(\mathbf{B}((k+1)\delta^i) - \mathbf{B}(k\delta^i)) \circ \Gamma_{y_k^i}^{\gamma_k^i(s)}F_k^i \\
=& \delta^i\Gamma_{y_k^i}^{\gamma_k^i(s)}\beta(y_k^i) + (I - 2\boldsymbol{\nu}_k^i\boldsymbol{\nu}_k^{i\,T})(\mathbf{B}((k+1)\delta^i) - \mathbf{B}(k\delta^i)) \circ \Gamma_{x_k^i}^{\gamma_k^i(s)}E_k^i \\
=& \delta^i\Gamma_{y_k^i}^{\gamma_k^i(s)}\beta(y_k^i) + (\mathbf{B}((k+1)\delta^i) - \mathbf{B}(k\delta^i)) \circ \Gamma_{x_k^i}^{\gamma_k^i(s)}E_k^i \\
& - 2\left\langle\boldsymbol{\nu}_k^i, \mathbf{B}((k+1)\delta^i) - \mathbf{B}(k\delta^i)\right\rangle\frac{\gamma_k^{i\,\prime}(s)}{\left\|\gamma_k^{i\,\prime}(s)\right\|}
\end{aligned}
$$

where the second equality is by definition of $\mathbf{M}_k^i$, the third equality is by definition of $F_k^i$, the fourth equality is by definition of $\boldsymbol{\nu}_k^i$ and the fact that $\gamma_k^i$ is a geodesic. It is convenient subsequently to note the following:

$$
\begin{aligned}
& v(s) - u(s) \\
=& \delta^i(\Gamma_{y_k^i}^{\gamma_k^i(s)}\beta(y_k^i) - \Gamma_{x_k^i}^{\gamma_k^i(s)}\beta(x_k^i)) - 2\left\langle\boldsymbol{\nu}_k^i, \mathbf{B}((k+1)\delta^i) - \mathbf{B}(k\delta^i)\right\rangle\frac{\gamma_k^{i\,\prime}(s)}{\left\|\gamma_k^{i\,\prime}(s)\right\|}
\end{aligned}
$$

and

$$
\begin{aligned}
& (1-s)u(s) + sv(s) \\
=& (1-s)\delta^i\Gamma_{x_k^i}^{\gamma_k^i(s)}\beta(x_k^i) + s\delta^i\Gamma_{y_k^i}^{\gamma_k^i(s)}\beta(y_k^i) \\
& + (\mathbf{B}((k+1)\delta^i) - \mathbf{B}(k\delta^i)) \circ \Gamma_{x_k^i}^{\gamma_k^i(s)}E_k^i \\
& - 2s\left\langle\boldsymbol{\nu}_k^i, \mathbf{B}((k+1)\delta^i) - \mathbf{B}(k\delta^i)\right\rangle\frac{\gamma_k^{i\,\prime}(s)}{\left\|\gamma_k^{i\,\prime}(s)\right\|}
\end{aligned}
$$

**Step 3: Reorganizing Lemma 29**

With $u, v$ as defined above, Lemma 29 implies that

$$d(x_{k+1}^i, y_{k+1}^i)^2 - d(x_k^i, y_k^i)^2$$

$$\leq 2\left\langle \gamma_k^{i\,\prime}(0), v(0) - u(0)\right\rangle + \|v(0) - u(0)\|^2$$

$$- \int_0^1 \left\langle R(\gamma_k^{i\,\prime}(s), (1-s)u(s) + sv(s))(1-s)u(s) + sv(s), \gamma_k^{i\,\prime}(s)\right\rangle ds$$

$$+ (2\mathcal{C}^2 e^{\mathcal{C}} + 18\mathcal{C}^4 e^{2\mathcal{C}})\|v(0) - u(0)\|^2 + (18\mathcal{C}^4 e^{2\mathcal{C}} + 4\mathcal{C}')d(x_k^i, y_k^i)^2$$

$$+ 4\mathcal{C}^2 e^{2\mathcal{C}} d(x_k^i, y_k^i)\|v(0) - u(0)\| \tag{B.1}$$

where $\mathcal{C} := \sqrt{L_R}(\|u\| + \|v\|)$ and $\mathcal{C}' := L_R'(\|u\| + \|v\|)^3$.

Below, we bound each of the terms above

$$2\left\langle \gamma_k^{i\,\prime}(0), v(0) - u(0)\right\rangle = 2\delta^i\left\langle \gamma_k^{i\,\prime}(0), \Gamma_{y_k^i}^{x_k^i}\beta(y_k^i) - \beta(x_k^i)\right\rangle - 4\left\|\gamma_k^{i\,\prime}(0)\right\|\left\langle \boldsymbol{\nu}_k^i, \mathbf{B}((k+1)\delta^i) - \mathbf{B}(k\delta^i)\right\rangle$$

$$\|v(0) - u(0)\|^2 \quad \leq 4\left\langle \boldsymbol{\nu}_k^i, \mathbf{B}((k+1)\delta^i) - \mathbf{B}(k\delta^i)\right\rangle^2$$

$$+ \underbrace{\delta^{i^2}L_\beta^2 + 4\delta^i L_\beta\left|\left\langle \boldsymbol{\nu}_k^i, \mathbf{B}((k+1)\delta^i) - \mathbf{B}(k\delta^i)\right\rangle\right|}_{\tau_{k,1}^i}$$

$$- \int_0^1 \left\langle R(\gamma_k^{i\,\prime}(s), (1-s)u(s) + sv(s))(1-s)u(s) + sv(s), \gamma_k^{i\,\prime}(s)\right\rangle ds$$

$$\leq -\int_0^1 \left\langle R(\gamma_k^{i\,\prime}(s), (\mathbf{B}((k+1)\delta^i) - \mathbf{B}(k\delta^i)) \circ \Gamma_{x_k^i}^{\gamma_k^i(s)}E_k^i)(\mathbf{B}((k+1)\delta^i) - \mathbf{B}(k\delta^i)) \circ \Gamma_{x_k^i}^{\gamma_k^i(s)}E_k^i, \gamma_k^{i\,\prime}(s)\right\rangle ds$$

$$+ \underbrace{\delta^{i^2}L_R d(x_k^i, y_k^i)^2 L_\beta^2 + 4\delta^i L_R d(x_k^i, y_k^i)^2 L_\beta\left\|\mathbf{B}((k+1)\delta^i) - \mathbf{B}(k\delta^i)\right\|_2}_{\tau_{k,2}^i}$$

In the first equality above, we crucially use the fact that $\left\langle \boldsymbol{\nu}_k^i, \mathbf{B}((k+1)\delta^i) - \mathbf{B}(k\delta^i)\right\rangle \frac{\gamma_k^{i\,\prime}(s)}{\|\gamma_k^{i\,\prime}(s)\|}$ is a scalar multiple of $\gamma_k'(s)$, and the fact that $\langle R(u,u)v, u\rangle = \langle R(u,v)u, u\rangle = 0$ for all $u, v$ by symmetry of the Riemannian curvature tensor.

Finally, we will take the remaining terms, and denote them by

$$\tau_{k,3}^i := (2\mathcal{C}^2 e^{\mathcal{C}} + 18\mathcal{C}^4 e^{2\mathcal{C}})\|v(0) - u(0)\|^2$$

$$+ (18\mathcal{C}^4 e^{2\mathcal{C}} + 4\mathcal{C}')d(x_k^i, y_k^i)^2 + 4\mathcal{C}^2 e^{2\mathcal{C}} d(x_k^i, y_k^i)\|v(0) - u(0)\|$$

We claim that under our assumption on $i$,

$$\mathbb{E}_{\mathcal{F}_k}\left[|\tau_{k,1}^i + \tau_{k,2}^i + \tau_{k,3}^i|\right] = O(\delta^{i^{3/2}}(1 + L_\beta^4)(1 + d(x_k^i, y_k^i)^2))$$

where $O()$ hides dependencies on $L_R, L_R', d, T$.

We omit the proof for the above claim, which involves some tedious but straightforward algebra, but we note that the proof uses $\mathbb{E}\left[\left\|\mathbf{B}((k+1)\delta^i) - \mathbf{B}(k\delta^i)\right\|_2^j\right] = O(\delta^{i^{j/2}})$ (for all integer $j$) and that $\mathbb{E}\left[\exp\left(a\left\|\mathbf{B}((k+1)\delta^i) - \mathbf{B}(k\delta^i)\right\|_2\right)\right] \leq 4\exp\left(2a^2\delta^i d\right) \leq 8$ for $\delta^i a^2 \leq 1/32$ (see Lemma 21). It is also important to use our assumption on $\delta^i$ in the lemma statement.

We simplify (B.1) to

$$d(x_{k+1}^i, y_{k+1}^i)^2 - d(x_k^i, y_k^i)^2$$

$$\leq 2\delta^i \kappa(d(x_k^i, y_k^i))d(x_k^i, y_k^i)^2 - 4d(x_k^i, y_k^i)\left\langle \boldsymbol{\nu}_k^i, \mathbf{B}((k+1)\delta^i) - \mathbf{B}(k\delta^i)\right\rangle$$

$$+ 4\left\langle \boldsymbol{\nu}_k^i, \mathbf{B}((k+1)\delta^i) - \mathbf{B}(k\delta^i)\right\rangle^2$$

$$- \int_0^1 \left\langle R(\gamma_k^{i\,\prime}(s), (\mathbf{B}((k+1)\delta^i) - \mathbf{B}(k\delta^i)) \circ \Gamma_{x_k^i}^{\gamma_k^i(s)}E_k^i)(\mathbf{B}((k+1)\delta^i) - \mathbf{B}(k\delta^i)) \circ \Gamma_{x_k^i}^{\gamma_k^i(s)}E_k^i, \gamma_k^{i\,\prime}(s)\right\rangle ds$$

$$+ \tau_{k,1}^i + \tau_{k,2}^i + \tau_{k,3}^i \tag{B.2}$$

**Step 4: Pulling out the expectation**

We will further simplify (B.2) by replacing a few terms by their expectations. Define

$$\tau_{k,4}^i := \int_0^1 \left\langle R(\gamma_k^{i\,\prime}(s), (\mathbf{B}((k+1)\delta^i) - \mathbf{B}(k\delta^i)) \circ \Gamma_{x_k^i}^{\gamma_k^i(s)} E_k^i), (\mathbf{B}((k+1)\delta^i) - \mathbf{B}(k\delta^i)) \circ \Gamma_{x_k^i}^{\gamma_k^i(s)} E_k^i, \gamma_k^{i\,\prime}(s)\right\rangle$$
$$- \delta^i Ric(\gamma_k^{i\,\prime}(s)) ds$$
$$\tau_{k,5}^i := \delta^i - \left\langle \boldsymbol{\nu}_k^i, \mathbf{B}((k+1)\delta^i) - \mathbf{B}(k\delta^i) \right\rangle^2$$

By definition of Ricci Curvature and by Assumption 1,

$$- \mathbb{E}_{\mathcal{F}_k}\left[ \int_0^1 \left\langle R(\gamma_k^{i\,\prime}(s), (\mathbf{B}((k+1)\delta^i) - \mathbf{B}(k\delta^i)) \circ \Gamma_{x_k^i}^{\gamma_k^i(s)} E_k^i), (\mathbf{B}((k+1)\delta^i) - \mathbf{B}(k\delta^i)) \circ \Gamma_{x_k^i}^{\gamma_k^i(s)} E_k^i, \gamma_k^{i\,\prime}(s)\right\rangle ds\right]$$
$$\leq - \delta^i L_{Ric}\left\|\gamma_k^{i\,\prime}(s)\right\|^2 = \leq -\delta^i L_{Ric} \mathrm{d}(x_k^i, y_k^i)^2$$

where $Ric$ denotes the Ricci curvature tensor.

By definition of $\boldsymbol{\nu}_k^i$, $\mathbb{E}\left[\left\langle \boldsymbol{\nu}_k^i, \mathbf{B}((k+1)\delta^i) - \mathbf{B}(k\delta^i)\right\rangle^2\right] = \delta^i \mathbb{1}\left\{\mathrm{d}(x_k^i, y_k^i) > \varepsilon\right\}$.

Let $\tau_k^i := \tau_{k,1}^i + \tau_{k,2}^i + \tau_{k,3}^i$. We can thus further simplify (B.2) to

$$\mathrm{d}(x_{k+1}^i, y_{k+1}^i)^2 - \mathrm{d}(x_k^i, y_k^i)^2$$
$$\leq 2\delta^i \kappa(\mathrm{d}(x_k^i, y_k^i))\mathrm{d}(x_k^i, y_k^i)^2 - 4\mathrm{d}(x_k^i, y_k^i)\left\langle \boldsymbol{\nu}_k^i, \mathbf{B}((k+1)\delta^i) - \mathbf{B}(k\delta^i)\right\rangle$$
$$+ 4\delta^i \mathbb{1}\left\{\mathrm{d}(x_k^i, y_k^i) > \varepsilon\right\} - \delta^i L_{Ric}\mathrm{d}(x_k^i, y_k^i)^2$$
$$+ \tau_k^i$$
$$\leq \delta^i(2\kappa(\mathrm{d}(x_k^i, y_k^i)) + L_{Ric})\mathrm{d}(x_k^i, y_k^i)^2$$
$$- 4\mathrm{d}(x_k^i, y_k^i)\left\langle \boldsymbol{\nu}_k^i, \mathbf{B}((k+1)\delta^i) - \mathbf{B}(k\delta^i)\right\rangle + 4\delta^i \mathbb{1}\left\{\mathrm{d}(x_k^i, y_k^i) > \varepsilon\right\}$$
$$+ \tau_k^i \tag{B.3}$$

the conclusion follows by defining $\mathbf{W}^i(t) := \int_0^t \mathbb{1}\left\{t \in [k\delta^i, (k+1)\delta^i]\right\}\left\langle \bar{\boldsymbol{\nu}}_k^i, \mathbf{B}((k+1)\delta^i) - \mathbf{B}(k\delta^i)\right\rangle$ and verifying that it is a Brownian motion. (Recall our definition that $\boldsymbol{\nu}_k^i :=$ $\mathbb{1}\left\{\mathrm{d}(x_k^i, y_k^i) > \varepsilon\right\}\bar{\boldsymbol{\nu}}_k^i$) $\qquad\square$

## B.2 Lyapunov function and its smooth approximation

In this section, we consider a Lyapunov function $f$ taken from Eberle [2016]. By analyzing how $f(\mathrm{d}(x_k^i, y_k^i))$ evolves under the dynamic in Lemma 9, one can demonstrate that the distance function contracts.

Let $\mathcal{L}, \mathcal{R} \in \mathbb{R}^+$. We will see later that $\mathcal{L}$ and $\mathcal{R}$ will correspond to distant-dissipativity parameters in (2).

Let $\varepsilon \in [0, \infty)$. One should think of $\varepsilon$ as being arbitrarily small, as eventually we are only interested in the limit as $\varepsilon \to 0$.

Define functions $\psi_\varepsilon(r)$, $\Psi_\varepsilon(r)$ and $\nu(r)$, all from $\mathbb{R}^+$ to $\mathbb{R}$:

$$\mu_\varepsilon(r) = \begin{cases} 1, & \text{for } r \leq \mathcal{R} \\ 1 - (r - \mathcal{R})/(\varepsilon), & \text{for } r \in \mathcal{R}, \mathcal{R} + \varepsilon \\ 0, & \text{for } r \geq \mathcal{R} + \varepsilon \end{cases}$$

$$\nu_\varepsilon(r) := 1 - \frac{1}{2}\frac{\int_0^r \frac{\mu_\varepsilon(s)\Psi_\varepsilon(s)}{\psi_\varepsilon(s)}ds}{\int_0^\infty \frac{\mu_\varepsilon(s)\Psi_\varepsilon(s)}{\psi_\varepsilon(s)}ds} \qquad\qquad \psi_\varepsilon(r) := e^{-\frac{\mathcal{L}\int_0^r r\mu_\varepsilon(r)dr}{2}}$$

$$\Psi_\varepsilon(r) := \int_0^r \psi_\varepsilon(s)ds,$$

We defined an $\varepsilon$-smoothed Lyapunov function as

**Definition 1.**

$$f_\varepsilon(r) := \int_0^r \psi_\varepsilon(s)\nu_\varepsilon(s)ds$$

$$g_\varepsilon(s) = f_\varepsilon(\sqrt{s+\varepsilon})$$

The case when $\varepsilon = 0$ (when there is no smoothing) will be of particular interest to us:

**Definition 2.**

$$f(r) := f_0(r) = g_0(r)$$

**Remark 2.** The Lyapunov function from Eberle [2016] is more general, but for the specific case of $\mathcal{L}, \mathcal{R}$ distant dissipative functions, it is equal to $f$ as defined in (2).

**Lemma 10.** *Assume $\varepsilon \in [0, 1/(4\sqrt{\mathcal{L}})]$, then $f_\varepsilon$ as defined in (1) satisfies*

1. $f_\varepsilon(r) \in [\dfrac{1}{2}\exp\left(-(1+\varepsilon)\mathcal{L}\mathcal{R}^2/2\right)r, r]$             *for all $r$*

2. $f_\varepsilon'(r) \in [\dfrac{1}{2}\exp\left(-(1+\varepsilon)\mathcal{L}\mathcal{R}^2/2\right), 1]$             *for all $r$*

3. $f_\varepsilon''(r) \in [-4\mathcal{L}^{3/2}, 0]$                                   *for all $r$*

4. $f_\varepsilon''(r) + \mathcal{L}rf_\varepsilon'(r) \le -\dfrac{\exp\left(-(1+\varepsilon)\mathcal{L}\mathcal{R}^2/2\right)}{(1+\varepsilon)^2\mathcal{R}^2}f_\varepsilon(r)$     *for $r \in [0, \mathcal{R}]$*

*If in addition, $\varepsilon > 0$, $f_\varepsilon$ satisfies*

5. $|f_\varepsilon'''(r)| \le \dfrac{256\sqrt{\mathcal{L}}}{\varepsilon}$       *for all $r$*

*Proof.* We can verify that

$$
\begin{aligned}
f_\varepsilon'(r) &= \psi_\varepsilon(r)\nu_\varepsilon(r) \\
f_\varepsilon''(r) &= \psi_\varepsilon'(r)\nu_\varepsilon(r) + \psi_\varepsilon(r)\nu_\varepsilon'(r) \\
&= -\mathcal{L}\mu_\varepsilon(r)r\psi_\varepsilon(r)\nu_\varepsilon(r) + \psi_\varepsilon(r)\nu_\varepsilon'(r) \\
f_\varepsilon'''(r) &= -\mathcal{L}\psi_\varepsilon(r)\nu_\varepsilon(r) + \mathcal{L}r\psi_\varepsilon(r)\mu_\varepsilon'(r) + \mathcal{L}^2r^2\psi_\varepsilon(r)\nu_\varepsilon(r) - 2\mathcal{L}r\psi_\varepsilon(r)\nu_\varepsilon'(r) + \psi_\varepsilon(r)\nu_\varepsilon''(r)
\end{aligned}
$$

1. follows from integrating 2.

2. follows from $\nu_\varepsilon(r) \in [1/2, 1]$ and $\psi_\varepsilon \in [\exp\left(-(1+\varepsilon)\mathcal{L}\mathcal{R}^2/2\right), 1]$ and the expression for $f_\varepsilon'(r)$ above.

3. follows from $\mu_\varepsilon, \psi_\varepsilon, \nu_\varepsilon \ge 0$ and $\nu_\varepsilon' \le 0$, and the fact that $r\psi_\varepsilon(r) \le 2\sqrt{\mathcal{L}}$ and (B.4).

4. is a little more involved. First note that over $r \in [0, \mathcal{R}]$, $\mu_\varepsilon(r) = 1$. This will simplify some calculations. From the expression for $f_\varepsilon''$ above, we verify

$$f_\varepsilon''(r) + \mathcal{L}rf_\varepsilon'(r) = \psi_\varepsilon(r)\nu_\varepsilon'(r) = -\frac{\Psi_\varepsilon(r)}{2\int_0^\infty \frac{\mu_\varepsilon(s)\Psi_\varepsilon(s)}{\psi_\varepsilon(s)}ds}$$

We can bound the denominator as

$$\int_0^\infty \frac{\mu_\varepsilon(s)\Psi_\varepsilon(s)}{\psi_\varepsilon(s)}ds \le \int_0^{\mathcal{R}+\varepsilon} \frac{\Psi_\varepsilon(s)}{\psi_\varepsilon(s)}ds \le \frac{\int_0^{\mathcal{R}+\varepsilon}\Psi_\varepsilon(s)ds}{\psi(\mathcal{R}+\varepsilon)} \le \frac{(1+\varepsilon)^2\mathcal{R}^2}{2\exp\left(-\mathcal{L}(1+\varepsilon)\mathcal{R}^2/2\right)}$$

where the first inequality is by $\mu_\varepsilon(s) \le 1$, and $\mu_\varepsilon(r) = 0$ for $r \ge \mathcal{R} + \varepsilon$ the second inequality is by $\psi_\varepsilon(r)$ being monotonically decreasing, and the third inequality is by $\Psi_\varepsilon(r) \le r$. Finally, note that $\Psi_\varepsilon(r) \ge f_\varepsilon(r)$. Put together,

$$f_\varepsilon''(r) + \mathcal{L}rf_\varepsilon'(r) \le -\frac{\exp\left(-(1+\varepsilon)\mathcal{L}\mathcal{R}^2/2\right)}{(1+\varepsilon)^2\mathcal{R}^2}f_\varepsilon(r)$$

We now prove the bound for 5. It is useful to recall that $\psi_\varepsilon(r) \leq 1$ and $\nu_\varepsilon(r) \leq 1$.

$$\Psi_\varepsilon(r) = \int_0^r \exp\left(-\mathcal{L}s^2\right)ds \leq \frac{4}{\sqrt{\mathcal{L}}}$$

$$\int_0^\infty \frac{\mu_\varepsilon(s)\Psi_\varepsilon(s)}{\psi_\varepsilon(s)}ds \geq \int_0^{\mathcal{R}} \frac{\Psi_\varepsilon(s)}{\psi_\varepsilon(s)}ds \geq \frac{1}{2}\int_0^{1/\sqrt{2\mathcal{L}}} \Psi_\varepsilon(s)ds \geq \frac{1}{16\mathcal{L}} \tag{B.4}$$

$$|\psi_\varepsilon(r)\nu_\varepsilon'(r)| \leq \frac{\Psi(\mathcal{R}+\varepsilon)}{2\int_0^\infty \frac{\mu_\varepsilon(s)\Psi_\varepsilon(s)}{\psi_\varepsilon(s)}ds} \leq 8\sqrt{\mathcal{L}}$$

For $r \in [0, \mathcal{R}+\varepsilon]$ ($\nu_\varepsilon'' = 0$ outside this range),

$$|\psi_\varepsilon(r)\nu_\varepsilon''(r)| \leq \frac{\frac{1}{\varepsilon}\Psi(r) + r\psi_\varepsilon(r)/\varepsilon + \psi_\varepsilon(r) + 2r\Psi_\varepsilon(r)/\psi_\varepsilon(r)}{2\int_0^\infty \frac{\mu_\varepsilon(s)\Psi_\varepsilon(s)}{\psi_\varepsilon(s)}ds} \leq 32\mathcal{L}\cdot\Psi_\varepsilon(r)\cdot\left(\frac{2}{\varepsilon}+2\mathcal{L}\mathcal{R}\right) \leq \frac{128\sqrt{\mathcal{L}}}{\varepsilon}$$

We can thus bound $|f'''(r)|$ as

$$|f_\varepsilon'''(r)| \leq 2\mathcal{L} + 16\mathcal{L}^{3/2}\mathcal{R} + \frac{128\sqrt{\mathcal{L}}}{\varepsilon} \leq \frac{256\sqrt{\mathcal{L}}}{\varepsilon}$$

$\square$

**Lemma 11.** *Assume $\varepsilon \in (0, 1/(4\sqrt{\mathcal{L}})]$*

1. $g_\varepsilon'(s) = \dfrac{1}{2\sqrt{s+\varepsilon}}f_\varepsilon'(\sqrt{s+\varepsilon})$

2. $g_\varepsilon''(s) = \dfrac{1}{4(s+\varepsilon)}f_\varepsilon''(\sqrt{s+\varepsilon}) - \dfrac{1}{4(s+\varepsilon)^{3/2}}f_\varepsilon'(\sqrt{s+\varepsilon})$

3. $g_\varepsilon'''(s) = \dfrac{1}{8(s+\varepsilon)^{3/2}}f_\varepsilon'''(\sqrt{s+\varepsilon}) - \dfrac{1}{8(s+\varepsilon)^2}f_\varepsilon''(\sqrt{s+\varepsilon}) + \dfrac{1}{6(s+\varepsilon)^{5/2}}f_\varepsilon'(\sqrt{s+\varepsilon})$

4. $|g_\varepsilon'''(s)| \leq O(\varepsilon^{-5/2})$ *for all $s$*

*where $O()$ notation hides dependency on $\mathcal{L}$ and $\mathcal{R}$.*

*Proof.* The first 3 points follow from chain rule.

The last point follows from point 5 from Lemma 10.

$$|g_\varepsilon'''(s)| \leq \frac{64\sqrt{\mathcal{L}}}{\varepsilon^{5/2}\mathcal{R}} + \frac{\sqrt{\mathcal{L}}}{\varepsilon^2} + \frac{1}{\varepsilon^{5/2}}$$

$\square$

### B.3 Contraction of Lyapunov Function under Kendall Cranston Coupling

**Lemma 12.** *Consider the same setup as Lemma 9. For any $x, y$, let $\Lambda(x,y)$ denote the set of minimizing geodesics from $x$ to $y$, i.e. for any $\gamma \in \Lambda(x,y)$, $\gamma(0) = x$, $\gamma(1) = y$, $\forall t, \nabla_{\gamma'(t)}\gamma'(t) = 0$ and $d(x,y) = \|\gamma'(0)\|$. Let $\kappa(r) := \frac{1}{r^2}\sup_{d(x,y)=r}\inf_{\gamma \in \Lambda(x,y)}\left\langle \Gamma_y^x\beta(y) - \beta(x), \gamma'(0)\right\rangle$.*

*Assume there exists $\mathcal{R} \geq 0, q \leq 0$ such that $\kappa(r) \leq q$ for all $r \leq \mathcal{R}$. Let $\mathcal{L} = q + L_{Ric}/2$. Let $\varepsilon \in (0, 1/(4\sqrt{\mathcal{L}})]$. Let $g_\varepsilon$ be as defined in 1 with parameters $\mathcal{L}$ and $\mathcal{R}$. Let $\mathcal{F}_k$ denote the natural filtration generated by $x_k^i$ and $y_k^i$.*

*There exists a constant $c_1$, depending on $L_\beta, L_\beta', L_R, T, d$, and some constant $c_2$, depending on $L_\beta', L_{Ric}, \mathcal{R}$ such that for any $i > c_1$ and $\varepsilon > c_2$, there exists a coupling between $x_k^i$ and $y_k^i$ such that*

$$\mathbb{E}\left[g_\varepsilon(d(x_{k+1}^i, y_{k+1}^i)^2)\right]$$

$$\leq \mathbb{E}\left[\mathbb{1}\{r > \mathcal{R}\}\delta^i((\kappa(r_k) + L_{Ric}/2)\exp\left(-(1+\varepsilon)\mathcal{L}\mathcal{R}^2/2\right)/8)g_\varepsilon(d(x_{k+1}^i, y_{k+1}^i)^2)\right]$$

$$- \frac{\exp\left(-(1+\varepsilon)\mathcal{L}\mathcal{R}^2/2\right)}{2(1+\varepsilon)^2\mathcal{R}^2}\delta^i\mathbb{E}\left[\mathbb{1}\{r \leq \mathcal{R}\}g_\varepsilon(d(x_{k+1}^i, y_{k+1}^i)^2)\right] + O(\delta^i\varepsilon^{1/2} + \varepsilon^{-5/2}\delta^{i3/2})$$

*where $O()$ hides dependency on $L_R, L_\beta', T, d$.*

*Proof.* Let us define, for convenience, $r_k := \mathrm{d}(x_k^i, y_k^i)$. By Lemma 9, for any $i$ and any $\varepsilon$, there exists a coupling satisfying

$$
\begin{aligned}
r_{k+1}^2 \leq &(1 + \delta^i(2\kappa(r_k) + L_{Ric}))r_k^2 \\
&+ \mathbb{1}\left\{r_k > \varepsilon^{1/3}\right\}(4\delta^i - 4r_k \mathbf{W}^i((k+1)\delta^i) - \mathbf{W}^i(k\delta^i)) + \tau_k^i
\end{aligned}
$$

where $\tau_k^i$ satisfies

$$
\mathbb{E}\left[|\tau_k^i|\right] \leq O(\delta^{i3/2}(1 + L_\beta^4)(1 + r_k^2)) \qquad \mathbb{E}\left[\tau_k^{i\,2}\right] \leq O((1 + r_k^4)\delta^{i2})
$$

where $O()$ hides dependencies on $L_R, L_R', d, T$.

By third order Taylor expansion,

$$
\begin{aligned}
&\mathbb{E}\left[g_\varepsilon(r_{k+1}^2)\right] \\
=&\mathbb{E}\left[g_\varepsilon(r_k^2)\right] \\
&+ \mathbb{E}\left[g_\varepsilon'(r_k^2) \cdot (\delta^i(2\kappa(r_k) + L_{Ric}))r_k^2\right] \\
&+ \mathbb{E}\left[g_\varepsilon'(r_k^2) \cdot 4\delta^i\right] \\
&+ \mathbb{E}\left[\frac{1}{2}g_\varepsilon''(r_k^2) \cdot (4r_k \mathbb{1}\left\{r_k > \varepsilon^{1/3}\right\}(\mathbf{W}^i((k+1)\delta^i) - \mathbf{W}^i(k\delta^i)))^2\right] \\
&+ O(\varepsilon^{-5/2}\delta^{i3/2}) \qquad\qquad\qquad\qquad\qquad\qquad\qquad\qquad\qquad\qquad (\text{B.5})
\end{aligned}
$$

The last line uses two facts:

1. From Lemma 13, for any $j$, there exists a constant $\mathcal{C}$, depending on $T, d, L_R, L_\beta'$, but independent of $L_\beta$, such that for all $i, k$, $\mathbb{E}\left[\mathrm{d}(x_k^i, x_0)^{2j}\right] < \mathcal{C}$ and $\mathbb{E}\left[\mathrm{d}(y_k^i, y_0)^{2j}\right] < \mathcal{C}$.

2. Roughly speaking, $\mathbb{E}\left[\mathrm{d}(x_{k+1}^i, y_{k+1}^i)^2 - \mathrm{d}(x_k^i, y_k^i)^2\right] = O(\delta^{i3/2})$. More specifically:

$$
\begin{aligned}
&\left|\mathrm{d}(x_{k+1}^i, y_{k+1}^i) - \mathrm{d}(x_k^i, y_k^i)\right| \\
&\leq 2\mathrm{d}(x_k^i x_{k+1}^i) + 2\mathrm{d}(y_k^i y_{k+1}^i) \\
&\leq 2\delta^i(\|\beta(x_0)\| + \|\beta(y_0)\| + L_\beta'\mathrm{d}(\mathrm{d}(x_k^i, x_0)) + L_\beta'\mathrm{d}(\mathrm{d}(x_k^i, x_0))) \\
&\quad + 4\|\mathbf{B}((k+1)\delta^i) - \mathbf{B}(k\delta^i)\|_2
\end{aligned}
$$

Plugging in the definition of $g_\varepsilon'$ and $g_\varepsilon''$,

$$
\begin{aligned}
&g_\varepsilon'(r_k^2) \cdot (\delta^i((2\kappa(r_k) + L_{Ric}))r_k^2 + 4\mathbb{1}\left\{r_k > \varepsilon^{1/3}\right\}\delta^i) \\
=&\frac{\delta^i}{2\sqrt{r_k^2 + \varepsilon}}f_\varepsilon'(\sqrt{r_k^2 + \varepsilon})((2\kappa(r_k) + L_{Ric})r_k^2 + 4\mathbb{1}\left\{r_k > \varepsilon^{1/3}\right\}) \\
\leq&\frac{\delta^i}{2\sqrt{r_k^2 + \varepsilon}}f_\varepsilon'(\sqrt{r_k^2 + \varepsilon})((2\kappa(r_k) + L_{Ric})r_k^2) + 2\mathbb{1}\left\{r_k > \varepsilon^{1/3}\right\}\frac{\delta^i f_\varepsilon'(\sqrt{r_k^2 + \varepsilon})}{\sqrt{r_k^2 + \varepsilon}}
\end{aligned}
$$

where we use the assumption that $\varepsilon \leq \frac{1}{4\mathcal{R}^2}$ and $\varepsilon < 1$.

On the other hand,

$$
\begin{aligned}
&\mathbb{E}_{\mathcal{F}_k}\left[\frac{1}{2}g_\varepsilon''(r_k^2) \cdot (4r_k \mathbb{1}\left\{r_k > \varepsilon^{1/3}\right\}(\mathbf{W}^i((k+1)\delta^i) - \mathbf{W}^i(k\delta^i)))^2\right] \\
=&8\delta^i r_k^2 g_\varepsilon''(r_k^2) \cdot \mathbb{1}\left\{r_k > \varepsilon^{1/3}\right\} \\
=&\mathbb{1}\left\{r_k > \varepsilon^{1/3}\right\}\frac{2\delta^i r_k^2}{r_k^2 + \varepsilon}f_\varepsilon''(\sqrt{r_k^2 + \varepsilon}) - \mathbb{1}\left\{r_k > \varepsilon^{1/3}\right\}\frac{2\delta^i r_k^2 f_\varepsilon'(\sqrt{r_k^2 + \varepsilon})}{(r_k^2 + \varepsilon)^{3/2}}
\end{aligned}
$$

Note that $r_k > \varepsilon^{1/3}$ implies that $\frac{r_k^2}{(r_k^2+\varepsilon)^{3/2}} \geq \frac{1}{1+\varepsilon^{1/3}}$. Thus

$$2\mathbb{1}\left\{r_k > \varepsilon^{1/3}\right\}\frac{\delta^i f_\varepsilon'(\sqrt{r_k^2+\varepsilon})}{\sqrt{r_k^2+\varepsilon}} - \mathbb{1}\left\{r_k > \varepsilon^{1/3}\right\}\frac{2\delta^i r_k^2 f_\varepsilon'(\sqrt{r_k^2+\varepsilon})}{(r_k^2+\varepsilon)^{3/2}} \leq 4\delta^i \varepsilon^{1/3} \tag{B.6}$$

where we use the fact that $|f_\varepsilon'| \leq 1$.

We now bound $\frac{\delta^i}{2\sqrt{r_k^2+\varepsilon}}f_\varepsilon'(\sqrt{r_k^2+\varepsilon})((2\kappa(r_k)+L_{Ric})r_k^2)+\mathbb{1}\left\{r_k > \varepsilon^{1/3}\right\}\frac{2\delta^i r_k^2}{r_k^2+\varepsilon}f_\varepsilon''(\sqrt{r_k^2+\varepsilon})$. Consider three cases:

1. $r_k \leq \varepsilon^{1/3}$:

$$\frac{\delta^i}{2\sqrt{r_k^2+\varepsilon}}f_\varepsilon'(\sqrt{r_k^2+\varepsilon})((2\kappa(r_k)+L_{Ric})r_k^2) \leq \delta^i(q+L_{Ric}/2)\varepsilon^{1/2}$$

2. $r_k \in (\varepsilon^{1/3}, \mathcal{R}]$:

$$\frac{\delta^i}{2\sqrt{r_k^2+\varepsilon}}f_\varepsilon'(\sqrt{r_k^2+\varepsilon})((2\kappa(r_k)+L_{Ric})r_k^2) + \frac{2\delta^i r_k^2}{r_k^2+\varepsilon}f_\varepsilon''(\sqrt{r_k^2+\varepsilon})$$

$$\leq \frac{\delta^i r_k^2}{r_k^2+\varepsilon}(\mathcal{L}f_\varepsilon'(\sqrt{r_k^2+\varepsilon})\sqrt{r_k^2+\varepsilon}+2f_\varepsilon''(\sqrt{r_k^2+\varepsilon}))$$

$$\leq -\frac{\exp\left(-(1+\varepsilon)\mathcal{L}\mathcal{R}^2/2\right)}{2(1+\varepsilon)^2\mathcal{R}^2}\delta^i f_\varepsilon(\sqrt{r_k^2+\varepsilon})$$

where we use Lemma 10 and the definition of $\mathcal{L}$.

3. $r_k > \mathcal{R}$: We use the fact that $f_\varepsilon''(r) \leq 0$ for all $r \geq \mathcal{R} \geq \varepsilon$. Thus

$$\frac{\delta^i}{2\sqrt{r_k^2+\varepsilon}}f_\varepsilon'(\sqrt{r_k^2+\varepsilon})((2\kappa(r_k)+L_{Ric})r_k^2) + \frac{2\delta^i r_k^2}{r_k^2+\varepsilon}f_\varepsilon''(\sqrt{r_k^2+\varepsilon})$$

$$\leq \frac{\delta^i}{2\sqrt{r_k^2+\varepsilon}}f_\varepsilon'(\sqrt{r_k^2+\varepsilon})((2\kappa(r_k)+L_{Ric})r_k^2)$$

$$\leq -\frac{\delta^i((\kappa(r_k)+L_{Ric}/2))r_k^2 f_\varepsilon'(\sqrt{r_k^2+\varepsilon})}{8\sqrt{r_k^2+\varepsilon}}$$

$$\leq -\frac{1}{8}\delta^i((\kappa(r_k)+L_{Ric}/2)\exp\left(-(1+\varepsilon)\mathcal{L}\mathcal{R}^2/2\right))f_\varepsilon(\sqrt{r_k^2+\varepsilon})$$

$\square$

*Proof of Lemma 3.* Let $E^x$ be an orthonormal basis of $T_{x(0)}M$, $E^y$ be an orthonormal basis of $T_{y(0)}M$, and let $\mathbf{B}^x$ and $\mathbf{B}^y$ denote two Brownian motions which may be coupled in a non-trivial way. By definition of $\Phi$ in (A.4) and by Lemma 2, $x(t) = \Phi(t; x(0), E^x, \beta, \mathbf{B}^x)$ and $y(t) = \Phi(t; y(0), E^y, \beta, \mathbf{B}^y)$, where equivalence is in the sense of distribution.

Lemma 12 almost gives us what we need. However, because we assumed that $\beta$ satisfies Assumption 2, the assumption that $\|\beta(x)\| \leq L_\beta$ cannot possibly hold. We thus need to approximate $\beta$ by a sequence of increasingly non-Lipschitz functions.

Consider a fixed $i$. Let $s^j$ be a sequence of increasing radius, such that $s^j \to \infty$ as $j \to \infty$. Let $\beta^j$ denote the truncation of $\beta$ to norm $s^j$, i.e.

$$\beta^j(x) := \begin{cases} \beta(x) & \text{for } \|\beta(x)\| \leq s^j \\ \beta(x) \cdot \frac{s^j}{\|\beta(x)\|} & \text{for } \|\beta(x)\| > s^j \end{cases} .$$

We verify that $\beta^j$ also satisfies Assumption 3 with the same $L_\beta'$ as $\beta$.

Consider some fixed $j$. Let us now define the Euler Murayama discretization of $x(t)$ and $y(t)$ as

$$x^i(t) := \overline{\Phi}(t; x(0), E^x, \beta, \mathbf{B}^x, i)$$
$$y^i(t) := \overline{\Phi}(t; y(0), E^y, \beta, \mathbf{B}^y, i).$$

Where $\overline{\Phi}$ is as defined in (A.3), and is a short-hand for the (interpolated) Euler Murayama sequence with stepsize $\delta^i = T/2^i$, defined in (4.2) (equivalently (A.2)). It is by definition that $x(t) = \lim_{i \to \infty} x^i(t)$ (and similarly for $y(t)$ and $y^i(t)$).

Furthermore, define, for all $i, j$,

$$\tilde{x}^{i,j}(t) := \overline{\Phi}(t; x(0), E^x, \beta^j, \mathbf{B}^x, i),$$
$$\tilde{y}^{i,j}(t) := \overline{\Phi}(t; y(0), E^y, \beta^j, \mathbf{B}^y, i),$$
$$\tilde{x}^{\cdot,j}(t) := \Phi(t; x(0), E^x, \beta^j, \mathbf{B}^x),$$
$$\tilde{y}^{\cdot,j}(t) := \Phi(t; y(0), E^y, \beta^j, \mathbf{B}^y).$$

Note that the above definition implies a non-trivial coupling between $\tilde{x}^{i,j}(t)$ and $x^i(t)$, via the shared Brownian motion $\mathbf{B}^x$. In words, $\tilde{x}^{\cdot,j}(t)$ denotes the exact Langevin SDE, but with drift given by $\beta^j$ (the truncated version of $\beta$), and $\tilde{x}^{i,j}$ denotes the (interpolated) Euler Murayama discretization of $\tilde{x}^{\cdot,j}$ with stepsize $\delta^i$.

Let $L_0 := \max\{\|\beta(x(0))\|, \|\beta(y(0))\|\}$. Let us define $\tilde{r}_k^{i,j} := \mathrm{d}(\tilde{x}_k^{i,j}, \tilde{y}_k^{i,j})$. Let $\kappa(r)$ be as defined in the statement of Lemma 12.

Using Assumption 2, we verify that

$$\mathbb{1}\left\{\tilde{r}_k^{i,j} > \mathcal{R}\right\}(\kappa(\tilde{r}_k^{i,j}) + L_{Ric}/2)$$
$$< \mathbb{1}\left\{\mathcal{R} < \tilde{r}_k^{i,j} \leq \frac{s^j - L_0}{L_\beta'}\right\}(-m + L_{Ric}/2) + \mathbb{1}\left\{\mathcal{R} < \tilde{r}_k^{i,j}, \frac{s^j - L_0}{L_\beta'} \leq \tilde{r}_k^{i,j}\right\}\left(\frac{s^j}{\tilde{r}_k^{i,j}} + L_{Ric}/2\right).$$

Let $q, \mathcal{R}$ be the parameters in Assumption 2. This implies that $\kappa(r) \leq q$ for all $r \leq \mathcal{R}$. Let $\mathcal{L} := q + L_{Ric}/2$ and $\varepsilon$ be as defined in Lemma 12. Let $g_\varepsilon$ be as defined in Definition 1 with parameters $\mathcal{L}$ and $\mathcal{R}$. Then

$$\mathbb{E}\left[g_\varepsilon(r_{k+1}^2)\right]$$
$$\leq \mathbb{E}\left[\mathbb{1}\left\{r > \mathcal{R}\right\}\delta^i((\kappa(\tilde{r}_k^{i,j}) + L_{Ric}/2)\exp\left(-(1+\varepsilon)\mathcal{L}\mathcal{R}^2/2\right)/8)g_\varepsilon(r_{k+1}^2)\right]$$
$$\quad - \frac{\exp\left(-(1+\varepsilon)\mathcal{L}\mathcal{R}^2/2\right)}{2(1+\varepsilon)^2\mathcal{R}^2}\delta^i\mathbb{E}\left[\mathbb{1}\left\{r \leq \mathcal{R}\right\}g_\varepsilon(r_{k+1}^2)\right] + O(\delta^i\varepsilon^{1/2} + \varepsilon^{-5/2}\delta^{i3/2})$$
$$\leq -\frac{\delta^i(m - L_{Ric}/2)\exp\left(-(1+\varepsilon)\mathcal{L}\mathcal{R}^2/2\right)}{16}\mathbb{E}\left[\mathbb{1}\left\{\mathcal{R} < \tilde{r}_k^{i,j} \leq \frac{s^j - L_0}{L_\beta'}\right\}g_\varepsilon(r_{k+1}^2)\right]$$
$$\quad - \frac{\delta^i\exp\left(-(1+\varepsilon)\mathcal{L}\mathcal{R}^2/2\right)}{2(1+\varepsilon)^2\mathcal{R}^2}\mathbb{E}\left[\mathbb{1}\left\{r \leq \mathcal{R}\right\}g_\varepsilon(r_{k+1}^2)\right]$$
$$\quad + \delta^i\mathbb{E}\left[\mathbb{1}\left\{\frac{s^j - L_0}{L_\beta'} < \tilde{r}_k^{i,j}\right\}\left(s^j + \frac{1}{2}L_{Ric}\tilde{r}_k^{i,j}\right)\right] + O(\delta^i\varepsilon^{1/2} + \varepsilon^{-5/2}\delta^{i3/2})$$
$$\leq -\alpha_\varepsilon\delta^i\mathbb{E}\left[g_\varepsilon(r_{k+1}^2)\right] + \delta^i\mathbb{E}\left[\mathbb{1}\left\{\frac{s^j - L_0}{L_\beta'} < \tilde{r}_k^{i,j}\right\}\left(s^j + (m + L_{Ric}/2)\tilde{r}_k^{i,j}\right)\right] + O(\delta^i\varepsilon^{1/2} + \varepsilon^{-5/2}\delta^{i3/2})$$

$$\tag{B.7}$$

where we define $\alpha_\varepsilon := \min\left\{\frac{m - L_{Ric}/2}{16}, \frac{1}{2(1+\varepsilon)^2\mathcal{R}^2}\right\}\exp\left(-\frac{1}{2}(1+\varepsilon)\mathcal{L}\mathcal{R}^2\right)$. The first line follows from Lemma 12, the second line simply splits $\mathbb{1}\{r > \mathcal{R}\}$ into two cases: $\mathbb{1}\left\{\mathcal{R} < \tilde{r}_k^{i,j} \leq \frac{s^j - L_0}{L_\beta'}\right\}$ and $\mathbb{1}\left\{\frac{s^j - L_0}{L_\beta'} < \tilde{r}_k^{i,j}\right\}$, and bounds $\kappa(r) \leq -m$ when $r > \mathcal{R}$ under Assumption 2. The third inequality is by definition of $\alpha_\varepsilon$.

Applying (B.7) recursively for $k = 0...K$, where $K = T/2^i$, we get that

$$\mathbb{E}\left[g_\varepsilon((\tilde{r}_K^{i,j})^2)\right]$$

$$\leq \exp\left(-\alpha_\varepsilon K \delta^i\right)\mathbb{E}\left[g_\varepsilon(r_0^2)\right] + O(T\varepsilon^{1/2} + T\varepsilon^{-5/2}\delta^{i1/2})$$

$$+ \delta^i \sum_{k=0}^{K} \mathbb{E}\left[\mathbb{1}\left\{\frac{s^j - L_0}{L'_\beta} < \tilde{r}_k^{i,j}\right\}(s^j + (m + L_{Ric}/2)\tilde{r}_k^{i,j})\right] + O(T\varepsilon^{1/2} + \varepsilon^{-5/2}T\delta^{i1/2})$$

(B.8)

We now bound the second term of (B.8) more carefully:

$$\delta^i \sum_{k=0}^{K} \mathbb{E}\left[\mathbb{1}\left\{\frac{s^j - L_0}{L'_\beta} \leq \tilde{r}_k^{i,j}\right\}(s^j + (m + L_{Ric}/2)\tilde{r}_k^{i,j})\right]$$

$$\leq \delta^i \mathbb{E}\left[(\max_{k \leq K} \mathbb{1}\left\{\frac{s^j - L_0}{L'_\beta} \leq \tilde{r}_k^{i,j}\right\}) \sum_{k=0}^{K} (s^j + (m + L_{Ric}/2)\tilde{r}_k^{i,j})\right]$$

$$\leq \delta^i \mathbb{E}\left[(\mathbb{1}\left\{\frac{s^j - L_0}{L'_\beta} \leq \max_{k \leq K} \tilde{r}_k^{i,j}\right\}) \sum_{k=0}^{K} (s^j + (m + L_{Ric}/2)\tilde{r}_k^{i,j})\right]$$

$$\leq \delta^i \sqrt{\mathbb{P}(\frac{s^j - L_0}{L'_\beta} \leq \max_{k \leq K} \tilde{r}_k^{i,j})} \cdot \sqrt{\mathbb{E}\left[(\sum_{k=0}^{K} (s^j + (m + L_{Ric}/2)\tilde{r}_k^{i,j}))^2\right]}$$

$$\leq O(\frac{1}{s^j})$$

(B.9)

The last line is because of the following: from Lemma 16, $\delta^i \mathbb{P}(\frac{s^j - L_0}{L'_\beta} \leq \sup_{k \leq K} \tilde{r}_k^{i,j})^{1/2} = O(\frac{L'_\beta{}^2}{(s^j - L_0)^2}) = O(\frac{1}{s_j^2})$ assuming $j$ sufficiently large. Also from Lemma 16, $\delta^i \sqrt{\mathbb{E}\left[(\sum_{k=0}^{K} (s^j + (m + L_{Ric}/2)\tilde{r}_k^{i,j}))^2\right]} = O(T)$.

Plugging (B.9) into (B.8), and recalling the definition of $\tilde{r}$, and the fact that $\mathrm{d}(\tilde{x}_K^i, \tilde{y}_K^i) := \mathrm{d}(\tilde{x}^i(T), \tilde{y}^i(T))$,

$$\mathbb{E}\left[g_\varepsilon(\mathrm{d}(\tilde{x}^{i,j}(T), \tilde{y}^{i,j}(T)^2)\right] \leq \exp\left(-\alpha_\varepsilon K \delta^i\right)g_\varepsilon(\mathrm{d}(x(0), y(0))^2) + O(\varepsilon^{1/2} + \varepsilon^{-5/2}\delta^{i1/2} + \frac{1}{s^j})$$

where $O(\cdot)$ hides $T$ dependency as well.

First, by taking the limit of $i$ to infinity (e.g. for each $i$, we see that for any $j$ and any $\varepsilon$,

$$\lim_{i \to \infty} \mathbb{E}\left[g_\varepsilon(\mathrm{d}(\tilde{x}^{i,j}(T), \tilde{y}^{i,j}(T))^2)\right] \leq \exp\left(-\alpha_\varepsilon K \delta^i\right)g_\varepsilon(\mathrm{d}(x(0), y(0))^2) + O(\varepsilon^{1/2} + \frac{1}{s^j})$$

Let us define $\tilde{x}^{\cdot,j}(t)$ as the almost sure limit of $\tilde{x}^{i,j}(t)$, as $i \to \infty$, whose existence is shown in Lemma 5 (similarly for $\tilde{y}^{\cdot,j}(t)$). It follows that $g_\varepsilon(\mathrm{d}(\tilde{x}^{i,j}(T), \tilde{y}^{i,j}(T)^2)$ converges almost surely to $g_\varepsilon(\mathrm{d}(\tilde{x}^{\cdot,j}(T), \tilde{y}^{\cdot,j}(T))^2)$ as $i \to \infty$. By dominated convergence (Lemma 15 implies a single constant upper bounds $\mathbb{E}\left[\mathrm{d}(\tilde{x}^{i,j}(T), \tilde{y}^{i,j}(T)^2\right]$ for all $i$), $\mathbb{E}\left[g_\varepsilon(\mathrm{d}(\tilde{x}^{i,j}(T), \tilde{y}^{i,j}(T))^2)\right]$ converges to $\mathbb{E}\left[g_\varepsilon(\mathrm{d}(\tilde{x}^{\cdot,j}(T), \tilde{y}^{\cdot,j}(T))^2)\right]$ as $i \to \infty$. Let $\Omega$ denote the set of all couplings between $\tilde{x}^{\cdot,j}(t)$ and $\tilde{y}^{\cdot,j}(t)$. Then

$$\inf_\Omega \mathbb{E}\left[g_\varepsilon(\mathrm{d}(\tilde{x}^{\cdot,j}(T), \tilde{y}^{\cdot,j}(T))^2)\right]$$

$$\leq \lim_{i \to \infty} \mathbb{E}\left[g_\varepsilon(\mathrm{d}(\tilde{x}^{i,j}(T), \tilde{y}^{i,j}(T))^2)\right]$$

$$\leq \exp\left(-\alpha_\varepsilon K \delta^i\right)g_\varepsilon(\mathrm{d}(x(0), y(0))^2) + O(\varepsilon^{1/2} + \frac{1}{s^j}),$$

(B.10)

where the first inequality uses the fact that $\tilde{x}^{\cdot,j}(t)$ (resp $\tilde{y}^{\cdot,j}(t)$) is the limit, as $i \to \infty$, of $\tilde{x}^{i,j}(t)$ (resp $\tilde{y}^{i,j}(t)$).

From Lemma 16, we know that

$$\mathbb{P}(\sup_t \mathrm{d}(x(t), x(0)) \geq s) \leq O(\frac{1}{s^4}),$$

where we use the fact that, by definition in (A.3), $x^i(t)$ are linear interpolations of $x^i(k)$. Next, notice that when $\sup_{t \in [0,T]} \mathrm{d}(x(t), x(0)) \leq \frac{s^j - L_0}{L'_\beta}$, $x(t) = \tilde{x}^{\cdot,j}(t)$ for all $t \in [0, T]$. It thus follows that as $s^j \to \infty$, $\mathbb{E}\left[g_\varepsilon(\mathrm{d}(\tilde{x}^{\cdot,j}(T), \tilde{y}^{\cdot,j}(T)))\right]$ converges to $\mathbb{E}\left[g_\varepsilon(\mathrm{d}(x(T), y(T)))\right]$ almost surely. Thus taking limit of (B.10) as $j \to \infty$, i.e. $s^j \to \infty$,

$$\inf_\Omega \mathbb{E}\left[g_\varepsilon(\mathrm{d}(x(T), y(T))^2)\right]$$
$$\leq \exp\left(-\alpha_\varepsilon K \delta^i\right) g_\varepsilon(\mathrm{d}(x(0), y(0))^2) + O(\varepsilon^{1/2}).$$

Finally, take the limit of $\varepsilon \to 0$. Note that $g_\varepsilon(r^2) \to g_0(r^2) = f(r)$, where $f$ is defined in Definition 2. Note also that $\alpha_\varepsilon \to \alpha$ as defined in the lemma statement. Finally, the properties of $f$ follows from Lemma 10.

$\square$

# C    Tail Bounds

In this section, we establish various probability and moment bounds for and (1.1), (1.2) and (2.1). These bounds are used at many places in our proofs.

## C.1    One-Step Distance Bounds

### C.1.1    Under Lipschitz Continuity

**Lemma 13** (One-step distance evolution under Lipschitz Continuity). *Let $\beta$ be a vector field satisfying 3. Let $\delta \in \mathbb{R}^+$ be a stepsize satisfying $\delta \leq \frac{1}{16L'_\beta}$. Let $x_0 \in M$ be arbitrary, let $x_k$ denote the iterative process*

$$x_{k+1} = \mathrm{Exp}_{x_k}\left(\delta\beta(x_k) + \sqrt{\delta}\xi_k(x_k)\right)$$

*where $\xi_k$ denote a random variable that possibly depends on $x_k$. Let $\gamma_k(t) : [0,1] \to M$ denote any minimizing geodesic with $\gamma(0) = x_k$ and $\gamma(1) = x_0$. Then for any positive integer $K$ and for any $k \leq K$, we can bound,*

$$\mathrm{d}(x_{k+1}, x_0)^2 \leq (1 + 8\delta L'_\beta + \frac{1}{2K} + \delta L_R \|\xi_k(x_k)\|^2 + \delta^2 L_R L_0^2)\mathrm{d}(x_k, x_0)^2 + 2\delta\|\xi_k(x_k)\|^2 + 8K\delta^2 L_0^2$$

$$+ \mathbb{1}\left\{\mathrm{d}(x_k, x_0) \leq \frac{1}{\delta\sqrt{L_R}L'_\beta}\right\}\left(-2\left\langle\sqrt{\delta}\xi_k(x_k), \gamma'_k(0)\right\rangle\right)$$

*Proof.* We will be using the bound from Zhang and Sra [2016] (see Lemma 25). Let $v := \delta\beta(x_k) + \sqrt{\delta}\xi_k(x_k)$. Then Lemma 25 bounds

$$\mathrm{d}(x_{k+1}, x)^2 \leq \mathrm{d}(x_k, x_0)^2 - 2\langle v, \gamma'_k(0)\rangle + \zeta\left(\sqrt{L_R}\mathrm{d}(x_k, x_0)\right)\|v\|^2$$
$$\leq \mathrm{d}(x_k, x_0)^2 - 2\langle v, \gamma'_k(0)\rangle + (1 + \sqrt{L_R}\mathrm{d}(x_k, x_0))\|v\|^2 \qquad \text{(C.1)}$$

where $\zeta(r) := \frac{r}{\tanh(r)}$.

We will consider two cases:
**Case 1:** $\mathrm{d}(x_k, x_0) \leq \frac{1}{\delta\sqrt{L_R}L'_\beta}$.

From (C.1):

$$\mathrm{d}(x_{k+1},x_0)^2$$

$$\leq \mathrm{d}(x_k,x_0)^2 - 2\left\langle \delta\beta(x_k) + \sqrt{\delta}\xi_k(x_k), \gamma_k'(0)\right\rangle + (1+\sqrt{L_R}\mathrm{d}(x_k,x_0))\left\|\delta\beta(x_k) + \sqrt{\delta}\xi_k(x_k)\right\|^2$$

$$\leq \mathrm{d}(x_k,x_0)^2 - 2\left\langle \delta\beta(x_k) + \sqrt{\delta}\xi_k(x_k), \gamma_k'(0)\right\rangle + \delta^2(L_0{}^2 + L_\beta'{}^2\mathrm{d}(x_k,x_0)^2) + \delta\|\xi_k(x_k)\|^2$$

$$\quad + \delta^2\sqrt{L_R}(L_0^2\mathrm{d}(x_k,x_0) + L_\beta'{}^2\mathrm{d}(x_k,x_0)^3) + \delta\sqrt{L_R}\|\xi_k(x_k)\|^2\mathrm{d}(x_k,x_0)$$

$$\leq \mathrm{d}(x_k,x_0)^2 + \delta L_\beta'\mathrm{d}(x_k,x_0)^2 + K\delta^2 L_0^2 + \frac{1}{4K}\mathrm{d}(x_k,x_0)^2 - 2\left\langle \sqrt{\delta}\xi_k(x_k), \gamma_k'(0)\right\rangle$$

$$\quad + \delta^2 L_0^2 + \delta L_\beta'\mathrm{d}(x_k,x_0)^2 + \delta\|\xi_k(x_k)\|^2$$

$$\quad + \delta^2 L_R L_0^2\mathrm{d}(x,x_0)^2 + \delta^2 L_0^2 + \delta L_\beta'\mathrm{d}(x_k,x_0)^2 + \delta L_R\|\xi_k(x_k)\|^2\mathrm{d}(x_k,x_0)^2 + \delta\|\xi_k(x_k)\|^2$$

$$\leq (1 + 3\delta L_\beta' + \delta L_R\|\xi_k(x_k)\|^2 + \frac{1}{4K} + \delta^2 L_R L_0^2)\mathrm{d}(x_k,x_0)^2 - 2\left\langle \sqrt{\delta}\xi_k(x_k), \gamma_k'(0)\right\rangle$$

$$\quad + (2K\delta^2 L_0^2 + \delta\|\xi_k(x_k)\|^2 + \delta\|\xi_k(x_k)\|^2)$$

where the third inequality uses the definition of Case 1, and the fourth inequality is by several applications of Young's Inequality.

**Case 2:** $\mathrm{d}(x_k,x_0) > \frac{1}{4\delta\sqrt{L_R}L_\beta'}$.

Let us define

$$z(t) := \mathrm{Exp}_{x_k}\left(t(\delta\beta(x_k) + \sqrt{\delta}\xi_k(x_k))\right)$$

I.e. $z(t)$ interpolates between $x_k$ and $x_{k+1}$. We verify that $z'(t) = \Gamma_{z(0)}^{z(t)}(\delta\beta(x_k) + \sqrt{\delta}\xi_k(x_k))$. Let us also define a family of geodesics $\gamma_t$, where for each $t$, $\gamma_t$ is a minimizing geodesic with $\gamma_t(0) = z(t)$ and $\gamma_t(1) = x_0$. If such a minimizing geodesic is not unique, any choice will do. We verify that

$$\frac{d}{dt}\mathrm{d}(z(t),x_0)^2 \leq -2\left\langle \gamma_t'(0), z'(t)\right\rangle$$

$$\leq \underbrace{-2\left\langle \delta\beta(z(t)), \gamma_t'(0)\right\rangle}_{\textcircled{1}}$$

$$+ \underbrace{2\left\langle \delta\beta(z(t)) - \Gamma_{z(0)}^{z(t)}(\delta\beta(x_k)), \gamma_t'(0)\right\rangle}_{\textcircled{2}} - \underbrace{2\left\langle \Gamma_{z(0)}^{z(t)}(\sqrt{\delta}\xi_k(x_k)), \gamma_t'(0)\right\rangle}_{\textcircled{3}}.$$

Let's upper bound the terms one by one.

We first bound $\textcircled{2}$, which represents the "discretization error in drift":

$$\textcircled{2} := 2\left\langle \delta\beta(z(t)) - \Gamma_{z(0)}^{z(t)}(\delta\beta(x_k)), \gamma_t'(0)\right\rangle$$

$$\leq 2\left\|\delta\beta(z(t)) - \Gamma_{z(0)}^{z(t)}(\delta\beta(x_k))\right\|\mathrm{d}(z(t),x_0)$$

$$\leq 2\delta L_\beta'\mathrm{d}(z(t),x_k)\mathrm{d}(z(t),x_0)$$

$$\leq 2\delta L_\beta'\mathrm{d}(z(t),x_k)^2 + 2\delta L_\beta'\mathrm{d}(z(t),x_0)^2$$

By definition of $z(t)$, we know that $\mathrm{d}(z(t),x_k) \leq \left\|\delta\beta(x_k) + \sqrt{\delta}\xi_k(x_k)\right\| \leq \delta L_0 + \delta L_\beta'\mathrm{d}(x_k,x_0) + \sqrt{\delta}\|\xi_k(x_k)\|$, so that $2\delta L_\beta'\mathrm{d}(z(t),x_k)^2 \leq 8\delta^3 L_\beta'{}^3\mathrm{d}(x_k,x_0)^2 + 8\delta^2 L_\beta'\|\xi_k(x_k)\|^2 + 8\delta^3 L_\beta'L_0^2 \leq \delta L_\beta'\mathrm{d}(x_k,x_0)^2 + \delta\|\xi_k(x_k)\|^2 + \delta^2 L_0^2$, so that

$$\textcircled{2} \leq \delta L_\beta'\mathrm{d}(z(t),x_k)^2 + \delta L_\beta'\mathrm{d}(x_k,x_0)^2 + \delta\|\xi_k(x_k)\|^2 + \delta^2 L_0^2$$

Next, we bound ③, which is the most significant error term. From the definition of Case 2, $\mathrm{d}(x_k, x_0) > \frac{1}{\delta\sqrt{L_R}L'_\beta}$,

$$
\begin{aligned}
③ &\leq 2\left\langle \Gamma^{z(t)}_{z(0)}(\sqrt{\delta}\xi_k(x_k)), \gamma'_t(0)\right\rangle \\
&\leq 2\sqrt{\delta}\|\xi_k(x_k)\|\mathrm{d}(z(t), x_0) \\
&\leq \delta L'_\beta \mathrm{d}(z(t), x_0)^2 + \frac{1}{L'_\beta}\|\xi_k(x_k)\|^2 \\
&\leq \delta L'_\beta \mathrm{d}(z(t), x_0)^2 + \delta L_R\|\xi_k(x_k)\|^2 \mathrm{d}(x_k, x_0)^2
\end{aligned}
$$

where we use our assumption that $\delta \leq \frac{1}{L'_\beta}$.

Finally, we bound ① as

$$
-2\left\langle \delta\beta(z(t)), \gamma'_t(0)\right\rangle \leq 4\delta L'_\beta \mathrm{d}(z(t), x_0)^2 + 4K\delta^2 L_0^2 + \frac{1}{4K}\mathrm{d}(z(t), x_0)^2
$$

Putting everything together,

$$
\frac{d}{dt}\mathrm{d}(z(t), x_0)^2 \leq (6\delta L'_\beta + \frac{1}{4K})\mathrm{d}(z(t), x_0)^2 + (\delta L'_\beta + \delta L_R\|\xi_k(x_k)\|^2)\mathrm{d}(x_k, x_0)^2 + \delta\|\xi_k(x_k)\|^2 + 4K\delta^2 L_0^2
$$

By Gronwall's Lemma (integrating from $t = 0$ to $t = 1$),

$$
\begin{aligned}
&\mathrm{d}(x_{k+1}, x_0)^2 \\
=&\mathrm{d}(z(1), x_0)^2 \\
\leq& \exp\left(6\delta L'_\beta + \frac{1}{4K}\right)\mathrm{d}(x_k, x_0)^2 + (2\delta L'_\beta + 2\delta L_R\|\xi_k(x_k)\|^2)\mathrm{d}(x_k, x_0)^2 + 2\delta\|\xi_k(x_k)\|^2 + 8K\delta^2 L_0^2 \\
\leq& (1 + 8\delta L'_\beta + \frac{1}{2K} + \delta L_R\|\xi_k(x_k)\|^2)\mathrm{d}(x_k, x_0)^2 + 2\delta\|\xi_k(x_k)\|^2 + 8K\delta^2 L_0^2
\end{aligned}
$$

where we use the assumption that $\delta \leq \frac{1}{8L'_\beta}$.

**Combining Case 1 and Case 2:**

$$
\begin{aligned}
\mathrm{d}(x_{k+1}, x_0)^2 \leq& (1 + 8\delta L'_\beta + \frac{1}{2K} + \delta L_R\|\xi_k(x_k)\|^2 + \delta^2 L_R L_0^2)\mathrm{d}(x_k, x_0)^2 + 2\delta\|\xi_k(x_k)\|^2 + 8K\delta^2 L_0^2 \\
&+ \mathbb{1}\left\{\mathrm{d}(x_k, x_0) \leq \frac{1}{\delta\sqrt{L_R}L'_\beta}\right\}(-2\left\langle\sqrt{\delta}\xi_k(x_k), \gamma'_k(0)\right\rangle)
\end{aligned}
$$

$\square$

### C.1.2   Under Dissipativity

**Lemma 14** (One-step distance evolution under Dissipativity). *Let $\beta$ be a vector field satisfying 3. Let $\delta \in \mathbb{R}^+$ be a stepsize satisfying $\delta \leq \frac{m}{128L'_\beta{}^2}$. Let $x^*$ be some point with $\beta(x^*) = 0$. Let $x_0 \in M$ be arbitrary. Let $x_k$ be the iterative process*

$$
x_{k+1} = \mathrm{Exp}_{x_k}\left(\delta\beta(x_k) + \sqrt{\delta}\xi_k(x_k)\right)
$$

*Assume that for all $x$ such that $\mathrm{d}(x, x^*) \geq R$, there exists a minimizing geodesic $\gamma : [0, 1] \to M$ with $\gamma(0) = x, \gamma(1) = x^*$, and*

$$
\left\langle\beta(x), \gamma'(0)\right\rangle \leq -m\mathrm{d}(x, x^*)^2,
$$

*and let $\gamma_k$ denote such a geodesic for $x = x_k$. Then for any $k$,*

$$
\begin{aligned}
\mathrm{d}(x_{k+1}, x^*)^2 \leq& (1 - \delta m)\mathrm{d}(x_k, x^*)^2 + \frac{2048\delta L_R L'_\beta{}^4}{m^5}\|\xi_k(x_k)\|^4 + 4\delta L'_\beta R^2 \\
&+ \mathbb{1}\left\{\mathrm{d}(x_k, x^*) \leq \frac{m}{4\delta\sqrt{L_R}L'_\beta{}^2}\right\}(-2\left\langle\sqrt{\delta}\xi_k(x_k), \gamma'_k(0)\right\rangle)
\end{aligned}
$$

*Proof.* Throughout the proof, it is useful to note that by our assumptions, it must be that $m \leq L'_\beta$. We will be using the bound from Zhang and Sra [2016] (see Lemma 25). Let $v := \delta\beta(x_k) + \sqrt{\delta}\xi_k(x_k)$. Then Lemma 25 bounds

$$
\begin{aligned}
\mathrm{d}(x_{k+1}, x)^2 &\leq \mathrm{d}(x_k, x^*)^2 - 2\langle v, \gamma'_k(0)\rangle + \zeta\left(\sqrt{L_R}\mathrm{d}(x_k, x^*)\right)\|v\|^2 \\
&\leq \mathrm{d}(x_k, x^*)^2 - 2\langle v, \gamma'_k(0)\rangle + (1 + \sqrt{L_R}\mathrm{d}(x_k, x^*))\|v\|^2
\end{aligned} \tag{C.2}
$$

where $\zeta(r) := \frac{r}{\tanh(r)}$.

We will consider two cases:

**Case 1:** $\mathrm{d}(x_k, x^*) \leq \frac{m}{4\delta\sqrt{L_R}{L'_\beta}^2}$.

From (C.2):

$$
\begin{aligned}
&\mathrm{d}^2(x_{k+1}, x^*)^2 \\
&\leq \mathrm{d}^2(x_k, x^*) - 2\left\langle \delta\beta(x_k) + \sqrt{\delta}\xi_k(x_k), \gamma'_k(0)\right\rangle + (1 + \sqrt{L_R}\mathrm{d}(x_k, x^*))\left\|\delta\beta(x_k) + \sqrt{\delta}\xi_k(x_k)\right\|^2 \\
&\leq \mathrm{d}^2(x_k, x^*) - 2\left\langle \delta\beta(x_k) + \sqrt{\delta}\xi_k(x_k), \gamma'_k(0)\right\rangle + \delta^2{L'_\beta}^2\mathrm{d}(x_k, x^*)^2 + \delta\|\xi_k(x_k)\|^2 \\
&\quad + \delta^2\sqrt{L_R}{L'_\beta}^2\mathrm{d}(x_k, x^*)^3 + \delta\sqrt{L_R}\|\xi_k(x_k)\|^2\mathrm{d}(x_k, x^*) \\
&\leq (1 + \delta m/2)\mathrm{d}^2(x_k, x^*) - 2\left\langle \delta\beta(x_k) + \sqrt{\delta}\xi_k(x_k), \gamma'_k(0)\right\rangle + \frac{4\delta L_R}{m}\|\xi_k(x_k)\|^4
\end{aligned} \tag{C.3}
$$

where we use our assumptions that $\delta \leq m/(16{L'_\beta}^2)$ and the inequality under Case 1. We used Cauchy Schwarz a few times.

We can further bound

$$
\begin{aligned}
2\langle \delta\beta(x_k), \gamma'_k(0)\rangle &\leq \mathbb{1}\{\mathrm{d}(x_k, x^*) \geq \mathcal{R}\}(-2m\mathrm{d}(x_k, x^*)^2) + \mathbb{1}\{\mathrm{d}(x_k, x^*) \leq \mathcal{R}\}(2L'_\beta\mathrm{d}(x_k, x^*)^2) \\
&\leq -2\delta m\mathrm{d}(x_k, x^*)^2 + 2\delta(m + L'_\beta)\mathcal{R}^2
\end{aligned}
$$

Thus

$$
\begin{aligned}
\mathrm{d}^2(x_{k+1}, x^*)^2 &\leq (1 - \delta m)\mathrm{d}^2(x_k, x^*) - 2\left\langle \sqrt{\delta}\xi_k(x_k), \gamma'_k(0)\right\rangle \\
&\quad + 2\delta(m + L'_\beta)\mathcal{R}^2 + \frac{4\delta L_R}{m}\|\xi_k(x_k)\|^4
\end{aligned} \tag{C.4}
$$

where we use the fact that $e^{\delta m} \leq e^{\frac{m^2}{16{L'_\beta}^2}} \leq 2$.

**Case 2:** $\mathrm{d}(x_k, x^*) > \frac{m}{4\delta\sqrt{L_R}{L'_\beta}^2}$.

Let us define

$$
z(t) := \mathrm{Exp}_{x_k}\left(t(\delta\beta(x_k) + \sqrt{\delta}\xi_k(x_k))\right)
$$

I.e. $z(t)$ interpolates between $x_k$ and $x_{k+1}$. We verify that $z'(t) = \Gamma_{z(0)}^{z(t)}(\delta\beta(x_k) + \sqrt{\delta}\xi_k(x_k))$. Let us also define a family of geodesics $\gamma_t$, where for each $t$, $\gamma_t$ is a minimizing geodesic with $\gamma_t(0) = z(t)$ and $\gamma_t(1) = x^*$. If such a minimizing geodesic is not unique, any choice will do. We also verify that

$$
\begin{aligned}
\frac{d}{dt}\mathrm{d}(z(t), x^*)^2 &\leq -2\langle \gamma'_t(0), z'(t)\rangle \\
&\leq \underbrace{-2\langle \delta\beta(z(t)), \gamma'_t(0)\rangle}_{①} \\
&\quad + \underbrace{2\left\langle \delta\beta(z(t)) - \Gamma_{z(0)}^{z(t)}(\delta\beta(x_k)), \gamma'_t(0)\right\rangle}_{②} - \underbrace{2\left\langle \Gamma_{z(0)}^{z(t)}(\sqrt{\delta}\xi_k(x_k)), \gamma'_t(0)\right\rangle}_{③}
\end{aligned}
$$

Let's upper bound the terms one by one.

We first bound ②, which represents the "discretization error in drift":

$$② := 2\left\langle \delta\beta(z(t)) - \Gamma_{z(0)}^{z(t)}(\delta\beta(x_k)), \gamma_t'(0) \right\rangle$$

$$\leq 2\left\| \delta\beta(z(t)) - \Gamma_{z(0)}^{z(t)}(\delta\beta(x_k)) \right\| \mathrm{d}(z(t), x^*)$$

$$\leq 2\delta L_\beta' \mathrm{d}(z(t), x_k)\mathrm{d}(z(t), x^*)$$

$$\leq \frac{\delta m}{4}\mathrm{d}(z(t), x^*)^2 + \frac{4\delta L_\beta'^2}{m}\mathrm{d}(z(t), x_k)^2$$

By definition of $z(t)$, we know that $\mathrm{d}(z(t), x_k) \leq \left\| \delta\beta(x_k) + \sqrt{\delta}\xi_k(x_k) \right\| \leq \delta L_\beta'\mathrm{d}(x_k, x^*) + \sqrt{\delta}\|\xi_k(x_k)\|$, so that $\frac{4\delta L_\beta'^2}{m}\mathrm{d}(z(t), x_k)^2 \leq \frac{4\delta^3 L_\beta'^4}{m}\mathrm{d}(x_k, x^*)^2 + \frac{4\delta^2 L_\beta'^2}{m}\|\xi_k(x_k)\|^2 \leq \frac{\delta m}{8}\mathrm{d}(x_k, x^*)^2 + \delta\|\xi_k(x_k)\|^2$, so that

$$② \leq \frac{\delta m}{4}\mathrm{d}(z(t), x^*)^2 + \frac{\delta m}{8}\mathrm{d}(x_k, x^*)^2 + \delta\|\xi_k(x_k)\|^2$$

Next, we bound ③, which is the most significant error term. From the definition of Case 2, $\mathrm{d}(x_k, x^*) > \frac{1}{4\delta\sqrt{L_R L_\beta'}}$,

$$③ \leq 2\left\langle \Gamma_{z(0)}^{z(t)}(\sqrt{\delta}\xi_k(x_k)), \gamma_t'(0) \right\rangle$$

$$\leq 2\sqrt{\delta}\|\xi_k(x_k)\|\mathrm{d}(z(t), x^*)$$

$$\leq \frac{\delta m}{8}\mathrm{d}(z(t), x^*)^2 + \frac{8}{m}\|\xi_k(x_k)\|^2$$

$$\leq \frac{\delta m}{8}\mathrm{d}(z(t), x^*)^2 + \frac{32\delta\sqrt{L_R}L_\beta'^2}{m^2}\|\xi_k(x_k)\|^2\mathrm{d}(x_k, x^*)$$

$$\leq \frac{\delta m}{8}\mathrm{d}(z(t), x^*)^2 + \frac{\delta m}{8}\mathrm{d}(x_k, x^*)^2 + \frac{2048\delta L_R L_\beta'^4}{m^5}\|\xi_k(x_k)\|^4$$

where we use our assumption that $\delta \leq \frac{m}{128 L_\beta'^2}$.

Finally, we bound ① as

$$-2\left\langle \delta\beta(z(t)), \gamma_t'(0) \right\rangle$$

$$\leq \mathbb{1}\left\{ \mathrm{d}(z(t), x^*)^2 \geq \mathcal{R} \right\}(-2\delta m\mathrm{d}(z(t), x^*)^2) + \mathbb{1}\left\{ \mathrm{d}(z(t), x^*) \leq \mathcal{R} \right\}(2\delta L_\beta'\mathrm{d}(z(t)x^*))$$

$$\leq 4\delta L_\beta'\mathcal{R}^2 - 2\delta m\mathrm{d}(\mathrm{d}(z(t), x^*)^2)$$

Putting everything together,

$$\frac{d}{dt}\mathrm{d}(z(t), x^*)^2 \leq -\frac{3\delta m}{2}\mathrm{d}(z(t), x^*)^2 + \frac{\delta m}{4}\mathrm{d}(x_k, x^*)^2 + \frac{2048\delta L_R L_\beta'^4}{m^5}\|\xi_k(x_k)\|^4 + 4\delta L_\beta'\mathcal{R}^2$$

By Gronwall's Lemma (integrating from $t = 0$ to $t = 1$),

$$\mathrm{d}(x_{k+1}, x^*)^2 = \mathrm{d}(z(1), x^*)^2$$

$$\leq \exp\left(-3\delta m/2\right)\mathrm{d}(x_k, x^*)^2 + \frac{\delta m}{4}\mathrm{d}(x_k, x^*)^2 + \frac{2048\delta L_R L_\beta'^4}{m^5}\|\xi_k(x_k)\|^4 + 4\delta L_\beta'\mathcal{R}^2$$

$$\leq (1 - \delta m)\mathrm{d}(x_k, x^*)^2 + \frac{2048\delta L_R L_\beta'^4}{m^5}\|\xi_k(x_k)\|^4 + 4\delta L_\beta'\mathcal{R}^2 \tag{C.5}$$

where we use the assumption that $\delta \leq \frac{1}{128m}$ so that $\exp\left(-3\delta m/2\right) \leq 1 - 5\delta m/4$.

**Combining Case 1 and Case 2:**

Combining (C.4) and (C.5),

$$d(x_{k+1}, x^*)^2 \leq (1 - \delta m)d(x_k, x^*)^2 + \left(\frac{2048\delta L_R {L'_\beta}^4}{m^5}\|\xi_k(x_k)\|^4 + 4\delta L'_\beta \mathcal{R}^2\right)$$
$$+ \mathbb{1}\left\{d(x_k, x^*) \leq \frac{1}{4\delta\sqrt{L_R}L'_\beta}\right\}\left(-2\left\langle\sqrt{\delta}\xi_k(x_k), \gamma'_k(0)\right\rangle\right)$$

$\square$

## C.2 $L_p$ Bounds

### C.2.1 Under Lipschitz Continuity

**Lemma 15** (L2 Bound and Chevyshev under Lipschitz Continuity). *Consider the same setup as Lemma 13. Assume in addition that there exists $\sigma_\xi \in \mathbb{R}^+$ such that for all $x$ and for all $k$, $\mathbb{E}\left[\|\xi_k(x)\|^2\right] \leq \sigma_\xi^2$. Then for any positive integer $K$, and for all $k \leq K$,*

$$\mathbb{E}\left[d(x_k, x_0)^2\right] \leq 4\exp\left(8K\delta L'_\beta + K\delta L_R\sigma_\xi^2 + K\delta^2 L_R L_0^2\right) \cdot (2K\delta\sigma_\xi^2 + 8K^2\delta^2 L_0^2)$$

*and*

$$\mathbb{P}(\max_{k \leq K} d(x_k, x_0) \geq s) \leq \frac{4}{s^2}\exp\left(8K\delta L'_\beta + K\delta L_R\sigma_\xi^2 + K\delta^2 L_R L_0^2\right) \cdot (2K\delta\sigma_\xi^2 + 8K^2\delta^2 L_0^2)$$

*Proof.* Let $\mathcal{F}_k$ denote the $\sigma$-field generated by $\xi_0...\xi_{k-1}$.

To bound the first claim, take expectation of the bound from Lemma 13 wrt $\mathcal{F}_k$:

$$\mathbb{E}_{\mathcal{F}_k}\left[d(x_{k+1}, x_0)^2\right]$$
$$\leq \mathbb{E}_{\mathcal{F}_k}\left[\left(1 + 8\delta L'_\beta + \frac{1}{2K} + \delta L_R\|\xi_k(x_k)\|^2 + \delta^2 L_R L_0^2\right)d(x_k, x_0)^2\right] + 2\delta\mathbb{E}_{\mathcal{F}_k}\left[\|\xi_k(x_k)\|^2\right] + 8K\delta^2 L_0^2$$
$$+ \mathbb{E}_{\mathcal{F}_k}\left[\mathbb{1}\left\{d(x_k, x_0) \leq \frac{1}{\delta\sqrt{L_R}L'_\beta}\right\}\left(-2\left\langle\sqrt{\delta}\xi_k(x_k), \gamma'_k(0)\right\rangle\right)\right]$$
$$\leq \left(1 + 8\delta L'_\beta + \frac{1}{2K} + \delta L_R\sigma_\xi^2 + \delta^2 L_R L_0^2\right)d(x_k, x_0)^2 + 2\delta\sigma_\xi^2 + 8K\delta^2 L_0^2$$
$$\leq \exp\left(8\delta L'_\beta + \frac{1}{2K} + \delta L_R\sigma_\xi^2 + \delta^2 L_R L_0^2\right)d(x_k, x_0)^2 + 2\delta\sigma_\xi^2 + 8K\delta^2 L_0^2$$

In line 3 above, $\gamma_k(t)$ is a minimizing geodesic from $x_k$ to $x_0$, as defined in Lemma 13.

Applying the above recursively,

$$\mathbb{E}\left[d(x_K, x_0)^2\right] \leq \exp\left(1 + 8K\delta L'_\beta + K\delta L_R\sigma_\xi^2 + K\delta^2 L_R L_0^2\right) \cdot (2K\delta\sigma_\xi^2 + 8K^2\delta^2 L_0^2)$$

The above upper bound clearly also holds for $\mathbb{E}\left[d(x_k, x_0)^2\right]$ for all $k \leq K$. This proves our first claim.

To prove the second claim, let us define

$$r_0^2 := 0$$
$$r_{k+1}^2 := \left(1 + 8\delta L'_\beta + \frac{1}{4K} + \delta L_R\|\xi_k(x_k)\|^2 + \delta^2 L_R L_0^2\right)r_k^2 + 2\delta\|\xi_k(x_k)\|^2 + 8K\delta^2 L_0^2$$
$$+ \mathbb{1}\left\{d(x_k, x_0) \leq \frac{1}{\delta\sqrt{L_R}L'_\beta}\right\}\left(-2\left\langle\sqrt{\delta}\xi_k(x_k), \gamma'_k(0)\right\rangle\right)$$

We verify that $r_k$ as defined above is a sub-martingale. Thus by Doob's martingale inequality,

$$\mathbb{P}(\max_{k \leq K} r_k^2 \geq s) \leq \frac{\mathbb{E}\left[r_K^2\right]}{s}$$

Furthermore, notice that

$$r_0^2 = \mathrm{d}(x_0, x_0)^2 = 0$$

$$r_{k+1}^2 - \mathrm{d}(x_{k+1}^2, x_0)^2 \geq (1 + 8\delta L_\beta' + \frac{2}{K} + \delta L_R \|\xi_k(x_k)\|^2 + \delta^2 L_R L_0^2)(r_k^2 - \mathrm{d}(x_k, x_0)^2) \geq 0$$

so that $r_k \geq \mathrm{d}(x_k, x_0)$ with probability 1, for all $k$.

Thus

$$\mathbb{P}(\max_{k \leq K} \mathrm{d}(x_k, x_0) \geq s) \leq \mathbb{P}(\max_{k \leq K} r_k^2 \geq s^2) \leq \frac{\mathbb{E}\left[r_K^2\right]}{s^2}$$

$$\leq \frac{1}{s^2} \exp\left(1 + 8K\delta L_\beta' + K\delta L_R \sigma_\xi^2 + K\delta^2 L_R L_0^2\right) \cdot (2K\delta\sigma_\xi^2 + 8K^2\delta^2 L_0^2)$$

The proof for the bound on $r_K^2$ is identical to the proof of the first claim. We conclude our proof of the second claim $\qquad\square$

**Lemma 16** (L4 Bound and Chevyshev under Lipschitz Continuity). *Let $\beta$ be a vector field satisfying 3. Assume in addition that $\delta \in \mathbb{R}^+$ satisfies $\delta \leq \min\left\{\frac{1}{16L_\beta'}, \frac{1}{16\sqrt{L_R L_0}}, \frac{1}{L_R d}\right\}$. Let $L_0 := \|\beta(x_0)\|$. Let $x_k$ be the following stochastic process:*

$$x_{k+1} = \mathrm{Exp}_{x_k}\left(\delta\beta(x_k) + \sqrt{\delta}\xi_k(x_k)\right)$$

*Assume in addition that for all $x$ and for all $k$, $\mathbb{E}\left[\|\xi_k(x)\|^4\right] \leq 2d^2$. Then for any positive $K \geq 4$, and for all $k \leq K$,*

$$\mathbb{E}\left[\mathrm{d}(x_k, x_0)^4\right] \leq \exp\left(2 + 16K\delta L_\beta' + 4K\delta L_R d + 3K\delta^2 L_R L_0^2\right)(5K^2\delta^2 d^2 + 64K^4\delta^4 L_0^4)$$

*and*

$$\mathbb{P}(\max_{k \leq K} \mathrm{d}(x_k, x_0) \geq s) \leq \frac{1}{s^4} \exp\left(2 + 16K\delta L_\beta' + 4K\delta L_R d + 3K\delta^2 L_R L_0^2\right)(5K^2\delta^2 d^2 + 64K^4\delta^4 L_0^4)$$

*Proof.* Let $\mathcal{F}_k$ denote the $\sigma$-field generated by $\xi_0...\xi_{k-1}$.

We will use the following inequality from Lemma 13:

$$\mathrm{d}(x_{k+1}, x_0)^2 \leq (1 + 8\delta L_\beta' + \frac{1}{2K} + \delta L_R \|\xi_k(x_k)\|^2 + \delta^2 L_R L_0^2)\mathrm{d}(x_k, x_0)^2 + 2\delta\|\xi_k(x_k)\|^2 + 8K\delta^2 L_0^2$$

$$+ \mathbb{1}\left\{\mathrm{d}(x_k, x_0) \leq \frac{1}{\delta\sqrt{L_R L_\beta'}}\right\}(-2\left\langle\sqrt{\delta}\xi_k(x_k), \gamma_k'(0)\right\rangle)$$

Squaring both sides,

$$\mathrm{d}(x_{k+1}, x_0)^4$$
$$\leq (1 + 16\delta L_\beta' + 3\delta L_R \|\xi_k(x_k)\|^2 + 3\delta^2 L_R L_0^2 + 2\delta^2 L_R^2\|\xi_k(x_k)\|^4 + \frac{2}{K})\mathrm{d}(x_k, x_0)^4$$
$$+ 5K\delta^2\|\xi_k(x_k)\|^4 + 64K^3\delta^4 L_0^4$$
$$+ \circledast \qquad\qquad\qquad\qquad\qquad\qquad\qquad\qquad\qquad\qquad\qquad\qquad\text{(C.6)}$$

where $\circledast$ has 0-mean, and we used a few times Cauchy Schwarz and Young's Inequality.

Taking expectation wrt $\mathcal{F}_k$,

$$\mathbb{E}_{\mathcal{F}_k}\left[\mathrm{d}(x_{k+1}, x_0)^4\right]$$
$$\leq (1 + 16\delta L_\beta' + 3\delta L_R d + 3\delta^2 L_R L_0^2 + 2\delta^2 L_R^2 d^2 + \frac{2}{K})\mathrm{d}(x_k, x_0)^4 + 5K\delta^2 d^2 + 64K^3\delta^4 L_0^4$$
$$\leq (1 + 16\delta L_\beta' + 4\delta L_R d + 3\delta^2 L_R L_0^2 + \frac{2}{K})\mathrm{d}(x_k, x_0)^4 + 5K\delta^2 d^2 + 64K^3\delta^4 L_0^4$$

Applying the above recursively,

$$\mathbb{E}\left[\mathrm{d}(x_K, x_0)^4\right] \leq \exp\left(2 + 16K\delta L'_\beta + 4K\delta L_R d + 3K\delta^2 L_R L_0^2\right)(5K^2\delta^2 d^2 + 64K^4\delta^4 L_0^4)$$

To prove the second claim, define

$$r_0^4 := 0$$

$$r_{k+1}^4 := (1 + 16\delta L'_\beta + 3\delta L_R\|\xi_k(x_k)\|^2 + 3\delta^2 L_R L_0^2 + 2\delta^2 L_R^2\|\xi_k(x_k)\|^4 + \frac{2}{K})r_k^4$$
$$+ 5K\delta^2\|\xi_k(x_k)\|^4 + 64K^3\delta^4 L_0^4$$
$$+ \circledast$$

where $\circledast$ is the same term as (C.6).

We verify that $r_k$ as defined above is a sub-martingale. Thus by Doob's martingale inequality,

$$\mathbb{P}(\max_{k \leq K} r_k^2 \geq s) \leq \frac{\mathbb{E}\left[r_K^2\right]}{s}$$

Furthermore, notice that

$$r_0^2 = \mathrm{d}(x_0, x_0)^2 = 0$$

$$r_{k+1}^2 - \mathrm{d}(x_{k+1}^2, x_0)^2 \geq (1 + 16\delta L'_\beta + 3\delta L_R\|\xi_k(x_k)\|^2 + 3\delta^2 L_R L_0^2 + 2\delta^2 L_R^2\|\xi_k(x_k)\|^4 + \frac{2}{K})(r_k^2 - \mathrm{d}(x_k, x_0)^2)$$
$$\geq 0$$

so that $r_k \geq \mathrm{d}(x_k, x_0)$ with probability 1, for all $k$.

Thus

$$\mathbb{P}(\max_{k \leq K} \mathrm{d}(x_k, x_0) \geq s) \leq \mathbb{P}(\max_{k \leq K} r_k \geq s) \leq \frac{\mathbb{E}\left[r_K^4\right]}{s^4}$$
$$\leq \frac{1}{s^4}\exp\left(2 + 16K\delta L'_\beta + 4K\delta L_R d + 3K\delta^2 L_R L_0^2\right)(5K^2\delta^2 d^2 + 64K^4\delta^4 L_0^4)$$

The proof for the bound on $r_K^2$ is identical to the proof of the first claim. We conclude our proof of the second claim $\qquad\square$

### C.2.2 Under Dissipativity

**Lemma 17** (L4 Bound under Dissipativity, Discretized SDE). *Let $\beta$ be a vector field satisfying 3. Let $x^*$ be some point with $\beta(x^*) = 0$. Assume that for all $x$ such that $\mathrm{d}(x, x^*) \geq \mathcal{R}$, there exists a minimizing geodesic $\gamma : [0, 1] \to M$ with $\gamma(0) = x, \gamma(1) = x^*$, and*

$$\langle \beta(x), \gamma'(0) \rangle \leq -m\mathrm{d}(x, x^*)^2$$

*. Assume in addition that $\delta \in \mathbb{R}^+$ satisfies $\delta \leq \frac{m}{128L'^2_\beta}$ Let $x_k$ be the following stochastic process:*

$$x_{k+1} = \mathrm{Exp}_{x_k}\left(\delta\beta(x_k) + \sqrt{\delta}\xi_k(x_k)\right)$$

*where $\xi_k$ is a random vector field satisfying $\mathbb{E}\left[\xi_k(x)\right] = 0$ and $\mathbb{E}\left[\|\xi_k\|^4\right] \leq (\sigma_\xi)^4$.*

*For any $k$,*

$$\mathbb{E}\left[\mathrm{d}(x_K, x^*)^4\right] \leq e^{-K\delta m}\mathbb{E}\left[\mathrm{d}(x_0, x^*)^4\right] + \frac{2^{24}L_R^2 L'^8_\beta}{m^{12}}\sigma_\xi^8 + \frac{64L'^2_\beta\mathcal{R}^4}{m^2} + \frac{128}{m^2}\sigma_\xi^4$$

*Proof.* Let $\gamma_k$ denote a geodesic with $\gamma_k(0) = x_k, \gamma(1) = x^*$, and

$$\langle \beta(x_k), \gamma'_k(0) \rangle \leq -m\mathrm{d}(x_k, x^*)^2$$

From Lemma 14,

$$\mathrm{d}(x_{k+1}, x^*)^2 \leq (1 - \delta m)\mathrm{d}(x_k, x^*)^2 + \frac{2048\delta L_R {L_\beta'}^4}{m^5}\|\xi_k(x_k)\|^4 + 4\delta L_\beta' \mathcal{R}^2$$

$$+ \mathbb{1}\left\{\mathrm{d}(x_k, x^*) \leq \frac{m}{4\delta\sqrt{L_R}{L_\beta'}^2}\right\}(-2\left\langle\sqrt{\delta}\xi_k(x_k), \gamma_k'(0)\right\rangle)$$

Squaring both sides and taking expectation wrt $\xi_k$ (and applying Young's Inequality),

$$\mathbb{E}\left[\mathrm{d}(x_{k+1}, x^*)^4\right]$$

$$\leq (1 - \frac{3\delta m}{2})\mathrm{d}(x_k, x^*)^4 + \frac{2^{24}\delta L_R^2 {L_\beta'}^8}{m^{11}}\mathbb{E}\left[\|\xi_k(x_k)\|^8\right] + \frac{64\delta {L_\beta'}^2 \mathcal{R}^4}{m}$$

$$+ \frac{\delta m}{2}\mathrm{d}(x_k, x^*)^4 + \frac{128\delta}{m}\mathbb{E}\left[\|\xi_k(x_k)\|^4\right]$$

$$\leq (1 - \delta m)\mathrm{d}(x_k, x^*)^4 + \frac{2^{24}\delta L_R^2 {L_\beta'}^8}{m^{11}}\mathbb{E}\left[\|\xi_k(x_k)\|^8\right] + \frac{64\delta {L_\beta'}^2 \mathcal{R}^4}{m} + \frac{128\delta}{m}\mathbb{E}\left[\|\xi_k(x_k)\|^4\right]$$

Applying the above recursively,

$$\mathbb{E}\left[\mathrm{d}(x_K, x^*)^4\right] \leq e^{-K\delta m}\mathbb{E}\left[\mathrm{d}(x_0, x^*)^4\right] + \frac{2^{24}L_R^2 {L_\beta'}^8}{m^{12}}\sigma_\xi^8 + \frac{64{L_\beta'}^2\mathcal{R}^4}{m^2} + \frac{128}{m^2}\sigma_\xi^4$$

$\square$

**Lemma 18** (L4 Bound under Dissipativity, Exact SDE). *Let $\beta$ be a vector field satisfying 3. Let $x^*$ be some point with $\beta(x^*) = 0$. Assume that for all $x$ such that $\mathrm{d}(x, x^*) \geq \mathcal{R}$, there exists a minimizing geodesic $\gamma : [0, 1] \to M$ with $\gamma(0) = x, \gamma(1) = x^*$, and*

$$\langle\beta(x), \gamma'(0)\rangle \leq -m\mathrm{d}(x, x^*)^2$$

*. Let $x(t)$ denote the solution to (1.2). For any $k$,*

$$\mathbb{E}\left[\mathrm{d}(x(T), x^*)^4\right] \leq \exp\left(-K\delta m\right)\mathbb{E}\left[\mathrm{d}(x_0^i, x^*)^4\right] + \frac{2^{26}L_R^2 {L_\beta'}^8}{m^{12}}d^4 + \frac{64{L_\beta'}^2\mathcal{R}^4}{m^2} + \frac{256}{m^2}d^2$$

*Proof.* For $i \in \mathbb{Z}^+$, let $\delta^i$ be a sequence of stepsizes going to 0, let $K^i := T/\delta^i$, and let $x_k^i$ be a discretization of $x(t)$ with stepsize $\delta^i$ of the form (1.2), i.e.

$$x_{k+1}^i = \mathrm{Exp}_{x_k^i}\left(\delta^i\beta(x_k^i) + \sqrt{\delta^i}\zeta_k\right)$$

where $\zeta_k \sim \mathcal{N}_{x_k^i}(0, I)$. Applying Lemma 17 to $x_k^i$ with $\xi_k(x_k) = \zeta_k$ and $\sigma_\xi = \sqrt{2}d$, and for $i$ sufficiently large, gives the bound

$$\mathbb{E}\left[\mathrm{d}(x_{K^i}^i, x^*)^4\right] \leq \exp\left(-K^i\delta^i m\right)\mathbb{E}\left[\mathrm{d}(x_0^i, x^*)^4\right] + \frac{2^{26}L_R^2 {L_\beta'}^8}{m^{12}}d^4 + \frac{64{L_\beta'}^2\mathcal{R}^4}{m^2} + \frac{256}{m^2}d^2$$

Our conclusion follows by Lemma 2 as $x_{K^i}^i$ converges to $x(T)$ almost surely as $i \to \infty$. $\square$

## C.3 Subgaussian Bounds

### C.3.1 Under Dissipativity

**Lemma 19** (Subgaussian Bound under Dissipativity, Discrete Time Semimartingale, Adaptive Stepsize). *Let $\beta$ be a vector field satisfying 3. Let $x^*$ be some point with $\beta(x^*) = 0$. Assume that for all $x$ such that $\mathrm{d}(x, x^*) \geq \mathcal{R}$, there exists a minimizing geodesic $\gamma : [0, 1] \to M$ with $\gamma(0) = x, \gamma(1) = x^*$, and*

$$\langle\beta(x), \gamma'(0)\rangle \leq -m\mathrm{d}(x, x^*)^2.$$

*Let $x_k$ be the following stochastic process:*

$$x_{k+1} = \text{Exp}_{x_k} \left( \delta_k \beta(x_k) + \sqrt{\delta_k} \xi_k(x_k) \right)$$

*where $\xi_k$ is a random vector field. Assume that for all $x$, $\mathbb{E}\left[\xi_k(x)\right] = 0$, and that for any $\rho \leq \frac{1}{8}$, $\mathbb{E}\left[\exp\left(\rho\|\xi_k(x)\|^2\right)\right] \leq \exp\left(\rho\sigma_\xi^2\right)$. For each $k$, $\delta_k$ is a positive stepsize that depends only on $x_k$ and satisfies $\delta_k \leq \min\left\{ \frac{m}{16L_\beta'^2(1+\sqrt{L_R}d(x_k,x^*))}, \frac{\sigma_\xi^2}{m(1+\sqrt{L_R}d(x_k,x^*))}, \frac{32\sigma_\xi^4}{m^2 d(x_k,x^*)^2} \right\}$. Assume that $d(x_0, x^*) \leq 2\mathcal{R}$. Finally, assume that there exists $\delta \in \mathbb{R}^+$ such that for all $k$, $\delta_k \leq \delta$. Then*

$$\mathbb{P}(\max_{i \leq K} d(x_k, x^*) \geq t) \leq 32K\delta\lambda \exp\left( \frac{2L_\beta'\mathcal{R}^2}{\sigma_\xi^2} + \frac{16L_R\sigma_\xi^2}{m} - \frac{mt^2}{64\sigma_\xi^2} \right)$$

*Proof.* For each $k$, let $\gamma_k$ be a minimizing geodesic with $\gamma_k(0) = x_k$ and $\gamma_k(1) = x^*$ satisfying $\langle \beta(x_k), \gamma_k'(0) \rangle \leq -md(x_k, x^*)^2$.

Using the result from Corollary 8 of Zhang and Sra [2016] (see Lemma 25),

$$d(x_{k+1}, x^*)^2 \leq d^2(x_k, x^*) - 2\left\langle \delta_k \beta(x_k) + \sqrt{\delta_k}\xi_k(x_k), \gamma_k'(0) \right\rangle$$
$$+ (1 + \sqrt{L_R}d(x_k, x^*))\left\| \delta_k \beta(x_k) + \sqrt{\delta_k}\xi_k(x_k) \right\|^2$$

By Assumption 2 and Assumption 3, $\langle \beta(x_k), \gamma_k'(0) \rangle \leq -md(x_k, x^*)^2 + 2L_\beta'\mathcal{R}^2$. Applying Cauchy Schwarz and simplifying,

$$d(x_{k+1}, x^*)^2$$
$$\leq (-2m\delta_k + 2\delta_k^2 L_\beta'^2 + 2\delta_k^2\sqrt{L_R}L_\beta'^2 d(x_k, x^*))d(x_k, x^*)^2 + 4\delta_k L_\beta'\mathcal{R}^2$$
$$+ 2\sqrt{\delta_k}\left\langle \xi_k(x_k), \gamma_k'(0) \right\rangle + 2\delta_k(1 + \sqrt{L_R}d(x_k, x^*))\|\xi_k(x_k)\|^2$$
$$\leq -\delta_k m d(x_k, x^*)^2 + 4\delta_k L_\beta'\mathcal{R}^2 + 2\sqrt{\delta_k}\left\langle \xi_k(x_k), \gamma_k'(0) \right\rangle + 2\delta_k(1 + \sqrt{L_R}d(x_k, x^*))\|\xi_k(x_k)\|^2$$

where we used our assumption that $\delta_k \leq \frac{m}{16L_\beta'^2(1+\sqrt{L_R}d(x_k,x^*))}$.

Let $s := \frac{m}{32\sigma_\xi^2}$ We will now apply Lemma 24 with

$$q_k = sd(x_k, x^*)^2 \qquad \nu_k = s\left\langle \xi_k(x_k), \gamma_k'(0) \right\rangle + 2\sqrt{\delta_k}s(1 + \sqrt{L_R}d(x_k, x^*))\|\xi_k(x_k)\|^2$$
$$\lambda = m \qquad \gamma = 4sL_\beta'\mathcal{R}^2 \qquad \mu = 2s\sigma_\xi^2 + \frac{64sL_R\sigma_\xi^4}{m} \qquad \text{(C.7)}$$

We will verify Lemma 24's condition regarding $\mathbb{E}\left[\exp\left(\sqrt{\delta_k}\nu_k\right)\right]$. Taking expectation conditioned on $\xi_0(x_0)...\xi_{k-1}(x_{k-1})$,

$$\mathbb{E}\left[\exp\left(\sqrt{\delta_k}\nu_k\right)\right]$$
$$= \mathbb{E}\left[\exp\left(\sqrt{\delta_k}s\left\langle \xi_k(x_k), \gamma_k'(0) \right\rangle + 2\delta_k s(1 + \sqrt{L_R}d(x_k, x^*))\|\xi_k(x_k)\|^2\right)\right]$$
$$\leq \mathbb{E}\left[\exp\left(\sqrt{\delta_k}2s\left\langle \xi_k(x_k), \gamma_k'(0) \right\rangle\right)\right]^{1/2} \cdot \mathbb{E}\left[\exp\left(4\delta_k s(1 + \sqrt{L_R}d(x_k, x^*))\|\xi_k(x_k)\|^2\right)\right]^{1/2}$$
$$\text{(C.8)}$$

By our assumption on $\xi_k$ and Lemma 22, and our assumption that $\delta_k \leq \frac{32\sigma_\xi^4}{m^2 d(x_k,x^*)^2}$,

$$\mathbb{E}\left[\exp\left(\sqrt{\delta_k}2s\left\langle \xi_k(x_k), \gamma_k'(0) \right\rangle\right)\right] \leq \mathbb{E}\left[\exp\left(8\delta_k s^2 d(x_k, x^*)^2\|\xi_k(x_k)\|^2\right)\right]$$
$$\leq \mathbb{E}\left[\exp\left(8\delta_k s^2 d(x_k, x^*)^2 \sigma_\xi^2\right)\right]$$
$$\leq \mathbb{E}\left[\exp\left(\frac{sm\delta_k}{2}d(x_k, x^*)^2\right)\right]$$

On the other hand, by our assumption on $\xi_k$ and $\delta_k \le \frac{\sigma_\xi^2}{m(1+\sqrt{L_R}\mathrm{d}(x_k,x^*))}$,

$$\mathbb{E}\left[\exp\left(4\delta_k s(1+\sqrt{L_R}\mathrm{d}(x_k,x^*))\|\xi_k(x_k)\|^2\right)\right] \le \exp\left(4\delta_k s(1+\sqrt{L_R}\mathrm{d}(x_k,x^*))\sigma_\xi^2\right)$$

$$\le \exp\left(\frac{sm\delta_k}{2}\mathrm{d}(x_k,x^*)^2 + \frac{128\delta_k s L_R \sigma_\xi^4}{m} + 4\delta_k s\sigma_\xi^2\right)$$

Plugging both of the above into (C.8),

$$\mathbb{E}\left[\exp\left(\sqrt{\delta_k}\nu_k\right)\right] \le \exp\left(\frac{\delta_k ms}{4}\mathrm{d}(x_k,x^*)^2 + \frac{64\delta_k s L_R \sigma_\xi^4}{m} + 2\delta_k s\sigma_\xi^2\right)$$

We thus verify that $(\nu_k, \lambda, \mu)$ satisfy the requirement for Lemma 24, which bounds

$$\mathbb{P}(\max_{i\le K} q_i \ge t^2) \le 8K\delta\lambda\exp\left(q_0 + \frac{8(\gamma+\mu)}{\lambda} - \frac{t^2}{2}\right)$$

$$\le 8K\delta\lambda\exp\left(\frac{m\mathcal{R}^2}{16\sigma_\xi^2} + \frac{L_\beta'\mathcal{R}^2}{\sigma_\xi^2} + \frac{1}{2} + \frac{16L_R\sigma_\xi^2}{m} - \frac{t^2}{2}\right)$$

$$\le 16K\delta\lambda\exp\left(\frac{2L_\beta'\mathcal{R}^2}{\sigma_\xi^2} + \frac{16L_R\sigma_\xi^2}{m} - \frac{t^2}{2}\right)$$

where the second inequality plugs in definitions from (C.7), uses our assumption on $x_0$. Finally, using the fact that $q_k = s\mathrm{d}(x_k, x^*)$,

$$\mathbb{P}(\max_{i\le K}\mathrm{d}(x_i, x^*) \ge t) \le 32K\delta\lambda\exp\left(\frac{2L_\beta'\mathcal{R}^2}{\sigma_\xi^2} + \frac{16L_R\sigma_\xi^2}{m} - \frac{mt^2}{64\sigma_\xi^2}\right)$$

$\square$

**Lemma 20** (Subgaussian Bound under Dissipativity, SGLD, fixed stepsize). *Let $\beta$ be a vector field satisfying Assumption 3. Let $x^*$ be some point with $\beta(x^*) = 0$. Assume that for all $x$ such that $\mathrm{d}(x, x^*) \ge \mathcal{R}$, there exists a minimizing geodesic $\gamma : [0,1] \to M$ with $\gamma(0) = x, \gamma(1) = x^*$, and*

$$\langle\beta(x), \gamma'(0)\rangle \le -m\mathrm{d}(x,x^*)^2$$

*. Let $r \in \mathbb{R}^+$ denote an arbitrary radius, and assume that $\delta$ is a stepsize satisfying*

$$\delta \le \min\left\{\frac{m}{16L_\beta'^2(1+\sqrt{L_R}r)}, \frac{d+\sigma^2}{m(1+\sqrt{L_R}r)}, \frac{32(d^2+\sigma^4)}{m^2r^2}\right\}$$

*. Let $x_k$ be the following stochastic process:*

$$x_{k+1} = \mathrm{Exp}_{x_k}\left(\delta\tilde{\beta}_k(x_k) + \sqrt{\delta}\zeta_k(x_k)\right)$$

*where $\zeta_k(x_k) \sim \mathcal{N}_{x_k}(0, I)$ and $\tilde{\beta}_k(x)$ satisfies, for all $x$, $\mathbb{E}\left[\tilde{\beta}_k(x)\right] = \beta(x)$ and $\left\|\tilde{\beta}_k(x) - \beta(x)\right\| \le \sigma$. Assume that $\mathrm{d}(x_0, x^*) \le 2\mathcal{R}$. Then*

$$\mathbb{P}(\max_{k\le K}\mathrm{d}(x_k, x^*) \ge r) \le 32K\delta m\exp\left(\frac{2L_\beta'^2\mathcal{R}^2}{d+\sigma^2} + \frac{64L_R(d+\sigma^2)}{m} - \frac{mr^2}{256(d+\sigma^2)}\right)$$

*Proof.* We begin by defining $\xi_k(x_k) := \zeta_k(x_k) + \sqrt{\delta}(\tilde{\beta}(x_k) - \beta(x_k))$. We verify that

$$x_{k+1} = \mathrm{Exp}_{x_k}\left(\delta\tilde{\beta}_k(x_k) + \sqrt{\delta}\xi_k(x_k)\right).$$

We verify that $\mathbb{E}\left[\xi_k\right] = 0$ and that for any $\lambda \le \frac{1}{8}$,

$$\mathbb{E}\left[\exp\left(\lambda\|\xi_k(x_k)\|^2\right)\right] \le \mathbb{E}\left[\exp\left(2\lambda\|\zeta_k(x_k)\|^2 + 2\lambda\sigma^2\right)\right] \le \exp\left(4\lambda d + 2\lambda\sigma^2\right)$$

where we use Lemma 21. Let us define $\sigma_\xi := \sqrt{4d + \sigma^2}$.

Next, let us define, for analysis purposes, the following process:

$$\tilde{x}_{k+1} = \text{Exp}_{\tilde{x}_k}\left(\delta_k \beta(\tilde{x}_k) + \sqrt{\delta_k}\xi_k(\tilde{x}_k)\right)$$

initialized at $\tilde{x}_0 = x_0$ and where

$$\delta_k := \min\left\{\delta, \frac{m}{16L_\beta'^2(1 + \sqrt{L_R}\text{d}(\tilde{x}_k, x^*))}, \frac{\sigma_\xi^2}{m(1 + \sqrt{L_R}\text{d}(\tilde{x}_k, x^*))}, \frac{32\sigma_\xi^4}{m^2\text{d}(\tilde{x}_k, x^*)^2}\right\}$$

Define the event $A_k := \{\max_{i \leq k}\text{d}(\tilde{x}_i, x^*) \leq r\}$. Under the event $A_k$, $\delta_i = \delta$ for all $i \leq k$, and consequently, $\tilde{x}_i = x_i$ for all $i \leq k$. Therefore, $A_k = \{\max_{i \leq k}\text{d}(x_i, x^*) \leq r\}$, and thus

$$\mathbb{P}(\max_{k \leq K}\text{d}(x_k, x^*) \geq r)$$
$$=\mathbb{P}(A_k^c)$$
$$=\mathbb{P}(\max_{k \leq K}\text{d}(\tilde{x}_k, x^*) \geq r)$$

$$\leq 32K\delta m \exp\left(\frac{2L_\beta'^2\mathcal{R}^2}{\sigma_\xi^2} + \frac{16L_R\sigma_\xi^2}{m} - \frac{mr^2}{64\sigma_\xi^2}\right)$$

where the last inequality follows from Lemma 19 with $\xi_k$ and $\sigma_\xi$ as defined above. □

## C.4 Misc

**Lemma 21.** *For $\lambda \leq \frac{1}{4}$ and $\xi \sim \mathcal{N}(0, I_{d \times d})$,*

$$\mathbb{E}\left[\exp\left(\lambda\|\xi\|_2^2\right)\right] \leq \exp\left(\lambda d + 2\lambda^2 d\right) \leq \exp\left(2\lambda d\right)$$

*Proof.* Consequence of $\chi^2$ distribution being subexponential. □

**Lemma 22** (Hoeffding's Lemma). *Let $\eta_k$ be a 0-mean random variable. Then for all $\lambda$,*

$$\mathbb{E}_\eta\left[\exp\left(\lambda\eta\right)\right] \leq \mathbb{E}_\eta\left[\exp\left(2\lambda^2\eta^2\right)\right]$$

*Proof.*

$$\mathbb{E}_\eta\left[\exp\left(\lambda\eta\right)\right] = \mathbb{E}_\eta\left[\exp\left(\lambda\eta - \mathbb{E}_{\eta'}\left[\lambda\eta'\right]\right)\right]$$
$$\leq \mathbb{E}_{\eta,\eta'}\left[\exp\left(\lambda(\eta - \eta')\right)\right]$$
$$= \mathbb{E}_{\eta,\eta',\varepsilon}\left[\exp\left(\lambda\varepsilon(\eta - \eta')\right)\right]$$
$$\leq \mathbb{E}_{\eta,\eta'}\left[\exp\left(\lambda^2(\eta - \eta')^2/2\right)\right]$$
$$\leq \mathbb{E}_{\eta,\eta'}\left[\exp\left(\lambda^2(\eta^2 + \eta'^2)\right)\right]$$
$$= \mathbb{E}_\eta\left[\exp\left(2\lambda^2\eta^2\right)\right]$$

where $\varepsilon$ is a Rademacher random variable. □

**Lemma 23** (Corollary of Doob's maximal inequality). *Let $K$ be any positive integer. For any $k \leq K$, let $a_k, b_k, c_k, d_k \in \mathbb{R}^+$ be arbitrary positive constants. Assume that for all $k$, $a_k \leq \frac{1}{4}$ and $a_k + c_k \leq \frac{1}{4}$. Let $q_k$ be a semi-martingale of the form*

$$q_{k+1} \leq (1 + a_k)q_k + b_k + \eta_k$$

*where $\eta_k$ are random variables. Assume that for all $k$, $\eta_k$ satisfy*

$$\mathbb{E}\left[\exp\left(\eta_k\right)|\eta_0...\eta_{k-1}\right] \leq \exp\left(c_k q_k + d_k\right)$$

*Assume in addition that $q_k \geq 0$ almost surely, for all $k$. Finally, assume that $\sum_{k=0}^K a_k + c_k \leq \frac{1}{8}$. Then*

$$\mathbb{P}(\max_{k \leq K}q_k \geq t^2) \leq \exp\left(q_0 + \sum_{k=0}^K(b_k + d_k) - \frac{t^2}{2}\right)$$

*Proof.* Let us first define

$$r_0 := q_0$$
$$r_{k+1} := (1 + a_k)r_k + b_k + \eta_k$$

i.e. $r_k$ is very similar to $q_k$, only difference being that we replaced $\leq$ by $=$.

We first verify that for all $k \leq K$, $r_k \geq q_k$. For $k = 0$, by definition, $r_0 = q_0$. Now assume that $r_k \geq q_k$ for some $k$. Then for $k + 1$,

$$r_{k+1} := (1 + a_k)r_k + b_k + \eta_k$$
$$\geq (1 + a_k)q_k + b_k + \eta_k$$
$$\geq q_{k+1}$$

We verify below that $\exp(r_k)$ is a sub-martingale: conditioning on $\eta_0...\eta_{k-1}$, and taking expectation wrt $\eta_k$,

$$\mathbb{E}\left[\exp(r_{k+1})|\eta_0...\eta_k\right]$$
$$= \mathbb{E}\left[\exp((1 + a_k)r_k + b_k + \eta_k)|\eta_0...\eta_k\right]$$
$$= \exp((1 + a_k)r_k + b_k) \cdot \mathbb{E}\left[\exp(\eta_k)|\eta_0...\eta_k\right]$$
$$\geq \exp((1 + a_k)r_k + b_k)$$
$$\geq \exp(r_k)$$

where the first inequality is by convexity of $\exp$, and $\mathbb{E}[\eta_k] = 0$, and Jensen's inequality.

Let us now define $s_k := \prod_{i=0}^{k-1}(1 + a_i + c_i)^{-1}$. We can upper bound

$$\mathbb{E}\left[\exp(s_{k+1}r_{k+1})\right]$$
$$= \mathbb{E}\left[\exp(s_{k+1}((1 + a_k)r_k + b_k + \eta_k))\right]$$
$$= \mathbb{E}\left[\exp(s_{k+1}(1 + a_k)r_k + s_{k+1}b_k) \cdot \mathbb{E}\left[\exp(s_{k+1}\eta_k)|\eta_0...\eta_k\right]\right]$$
$$\leq \mathbb{E}\left[\exp(s_{k+1}(1 + a_k)r_k + s_{k+1}b_k) \cdot \mathbb{E}\left[\exp(s_{k+1}\eta_k)|\eta_0...\eta_k\right]\right]$$
$$\leq \mathbb{E}\left[\exp(s_{k+1}(1 + a_k)r_k + s_{k+1}b_k) \cdot (\mathbb{E}\left[\exp(\eta_k)|\eta_0...\eta_k\right])^{s_{k+1}}\right]$$
$$\leq \mathbb{E}\left[\exp(s_{k+1}(1 + a_k)r_k + s_{k+1}(b_k + c_kq_k + d_k))\right]$$
$$= \mathbb{E}\left[\exp(s_kr_k)\right] \cdot \exp(s_{k+1}(b_k + d_k))$$

where the second inequality is by Lemma 22, the third inequality is by the fact that $s_k \leq 1$ for all $k$ and by Jensen's inequality, the fourth inequality uses our assumption on $\eta_k$ in the Lemma statement, as well as the fact that $s_k \leq 1$ for all $k$. The last equality is by definition of $s_k$ and because $q_k \leq r_k$. Applying this recursively gives

$$\mathbb{E}\left[\exp(s_Kr_K)\right] \leq \exp(r_0) \cdot \exp\left(\sum_{k=0}^{K} s_{k+1}(b_k + d_k)\right) \leq \exp\left(r_0 + \sum_{k=0}^{K}(b_k + d_k)\right)$$

By Doob's maximal inequality (recall that we $e^{r_k}$ is a sub-martingale),

$$\mathbb{P}(\max_{k \leq K} q_k \geq t^2) \leq \mathbb{P}(\max_{k \leq K} r_k \geq t^2)$$
$$= \mathbb{P}(\max_{k \leq K} \exp(s_Kr_k) \geq \exp(s_Kt^2))$$
$$\leq \mathbb{E}\left[\exp(s_Kr_K)\right] \cdot \exp\left(-\frac{t^2}{2}\right)$$
$$\leq \exp\left(r_0 + \sum_{k=0}^{K} s_{k+1}(b_k + d_k) - \frac{t^2}{2}\right) \quad\quad\quad (C.9)$$

The first equality uses our assumption that $s_K = \prod_{k=0}^{K-1}(1 + a_k + c_k)^{-1} \geq e^{-\sum_{k=0}^{K-1} 4(a_k + c_k)} \geq e^{-\frac{1}{4}} \geq \frac{1}{2}$ and the fact that $r_K \geq q_k \geq 0$. $\qquad\square$

**Lemma 24** (Uniform Bound). *Let $K$ be any positive integer. For $k \leq K$, let $\delta_k \in \mathbb{R}^+$, let $\lambda, \gamma, \mu \in \mathbb{R}^+$. Let $q_k$ be a sequence of random numbers of the form*

$$q_{k+1} \leq (1 - \delta_k \lambda) q_k + \delta_k \gamma + \sqrt{\delta_k} \nu_k$$

*where $\nu_k$ are random variables. Assume that $q_k$ and $\nu_k$ are measurable wrt some filtration $\mathcal{F}_k$. Assume that for all $k$, $\nu_k$ satisfy*

$$\mathbb{E}\left[ \exp\left( \sqrt{\delta_k} \nu_k \right) | \mathcal{F}_k \right] \leq \exp\left( \delta_k \lambda q_k / 2 + \delta_k \mu \right)$$

*Assume that there is a constant $\delta$ such that for all $k$, $\delta_k \leq \delta \leq \frac{1}{8\lambda}$. Assume in addition that $q_k \geq 0$ almost surely, for all $k$. Then*

$$\mathbb{P}(\max_{i \leq K} q_i \geq t^2) \leq 8K\delta\lambda \exp\left( q_0 + \frac{8(\gamma + \mu)}{\lambda} - \frac{t^2}{2} \right)$$

*Proof.* For any $s \leq 1$,

$$\begin{aligned}
\mathbb{E}\left[ \exp\left( sq_{k+1} \right) \right] &\leq \mathbb{E}\left[ \exp\left( s((1 - \delta_k\lambda)q_k + \delta_k\gamma + \sqrt{\delta_k}\nu_k) \right) \right] \\
&= \mathbb{E}\left[ \exp\left( s((1 - \delta_k\lambda)q_k + \delta_k\gamma) \right) \cdot \mathbb{E}\left[ \exp\left( s\sqrt{\delta_k}\nu_k \right) \right] \right] \\
&\leq \mathbb{E}\left[ \exp\left( s((1 - \delta_k\lambda)q_k + \delta_k\gamma) \right) \cdot \left( \mathbb{E}\left[ \exp\left( \sqrt{\delta_k}\nu_k \right) \right] \right)^s \right] \\
&\leq \mathbb{E}\left[ \exp\left( s((1 - \delta_k\lambda/2)q_k + \delta_k(\gamma + \mu)) \right) \right]
\end{aligned}$$

Applying the above recursively, for any $k$, we can bound

$$\begin{aligned}
\mathbb{E}\left[ \exp\left( q_k \right) \right] &\leq \mathbb{E}\left[ \exp\left( (1 - \delta_k\lambda/2)q_{k-1} + \delta_k(\gamma + \mu) \right) \right] \\
&\leq \dots \\
&\leq \mathbb{E}\left[ \exp\left( \prod_{i=0}^{k-1}(1 - \delta_i\lambda/2)q_0 + \sum_{i=0}^{k-1}\prod_{j=i}^{k-1}(1 - \delta_j\lambda/2)(\delta_i(\gamma + \mu)) \right) \right] \\
&\leq \mathbb{E}\left[ \exp\left( e^{-\frac{\lambda}{2}\sum_{i=0}^{k-1}\delta_i}q_0 + (\gamma + \mu)\sum_{i=0}^{k-1}e^{-\frac{\lambda}{2}\sum_{j=i}^{k-1}\delta_j}\delta_i \right) \right] \quad \text{(C.10)}
\end{aligned}$$

Let us define $t_k := \sum_{i=0}^{k}\delta_i$. By our assumption that $\delta_i \leq \frac{1}{4\lambda}$, we can verify that $\sum_{i=0}^{k-1} e^{\frac{\lambda}{2}\sum_{j=i}^{k-1}\delta_j}\delta_i \leq 2\int_0^{t_k} e^{-\frac{\lambda(t_k - t)}{2}}dt \leq \frac{4}{\lambda}$. Therefore, for all $k$,

$$\mathbb{E}\left[ \exp\left( q_k \right) \right] \leq \exp\left( e^{-\frac{\lambda}{2}\sum_{i=0}^{k-1}\delta_i}q_0 + \frac{4(\gamma + \mu)}{\lambda} \right)$$

Let us now define $N := \left\lceil \frac{1}{4\delta\lambda} \right\rceil \geq 1$ (inequality is because $\delta \leq \delta_k \leq \frac{1}{8\lambda}$). We verify that $q_{k+1} \leq q_{k-1} + \delta_k(\gamma + \mu) + \eta_k$. Let us now apply Lemma 23 with $\eta_k = \sqrt{\delta_k}\nu_k$, $a_k = 0$, $b_k = \delta_k\gamma$, $c_k = \lambda/2$, $d_k = \mu$ and the fact that $\delta_k\lambda \leq 1/4$ to bound, for any $k$,

$$\mathbb{P}(\max_{i \leq N} q_{k+i} \geq t^2) \leq \mathbb{E}\left[ \exp\left( q_k + (\gamma + \mu)\sum_{i=0}^{N}\delta_{i+N} - \frac{t^2}{2} \right) \right] \leq \exp\left( q_0 + \frac{8(\gamma + \mu)}{\lambda} - \frac{t^2}{2} \right)$$

where we use the fact that $N \leq \frac{1}{2\delta\lambda}$

Applying union bound over the events $\{\max_{i \leq N} q_{k+i} \geq t^2\}$ for $k = 0, N, 2N\dots$, we can bound, for any positive integer $M$,

$$\begin{aligned}
\mathbb{P}(\max_{i \leq MN} q_i \geq t^2) &\leq \sum_{j=0}^{M-1}\mathbb{P}(\max_{i \leq N} q_{jN+i} \geq t^2) \\
&\leq \sum_{j=0}^{M-1}\exp\left( e^{-\frac{\lambda}{2}\sum_{i=0}^{jN}\delta_i}q_0 + \frac{8(\gamma + \mu)}{\lambda} - \frac{t^2}{2} \right) \\
&\leq M\exp\left( q_0 + \frac{8(\gamma + \mu)}{\lambda} - \frac{t^2}{2} \right)
\end{aligned}$$

Plugging in $M = \frac{K}{N} \leq 8K\delta\lambda$, it follows that for any $K$,

$$\mathbb{P}(\max_{i \leq K} q_i \geq t^2) \leq 8K\delta\lambda \exp\left(q_0 + \frac{8(\gamma + \mu)}{\lambda} - \frac{t^2}{2}\right)$$

$\square$

**Lemma 25.** *Let $M$ satisfy Assumption 4. For any 3 points $x, y, z \in M$, let $u, v \in T_y M$ be such that $z = \mathrm{Exp}_y(v)$ and $x = \mathrm{Exp}_y(u)$. Assume in addition that $\|u\| = \mathrm{d}(x, y)$ (i.e. $t \to \mathrm{Exp}_x(tu)$ is a minimizing geodesic). Then*

$$\mathrm{d}(z, x)^2 \leq \mathrm{d}(y, x)^2 - 2\langle v, u \rangle + \zeta\left(\sqrt{L_R}\mathrm{d}(y, x)\right)\|v\|^2$$

*where $\zeta(r) := \frac{r}{\tanh(r)}$.*

The above lemma is a restatement of Corollary 8 from Zhang and Sra [2016]. Although Zhang and Sra [2016] required minimizizng geodesics to be unique, their proof, based on Lemma 6 of the same paper, works even if minimizing geodesics are not unique.

# D   Fundamental Manifold Results

In this section, we provide Taylor expansion style inequalities for the evolution of geodesics on manifold. By making use of our bounds for matrix ODEs in Section E, we can bound the distance between two points along geodesics under various conditions. Most of our analysis is based on some variant of the Jacobi equation $D_t^2 J + R(J, \gamma')\gamma' = 0$.

Notably,

1. Lemma 28 quantifies the distance evolution between $x(t) = \mathrm{Exp}_x(tu)$ and $y(t) = \mathrm{Exp}_y(tv)$. This is the key to proving Lemma 7, which bounds the distance between Euler Murayama discretization (1.2) and (1.1).

2. Lemma 29 is a more refined version of Lemma 28. Lemma 29 is key to proving Lemma 8, which is in turn key to proving Theorem 2. Lemma 29 is also used to analyze the distance evolution of two processes under the Kendall-Cranston coupling in Lemma 3.

## D.1   Jacobi Field Approximations

In the following lemma, we consider a variation field $\Lambda(s, t)$, where for each $s$, $\Lambda(s, t)$ is a geodesic. We bound the error between $\Lambda(s, t)$, and its Taylor approximation of various orders. This lemma is key to proving Lemma 28 and Lemma 28.

**Lemma 26.** *Let $\Lambda(s, t) : [0, 1] \times [0, 1] \to M$ be a variation field, where for each fixed $s$, $t \to \Lambda(s, t)$ is a geodesic. Let us define $\mathcal{C} := \sqrt{L_R \|\partial_t \Lambda(s, 0)\|^2}$. Then for all $s, t \in [0, 1]$,*

$$\|\partial_s \Lambda(s, t)\| \leq \cosh(\mathcal{C})\|\partial_s \Lambda(s, 0)\| + \frac{\sinh(\mathcal{C})}{\mathcal{C}}\|D_t \partial_s \Lambda(s, 0)\|$$

$$\left\|\partial_s \Lambda(s, t) - \Gamma_{\Lambda(s,0)}^{\Lambda(s,t)}(\partial_s \Lambda(s, 0) + tD_t \partial_s \Lambda(s, 0))\right\| \leq (\cosh(\mathcal{C}) - 1)\|\partial_s \Lambda(s, 0)\| + (\frac{\sinh(\mathcal{C})}{\mathcal{C}} - 1)\|D_t \partial_s \Lambda(s, 0)\|$$

$$\left\|\partial_s \Lambda(s, t) - \Gamma_{\Lambda(s,0)}^{\Lambda(s,t)}(\partial_s \Lambda(s, 0))\right\| \leq (\cosh(\mathcal{C}) - 1)\|\partial_s \Lambda(s, 0)\| + \frac{\sinh(\mathcal{C})}{\mathcal{C}}\|D_t \partial_s \Lambda(s, 0)\|$$

$$\|D_t \partial_s \Lambda(s, t)\| \leq \mathcal{C} \sinh(\mathcal{C})\|\partial_s \Lambda(s, 0)\| + \cosh(\mathcal{C})\|D_t \partial_s \Lambda(s, 0)\|$$

$$\left\|D_t \partial_s \Lambda(s, t) - \Gamma_{\Lambda(s,0)}^{\Lambda(s,t)}(D_t \partial_s \Lambda(s, 0))\right\| \leq \mathcal{C} \sinh(\mathcal{C})\|\partial_s \Lambda(s, 0)\| + (\cosh(\mathcal{C}) - 1)\|D_t \partial_s \Lambda(s, 0)\|$$

*If in addition, the derivative of the Riemannian curvature tensor is globally bounded by $L_R'$, then*

$$\left\|D_t \partial_s \Lambda(s, t) - \Gamma_{\Lambda(s,0)}^{\Lambda(s,t)}(D_t \partial_s \Lambda(s, 0)) - t\Gamma_{\Lambda(s,0)}^{\Lambda(s,t)}(D_t D_t \partial_s \Lambda(s, 0))\right\|$$
$$\leq (L_R'\|\partial_t \Lambda(s, 0)\|^3 + \mathcal{C}^4)e^{\mathcal{C}}\|\partial_s \Lambda(s, 0)\| + (L_R'\|\partial_t \Lambda(s, 0)\|^3 + \mathcal{C}^2)e^{\mathcal{C}}\|D_t \partial_s \Lambda(s, 0)\|$$

*Proof.* For any fixed $s$, let $E_i(s, 0)$ be a basis of $T_{\Lambda(s,0)}M$.

Let $E_i(s, t)$ denote an orthonormal frame along $\gamma_s(t) := \Lambda(s, t)$, by parallel transporting $E_i(s, 0)$.

Let $\mathbf{J}(s, t) \in \mathbb{R}^d$ denote the coordinates of $\partial_s\Lambda(s, t)$ wrt $E_i(s, t)$. Let $\mathbf{K}(s, t) \in \mathbb{R}^d$ denote the coordinates of $D_t\partial_s\Lambda(s, t)$ wrt $E_i(s, t)$. Let $\mathbf{a}(s, t) \in \mathbb{R}^d$ denote the coordinates of $\partial_t\Lambda(s, t)$ wrt $E_i(s, t)$ (this is constant for fixed $s$, for all $t$). Let $\mathbf{R}(s, t) \in \mathbb{R}^{4d}$ be such that $\mathbf{R}^i_{jkl} = \langle R(E_j(s, t), E_k(s, t))E_l(s, t), E_i(s, t)\rangle$. Let $\mathbf{M}(s, t)$ denote the matrix with $\mathbf{M}_{i,j}(s, t) := -\sum_{k,l} \mathbf{R}^i_{jkl}(s, t)\mathbf{a}_k(s, t)\mathbf{a}_l(s, t)$.

Notice that $\mathbf{M}_{i,j}$ is symmetric, since $\mathbf{M}_{i,j} = \langle R(E_j, \mathbf{a})\mathbf{a}, E_i\rangle = \langle R(\mathbf{a}, E_i)E_j, \mathbf{a}\rangle = \langle R(E_i, \mathbf{a})\mathbf{a}, E_j\rangle$, where the first equality is by interchange symmetry and second equality is by skew symmetry of the Riemannian curvature tensor. Therefore, by definition of $L_R$ in Assumption 4, it follows that $\|\mathbf{M}(s, t)\|_2 \leq L_R\|\partial_t\Lambda(s, t)\|^2 = L_R\|\partial_t\Lambda(s, 0)\|^2$.

The Jacobi Equation states that $D_tD_t\partial_s\Lambda(s, t) = -R(\partial_s\Lambda(s, t), \partial_t\Lambda(s, t))\partial_t\Lambda(s, t)$. We verify that $-\langle R(\partial_s, \partial_t)\partial_t, E_i\rangle = -\sum_{j,k,l} \mathbf{R}^i_{jkl}\mathbf{J}_j\mathbf{a}_k\mathbf{a}_l = [\mathbf{M}(s, t)\mathbf{J}(s, t)]_i$, thus $D_tD_t\partial_s\Lambda(s, t) = \sum_{i=1}^d [\mathbf{M}(s, t)\mathbf{J}(s, t)]_i \cdot E_i(s, t)$.

We verify that $\frac{d}{dt}\mathbf{J}_i(s, t) = D_t\langle\partial_s\Lambda(s, t), E_i(s, t)\rangle = \langle D_t\partial_s\Lambda(s, t), E_i(s, t)\rangle = \mathbf{K}_i(s, t)$. We also verify that $\frac{d}{dt}\mathbf{K}_i(s, t) = D_t\langle D_t\partial_s\Lambda(s, t), E_i(s, t)\rangle = \langle D_tD_t\partial_s\Lambda(s, t), E_i(s, t)\rangle = [\mathbf{M}(s, t)\mathbf{J}(s, t)]_i$.

Let us now consider a fixed $s$. To simplify notation, we drop the $s$ dependence. The Jacobi Equation, in coordinate form, corresponds to the following second-order ODE:

$$\frac{d}{dt}\mathbf{J}(t) = \mathbf{K}(t)$$
$$\frac{d}{dt}\mathbf{K}(t) = \mathbf{M}(t)\mathbf{J}(t)dt$$

Define $L_{\mathbf{M}} := L_R\|\partial_t\Lambda(s, 0)\|^2 = \mathcal{C}^2$. We verify that $L_{\mathbf{M}} \geq \max_{t \in [0,1]} \|\mathbf{M}(t)\|_2$. Then from 34, we see that

$$\begin{bmatrix}\mathbf{J}(t)\\\mathbf{K}(t)\end{bmatrix} = \mathbf{exp_{mat}}(t; \mathbf{M})\begin{bmatrix}\mathbf{J}(0)\\\mathbf{K}(0)\end{bmatrix}$$

From Lemma 35,

$$\mathbf{exp_{mat}}(t; \begin{bmatrix}0 & I\\\mathbf{M}(t) & 0\end{bmatrix}) = \begin{bmatrix}\mathbf{A}(t) & \mathbf{B}(t)\\\mathbf{C}(t) & \mathbf{D}(t)\end{bmatrix}$$

where each block is $\mathbb{R}^{2d}$, and can be bounded as

$$\|\mathbf{A}(t)\|_2 \leq \cosh(\mathcal{C}t) \leq \cosh(\mathcal{C})$$
$$\|\mathbf{B}(t)\|_2 \leq \frac{\sinh(\mathcal{C}t)}{\mathcal{C}} \leq \frac{\sinh(\mathcal{C})}{\mathcal{C}}$$
$$\|\mathbf{C}(t)\|_2 \leq \mathcal{C}\sinh(\mathcal{C}t) \leq \mathcal{C}\sinh(\mathcal{C})$$
$$\|\mathbf{D}(t)\|_2 \leq \cosh(\mathcal{C}t) \leq \cosh(\mathcal{C})$$
$$\|\mathbf{A}(t) - I\|_2 \leq \cosh(\mathcal{C}t) - 1 \leq \cosh(\mathcal{C}) - 1$$
$$\|\mathbf{B}(t) - tI\|_2 \leq \frac{\sinh(\mathcal{C}t)}{\mathcal{C}} - t \leq \frac{\sinh(\mathcal{C})}{\mathcal{C}} - 1$$
$$\|\mathbf{D}(t) - I\|_2 \leq \cosh(\mathcal{C}t) - 1 \leq \cosh(\mathcal{C}) - 1$$

where we use the fact that $\cosh(r)$, $\sinh(r)$ and $\frac{\sinh(r)}{r}$ are monotonically increasing and that $\frac{\sinh(r)}{r} - 1 \geq 0$ for positive $r$.

It follows that

$$\mathbf{J}(t) = \mathbf{A}(t)\mathbf{J}(0) + \mathbf{B}(t)\mathbf{K}(0)$$
$$\mathbf{K}(t) = \mathbf{C}(t)\mathbf{J}(0) + \mathbf{D}(t)\mathbf{K}(0) \tag{D.1}$$

Thus

$$\|\partial_s\Lambda(s,t)\|$$
$$=\|\mathbf{J}(t)\|$$
$$=\|\mathbf{A}(t)\mathbf{J}(0) + \mathbf{B}(t)\mathbf{K}(0)\|$$
$$\leq \cosh\left(\mathcal{C}\right)\|\partial_s\Lambda(s,0)\| + \frac{\sinh\left(\mathcal{C}\right)}{\mathcal{C}}\|D_t\partial_s\Lambda(s,0)\|$$

and

$$\left\|\partial_s\Lambda(s,t) - \{\partial_s\Lambda(s,0) + tD_t\partial_s\Lambda(s,0)\}^{\rightarrow\Lambda(s,t)}\right\|$$
$$=\|\mathbf{J}(t) - \mathbf{J}(0) - t\mathbf{K}(0)\|_2$$
$$=\|(\mathbf{A}(t) - I)\mathbf{J}(0) + (\mathbf{B}(t) - tI)\mathbf{K}(0)\|_2$$
$$\leq(\cosh\left(\mathcal{C}\right) - 1)\|\partial_s\Lambda(s,0)\| + (\frac{\sinh\left(\mathcal{C}\right)}{\mathcal{C}} - 1)\|D_t\partial_s\Lambda(s,0)\|$$

and

$$\left\|\partial_s\Lambda(s,t) - \{\partial_s\Lambda(s,0)\}^{\rightarrow\Lambda(s,t)}\right\|$$
$$\leq\|(\mathbf{A}(t) - I)\mathbf{J}(0) + \mathbf{B}(t)\mathbf{K}(0)\|_2$$
$$\leq(\cosh\left(\mathcal{C}\right) - 1)\|\partial_s\Lambda(s,0)\| + \frac{\sinh\left(\mathcal{C}\right)}{\mathcal{C}}\|D_t\partial_s\Lambda(s,0)\|$$

Similarly,

$$\|D_t\partial_s\Lambda(s,t)\|$$
$$=\|\mathbf{K}(t)\|_2$$
$$\leq\|\mathbf{C}(t)\mathbf{J}(0)\|_2 + \|\mathbf{D}(t)\mathbf{K}(0)\|_2$$
$$\leq\mathcal{C}\sinh\left(\mathcal{C}\right)\|\partial_s\Lambda(s,0)\| + \cosh\left(\mathcal{C}\right)\|D_t\partial_s\Lambda(s,0)\|$$

and

$$\left\|D_t\partial_s\Lambda(s,t) - \{D_t\partial_s\Lambda(s,0)\}^{\rightarrow\Lambda(s,t)}\right\|$$
$$=\|\mathbf{K}(t) - \mathbf{K}(0)\|_2$$
$$=\|\mathbf{C}(t)\mathbf{J}(0) + (\mathbf{D}(t) - I)\mathbf{K}(0)\|_2$$
$$\leq\mathcal{C}\sinh\left(\mathcal{C}\right)\|\partial_s\Lambda(s,0)\| + (\cosh\left(\mathcal{C}\right) - 1)\|D_t\partial_s\Lambda(s,0)\|$$

To prove the last bound, let us define $L'_{\mathbf{M}} := L'_R\|\partial_t\Lambda(s,0)\|^3$. We verify that $L'_{\mathbf{M}} \geq \max_{t\in[0,1]}\|\mathbf{M}(t) - \mathbf{M}(0)\|_2$.

we know that

$$\left\|D_t\partial_s\Lambda(s,t) - \{D_t\partial_s\Lambda(s,0)\}^{\rightarrow\Lambda(s,t)} - t\{D_tD_t\partial_s\Lambda(s,0)\}^{\rightarrow\Lambda(s,t)}\right\|$$
$$=\|\mathbf{K}(t) - \mathbf{K}(0) - t\mathbf{M}(0)\mathbf{J}(0)\|_2$$
$$=\left\|\int_0^t \mathbf{M}(r)\mathbf{J}(r) - \mathbf{M}(0)\mathbf{J}(0)dr\right\|_2$$
$$\leq\int_0^t\|\mathbf{M}(r) - \mathbf{M}(0)\|_2\|\mathbf{J}(0)\|_2 dr + \int_0^t\|\mathbf{M}(r)\|_2\|\mathbf{J}(r) - \mathbf{J}(0)\|_2 dr$$
$$\leq\int_0^t L'_{\mathbf{M}}\|\partial_s\Lambda(s,r)\| + L_{\mathbf{M}}\left\|\partial_s\Lambda(s,r) - \Gamma_{\Lambda(s,0)}^{\Lambda(s,r)}(\partial_s\Lambda(s,0))\right\|dr$$

where the last line follows from (D.1). From our earlier results in this lemma,

$$\|\partial_s\Lambda(s,r)\| \leq \cosh\left(\mathcal{C}\right)\|\partial_s\Lambda(s,0)\| + \frac{\sinh\left(\mathcal{C}\right)}{\mathcal{C}}\|D_t\partial_s\Lambda(s,0)\|$$
$$\leq e^{\mathcal{C}}\|\partial_s\Lambda(s,0)\| + e^{\mathcal{C}}\|D_t\partial_s\Lambda(s,0)\|$$
$$\left\|\partial_s\Lambda(s,t) - \Gamma_{\Lambda(s,0)}^{\Lambda(s,t)}(\partial_s\Lambda(s,0))\right\| \leq (\cosh\left(\mathcal{C}\right) - 1)\|\partial_s\Lambda(s,0)\| + \frac{\sinh\left(\mathcal{C}\right)}{\mathcal{C}}\|D_t\partial_s\Lambda(s,0)\|$$
$$\leq \mathcal{C}^2 e^{\mathcal{C}}\|\partial_s\Lambda(s,0)\| + e^{\mathcal{C}}\|D_t\partial_s\Lambda(s,0)\|$$

where the simplifications are from Lemma 39. Put together,

$$\left\| D_t \partial_s \Lambda(s,t) - \{D_t \partial_s \Lambda(s,0)\}^{\rightarrow \Lambda(s,t)} - t \{D_t D_t \partial_s \Lambda(s,0)\}^{\rightarrow \Lambda(s,t)} \right\|$$
$$\leq (L_R' \|\partial_t \Lambda(s,0)\|^3 + \mathcal{C}^4) e^{\mathcal{C}} \|\partial_s \Lambda(s,0)\| + (L_R' \|\partial_t \Lambda(s,0)\|^3 + \mathcal{C}^2) e^{\mathcal{C}} \|D_t \partial_s \Lambda(s,0)\|$$

$\square$

**Lemma 27.** *Let $x, y, z \in M$, with $x = \text{Exp}_z(u)$, $y = \text{Exp}_z(v)$.*

$$\text{d}(x,y) \leq \frac{\sinh\left(\sqrt{L_R}(\|u\| + \|v\|)t\right)}{\sqrt{L_R}(\|u\| + \|v\|)} \|v - u\|$$

*Proof.* Let us define the variational field

$$\Lambda(s,t) = \text{Exp}_z\left(t(u + s(v - u))\right)$$

We verify that

$$\partial_s \Lambda(s,0) = 0$$
$$\partial_t \Lambda(s,0) = u + s(v - u)$$
$$D_t \partial_s \Lambda(s,0) = v - u$$

Lemma 26 then immediately gives

$$\|\partial_s \Lambda(s,t)\| \leq \frac{\sinh\left(\sqrt{L_R}(\|u\| + \|v\|)t\right)}{\sqrt{L_R}(\|u\| + \|v\|)} \|v - u\|$$

$\square$

### D.2 Discrete Coupling Bounds

This section presents two key lemmas which play an important role in many of our proofs. Lemma 28 analyzes the distance between $\text{Exp}_x(u)$ and $\text{Exp}_y(v)$, as a function of $x, y$ and $u, v$. Notably, Lemma 28 implies that when $u, v$ are "parallel", i.e. $u - \Gamma_y^x v = 0$, then the distance between $\text{d}(\text{Exp}_x(u), \text{Exp}_y(v))$ is not much larger than $\text{d}(x, y)$. The proof of Lemma 28 is based on a first-order expansion of the Jacobi equation for Riemannian manifolds.

The second key lemma is Lemma 29. It considers a similar problem setup as Lemma 28, but is based on a second-order expansion of the Jacobi equation. It thus requires an additional bound on the derivative of the Riemannian curvature tensor. The more refined distance bound in Lemma 29 is required to properly analyze the convergence of both continuous-time SDEs (Lemma 3) as well as discrete-time stochastic processes (Lemma 8). The term $\int_0^1 \langle R...\gamma'(s) \rangle \, ds$ in the upper bound of Lemma 29 gives rise to the Ricci curvature (as opposed to sectional curvature) dependencies in our results.

**Lemma 28.** *Let $x, y \in M$. Let $\gamma(s) : [0,1] \rightarrow M$ be a minimizing geodesic between $x$ and $y$ with $\gamma(0) = x$ and $\gamma(1) = y$. Let $u \in T_x M$ and $v \in T_y M$. Let $u(s)$ and $v(s)$ be the parallel transport of $u$ and $v$ along $\gamma$, with $u(0) = u$ and $v(1) = v$.*

*Then*

$$\text{d}(\text{Exp}_x(u), \text{Exp}_y(v))^2 \leq (1 + 4\mathcal{C}^2 e^{4\mathcal{C}})\text{d}(x,y)^2 + 32 e^{\mathcal{C}} \|v(0) - u(0)\|^2 + 2 \langle \gamma'(0), v(0) - u(0) \rangle$$

*where $\mathcal{C} := \sqrt{L_R}(\|u\| + \|v\|)$.*

*Proof.* Let us consider the length function $E(\gamma) = \int_0^1 \|\gamma'(s)\|^2 ds$. We define a variation of geodesics $\Lambda(s,t)$:

$$\Lambda(s,t) := \text{Exp}_{\gamma(s)}\left(t(u(s) + s(v(s) - u(s)))\right)$$

We verify that

$$\partial_s \Lambda(s,0) = \gamma'(s)$$
$$\partial_t \Lambda(s,0) = u(s) + s(v(s) - u(s))$$
$$D_t \partial_s \Lambda(s,0) = v(s) - u(s)$$

Consider a fixed $t$, and let $\gamma_t(s) := \Lambda(s,t)$ (so $\gamma_t'(s)$ is the velocity wrt $s$).

$$\frac{d}{dt} E(\gamma_t)$$
$$= \frac{d}{dt} \int_0^1 \|\gamma_t'(s)\|^2 ds$$
$$= \int_0^1 2 \left\langle \gamma_t'(s), D_t \gamma_t'(s) \right\rangle ds$$
$$= \int_0^1 2 \left\langle \partial_s \Lambda(s,t), D_t \partial_s \Lambda(s,t) \right\rangle ds$$
$$= \int_0^1 2 \left\langle \partial_s \Lambda(s,0), D_t \partial_s \Lambda(s,0) \right\rangle ds$$
$$\quad + \int_0^1 2 \left\langle \partial_s \Lambda(s,0), \{D_t \partial_s \Lambda(s,t)\}^{\to \Lambda(s,0)} - D_t \partial_s \Lambda(s,0) \right\rangle ds$$
$$\quad + \int_0^1 2 \left\langle \partial_s \Lambda(s,t) - \{\partial_s \Lambda(s,0)\}^{\to \Lambda(s,t)}, D_t \partial_s \Lambda(s,t) \right\rangle ds \tag{D.2}$$

For any $s$, and for $t = 0$, $\partial_s \Lambda(s,0) = \gamma'(s)$ and $D_t \partial_s \Lambda(s,0) = v(s) - u(s)$. Using the fact that norms and inner products are preserved under parallel transport, the first term can be simplified as

$$\int_0^1 2 \left\langle \partial_s \Lambda(s,0), D_t \partial_s \Lambda(s,0) \right\rangle ds = 2 \left\langle \gamma'(0), v(0) - u(0) \right\rangle$$

To bound the second and third term, we use Lemma 26:

$$\|\partial_s \Lambda(s,0)\| = \|\gamma'(0)\|$$

and

$$\left\| D_t \partial_s \Lambda(s,t) - \Gamma_{\Lambda(s,0)}^{\Lambda(s,t)} (D_t \partial_s \Lambda(s,0)) \right\|$$
$$\leq \sqrt{L_R \|\partial_t \Lambda(s,0)\|^2} \sinh\left(\sqrt{L_R \|\partial_t \Lambda(s,0)\|^2}\right) \|\partial_s \Lambda(s,0)\| + \left(\cosh\left(\sqrt{L_R \|\partial_t \Lambda(s,0)\|^2}\right) - 1\right) \|D_t \partial_s \Lambda(s,0)\|$$
$$\leq \mathcal{C} \sinh(\mathcal{C}) \|\gamma'(0)\| + (\cosh(\mathcal{C}) - 1) \|v(0) - u(0)\|$$

where we use the fact that $\sqrt{L_R \|\partial_t \Lambda(s,0)\|^2} \leq \mathcal{C}$.

We can thus bound the second term of (D.2) as

$$\left| \int_0^1 2 \left\langle \partial_s \Lambda(s,0), \{D_t \partial_s \Lambda(s,t)\}^{\to \Lambda(s,0)} - D_t \partial_s \Lambda(s,0) \right\rangle ds \right|$$
$$\leq 2 \|\gamma'(0)\| \cdot \left( \mathcal{C} \sinh(\mathcal{C}) \|\gamma'(0)\| + (\cosh(\mathcal{C}) - 1) \|v(0) - u(0)\| \right)$$
$$\leq 4 \|\gamma'(0)\|^2 \left( \mathcal{C} \sinh(\mathcal{C}) + (\cosh(\mathcal{C}) - 1)^2 \right) + 4 \|v(0) - u(0)\|^2$$

Finally, to bound the third term of (D.2), we again apply Lemma 26:

$$\left\|\partial_s\Lambda(s,t) - \Gamma^{\Lambda(s,t)}_{\Lambda(s,0)}(\partial_s\Lambda(s,0))\right\|$$

$$\leq(\cosh\left(\sqrt{L_R\|\partial_t\Lambda(s,0)\|^2}\right) - 1)\|\partial_s\Lambda(s,0)\| + (\frac{\sinh\left(\sqrt{L_R\|\partial_t\Lambda(s,0)\|^2}\right)}{\sqrt{L_R\|\partial_t\Lambda(s,0)\|^2}})\|D_t\partial_s\Lambda(s,0)\|$$

$$\leq(\cosh\left(\sqrt{L_R}(\|u(s)\| + \|v(s)\|)\right) - 1)\|\gamma'(s)\| + (\frac{\sinh\left(\sqrt{L_R}(\|u(s)\| + \|v(s)\|)\right)}{\sqrt{L_R}(\|u(s)\| + \|v(s)\|)})\|v(s) - u(s)\|$$

$$=(\cosh\left(\mathcal{C}\right) - 1)\|\gamma'(0)\| + (\frac{\sinh\left(\mathcal{C}\right)}{\mathcal{C}})\|v(0) - u(0)\|$$

and

$$\|D_t\partial_s\Lambda(s,t)\|$$

$$\leq\sqrt{L_R\|\partial_t\Lambda(s,0)\|^2}\sinh\left(\sqrt{L_R\|\partial_t\Lambda(s,0)\|^2}\right)\|\partial_s\Lambda(s,0)\| + \cosh\left(\sqrt{L_R\|\partial_t\Lambda(s,0)\|^2}\right)\|D_t\partial_s\Lambda(s,0)\|$$

$$\leq\sqrt{L_R}(\|u\| + \|v\|)\sinh\left(\sqrt{L_R}(\|u\| + \|v\|)\right)\|\gamma'(0)\| + \cosh\left(\sqrt{L_R}(\|u\| + \|v\|)\right)\|v(0) - u(0)\|$$

$$=\mathcal{C}\sinh\left(\mathcal{C}\right)\|\gamma'(0)\| + \cosh\left(\mathcal{C}\right)\|v(0) - u(0)\|$$

for $0 \leq t \leq 1$, where we usse the fact that $\cosh(r)$ and $\frac{\sinh(r)}{r}$ are monotonically increasing in $r$. Put together, the third term of (D.2) is bounded as

$$\left|\int_0^1 2\left\langle\partial_s\Lambda(s,t) - \{\partial_s\Lambda(s,0)\}^{\to\Lambda(s,t)}, D_t\partial_s\Lambda(s,t)\right\rangle ds\right|$$

$$\leq 2((\cosh\left(\mathcal{C}\right) - 1)\|\gamma'(0)\| + (\frac{\sinh\left(\mathcal{C}\right)}{\mathcal{C}})\|v(0) - u(0)\|)\cdot(\mathcal{C}\sinh\left(\mathcal{C}\right)\|\gamma'(0)\| + \cosh\left(\mathcal{C}\right)\|v(0) - u(0)\|)$$

$$\leq 8\|v(0) - u(0)\|^2(\cosh\left(\mathcal{C}\right)^2 + \frac{\sinh\left(\mathcal{C}\right)^2}{\mathcal{C}^2}) + 8\|\gamma'(0)\|^2((\cosh\left(\mathcal{C}\right) - 1)^2 + \mathcal{C}^2\sinh\left(\mathcal{C}\right)^2)$$

Put together, we get

$$\left|\frac{d}{dt}E(\gamma_t) - 2\left\langle\gamma'(0), v(0) - u(0)\right\rangle\right|$$

$$\leq 8\|v(0) - u(0)\|^2(\cosh\left(\mathcal{C}\right)^2 + \frac{\sinh\left(\mathcal{C}\right)^2}{\mathcal{C}^2}) + 8\|\gamma'(0)\|^2((\cosh\left(\mathcal{C}\right) - 1)^2 + \mathcal{C}^2\sinh\left(\mathcal{C}\right)^2)$$

$$+ 4\|v(0) - u(0)\|^2 + 4\|\gamma'(0)\|^2(\mathcal{C}\sinh\left(\mathcal{C}\right) + (\cosh\left(\mathcal{C}\right) - 1)^2)$$

$$\leq 8\|v(0) - u(0)\|^2(\cosh\left(\mathcal{C}\right)^2 + \frac{\sinh\left(\mathcal{C}\right)^2}{\mathcal{C}^2} + 1)$$

$$+ 8\|\gamma'(0)\|^2(2(\cosh\left(\mathcal{C}\right) - 1)^2 + \mathcal{C}^2\sinh\left(\mathcal{C}\right)^2 + \mathcal{C}\sinh\left(\mathcal{C}\right))$$

Integrating for $t =\in [0, 1]$, and noting that $E(\gamma_0) = \|\gamma'(0)\|$,

$$E(\gamma_1) \leq(1 + 8(2(\cosh\left(\mathcal{C}\right) - 1)^2 + \mathcal{C}^2\sinh\left(\mathcal{C}\right)^2 + \mathcal{C}\sinh\left(\mathcal{C}\right)))E(\gamma_0)$$

$$+ 8\|v(0) - u(0)\|^2(\cosh\left(\mathcal{C}\right)^2 + \frac{\sinh\left(\mathcal{C}\right)^2}{\mathcal{C}^2} + 1)$$

$$+ 2\left\langle\gamma'(0), v(0) - u(0)\right\rangle$$

From Lemma 39, we can upper bound

$$8(2(\cosh\left(\mathcal{C}\right) - 1)^2 + \mathcal{C}^2\sinh\left(\mathcal{C}\right)^2 + \mathcal{C}\sinh\left(\mathcal{C}\right)) \leq 8r^4e^{2r} + 8r^4e^{2r} + r^2e^r$$

$$\leq 4r^2e^{4r}$$

$$\cosh\left(\mathcal{C}\right)^2 + \frac{\sinh\left(\mathcal{C}\right)^2}{\mathcal{C}^2} + 1 \leq 4e^{2r}$$

where we use the fact that $r^2 \leq e^{2r}/6$ for all $r \geq 0$.

The conclusion follows by noting that $\mathrm{d}(x, y) = \sqrt{E(\gamma_0)}$ and $\mathrm{d}(\mathrm{Exp}_x(u), \mathrm{Exp}_y(v)) \leq \sqrt{E(\gamma_1)}$.

$\square$

**Lemma 29.** *Let $x, y \in M$. Let $\gamma(s) : [0, 1] \to M$ be a minimizing geodesic between $x$ and $y$ with $\gamma(0) = x$ and $\gamma(1) = y$. Let $u \in T_x M$ and $v \in T_y M$. Let $u(s)$ and $v(s)$ be the parallel transport of $u$ and $v$ along $\gamma$. Let $u = u_1 + u_2$ and $v = v_1 + v_2$ be a decomposition such that $v_2 = \Gamma_x^y u_2$, where the parallel transport is along $\gamma(s)$.*

*Let us define $u_1(s), u_2(s), v_1(s)$, all mapping from $[0, 1] \to T_{\gamma(s)} M$, such that they are the parallel transport of $u_1, u_2, v_1$ along $\gamma(s)$ respectively $(u_1(0) = u_1, u_2(0) = u_2, v_1(1) = v_1, u_2(1) = v_2)$*

*Then*

$$
\begin{aligned}
&\mathrm{d}(\mathrm{Exp}_x(u), \mathrm{Exp}_y(v))^2 - \mathrm{d}(x, y)^2 \\
&\leq 2 \langle \gamma'(0), v(0) - u(0) \rangle + \|v(0) - u(0)\|^2 \\
&\quad - \int_0^1 \langle R(\gamma'(s), (1 - s)u(s) + sv(s))(1 - s)u(s) + sv(s), \gamma'(s) \rangle \, ds \\
&\quad + (2\mathcal{C}^2 e^{\mathcal{C}} + 18\mathcal{C}^4 e^{2\mathcal{C}})\|v(0) - u(0)\|^2 + (18\mathcal{C}^4 e^{2\mathcal{C}} + 4\mathcal{C}')\mathrm{d}(x, y)^2 + 4\mathcal{C}^2 e^{2\mathcal{C}} \mathrm{d}(x, y)\|v(0) - u(0)\|
\end{aligned}
$$

*where $\mathcal{C} := \sqrt{L_R}(\|u\| + \|v\|)$ and $\mathcal{C}' := L_R'(\|u\| + \|v\|)^3$.*

*Proof.* The proof is similar to Lemma 28. Let us consider the length function $E(\gamma) = \int_0^1 \|\gamma'(s)\|^2 ds$. We define a variation of geodesics $\Lambda(s, t)$:

$$
\Lambda(s, t) := \mathrm{Exp}_{\gamma(s)} \left( t(u(s) + s(v(s) - u(s))) \right)
$$

We verify that

$$
\begin{aligned}
\partial_s \Lambda(s, 0) &= \gamma'(s) \\
\partial_t \Lambda(s, 0) &= u(s) + s(v(s) - u(s)) \\
D_t \partial_s \Lambda(s, 0) &= v(s) - u(s)
\end{aligned}
$$

Consider a fixed $t$, and let $\gamma_t(s) := \Lambda(s, t)$ (so $\gamma_t'(s)$ is the velocity wrt $s$).

$$
\begin{aligned}
&\frac{d}{dt} E(\gamma_t) \\
&= \frac{d}{dt} \int_0^1 \|\gamma_t'(s)\|^2 ds \\
&= \int_0^1 2 \langle \gamma_t'(s), D_t \gamma_t'(s) \rangle \, ds \\
&= \int_0^1 2 \langle \partial_s \Lambda(s, t), D_t \partial_s \Lambda(s, t) \rangle \, ds
\end{aligned}
$$

and

$$\frac{d^2}{dt^2}E(\gamma_t)$$

$$= \int_0^1 2\left\langle D_t\partial_s\Lambda(s,t), D_t\partial_s\Lambda(s,t)\right\rangle ds + \int_0^1 2\left\langle \partial_s\Lambda(s,t), D_t D_t\partial_s\Lambda(s,t)\right\rangle ds$$

$$= \int_0^1 2\|D_t\partial_s\Lambda(s,t)\|^2 ds - \int_0^1 2\left\langle R(\partial_s(\Lambda(s,t)), \partial_t\Lambda(s,t))\partial_t\Lambda(s,t), \partial_s\Lambda(s,t)\right\rangle ds$$

$$= \int_0^1 2\|D_t\partial_s\Lambda(s,t)\|^2 ds - \int_0^1 2\left\langle R(\partial_s(\Lambda(s,t)), \Gamma^{\Lambda(s,t)}_{\Lambda(s,0)}\partial_t\Lambda(s,0))\Gamma^{\Lambda(s,t)}_{\Lambda(s,0)}\partial_t\Lambda(s,t), \partial_s\Lambda(s,t)\right\rangle ds$$

$$\leq \int_0^1 2\|D_t\partial_s\Lambda(s,t)\|^2 ds - \int_0^1 2\left\langle R(\partial_s(\Lambda(s,0)), \partial_t\Lambda(s,0))\partial_t\Lambda(s,t), \partial_s\Lambda(s,0)\right\rangle ds$$

$$+ \int_0^1 4L_R\|\partial_t\Lambda(s,0)\|^2\|\partial_s\Lambda(s,t)\|\left\|\partial_s\Lambda(s,t) - \Gamma^{\Lambda(s,t)}_{\Lambda(s,0)}\partial_s\Lambda(s,0)\right\| + 4L'_R\|\partial_t\Lambda(s,0)\|^3\|\partial_s\Lambda(s,0)\|^2 ds$$

where the second equality uses the Jacobi equation.

The Riemannian curvature tensor term can be simplified as

$$-\int_0^1 2\left\langle R(\partial_s(\Lambda(s,0)), \partial_t\Lambda(s,0))\partial_t\Lambda(s,t), \partial_s\Lambda(s,0)\right\rangle ds$$

$$= -2\int_0^1 \left\langle R(\gamma'(s), (1-s)u(s)+sv(s))(1-s)u(s)+sv(s), \gamma'(s)\right\rangle ds$$

We further bound

$$\int_0^1 2\|D_t\partial_s\Lambda(s,t)\|^2 ds$$

$$\leq \int_0^1 2(\mathcal{C}\sinh(\mathcal{C})\|\partial_s\Lambda(s,0)\| + \cosh(\mathcal{C})\|D_t\partial_s\Lambda(s,0)\|)^2 ds$$

$$\leq 2\cosh(\mathcal{C})^2\int_0^1 \|D_t\partial_s\Lambda(s,0)\|^2 ds$$

$$+ 2\mathcal{C}^2\sinh(\mathcal{C})^2\int_0^1 \|\partial_s\Lambda(s,0)\|^2 ds + 4\mathcal{C}\int_0^1 \sinh(\mathcal{C})\cosh(\mathcal{C})\|\partial_s\Lambda(s,0)\|\|D_t\partial_s\Lambda(s,0)\| ds$$

$$= 2\cosh(\mathcal{C})^2\|v(0)-u(0)\|^2$$

$$+ 2\mathcal{C}^2\sinh(\mathcal{C})^2 \mathrm{d}(x,y)^2 + 4\mathcal{C}\int_0^1 \sinh(\mathcal{C})\cosh(\mathcal{C})\|\partial_s\Lambda(s,0)\|\|D_t\partial_s\Lambda(s,0)\| ds$$

$$\leq 2\|v(0)-u(0)\|^2 + 2(\mathcal{C}^2 e^{\mathcal{C}} + \mathcal{C}^4 e^{2\mathcal{C}})\|v(0)-u(0)\|^2$$

$$+ 2\mathcal{C}^4 e^{2\mathcal{C}}\mathrm{d}(x,y)^2 + 4\mathcal{C}^2 e^{2\mathcal{C}}\mathrm{d}(x,y)\|v(0)-u(0)\|$$

where we use Lemma 26 and Lemma 39.

We also bound

$$\int_0^1 4L_R\|\partial_t\Lambda(s,0)\|^2\|\partial_s\Lambda(s,t)\|\left\|\partial_s\Lambda(s,t) - \Gamma^{\Lambda(s,t)}_{\Lambda(s,0)}\partial_s\Lambda(s,0)\right\| ds$$

$$\leq 4\mathcal{C}^2(\cosh(\mathcal{C})\mathrm{d}(x,y) + \frac{\sinh(\mathcal{C})}{\mathcal{C}}\|u(0)-v(0)\|)((\cosh(\mathcal{C})-1)\mathrm{d}(x,y) + (\frac{\sinh(\mathcal{C})}{\mathcal{C}}-1)\|u(0)-v(0)\|)$$

$$\leq 16\mathcal{C}^4 e^{2\mathcal{C}}\mathrm{d}(x,y)^2 + 16\mathcal{C}^4 e^{2\mathcal{C}}\|u(0)-v(0)\|^2$$

where we use Lemma 26 and Lemma 39.

We finally bound

$$\int_0^1 4L'_R\|\partial_t\Lambda(s,0)\|^3\|\partial_s\Lambda(s,0)\|^2 ds \leq 4\mathcal{C}'\mathrm{d}(x,y)^2$$

by definition of $\mathcal{C}'$.

Combining the above bounds,

$$
\begin{aligned}
E(\gamma_1) &= \frac{d}{dt}E(\gamma_t)\bigg|_{t=0} + \int_0^1 \int_0^r \frac{d^2}{dt^2}E(\gamma_t)dtdr \\
&\leq\ 2\left\langle \gamma'(0), v(0) - u(0)\right\rangle + \|v(0) - u(0)\|^2 \\
&\quad - \int_0^1 \left\langle R(\gamma'(s), (1-s)u(s) + sv(s))(1-s)u(s) + sv(s), \gamma'(s)\right\rangle ds \\
&\quad + 2(\mathcal{C}^2 e^{\mathcal{C}} + \mathcal{C}^4 e^{2\mathcal{C}})\|v(0) - u(0)\|^2 + 2\mathcal{C}^4 e^{2\mathcal{C}}\mathrm{d}(x,y)^2 + 4\mathcal{C}^2 e^{2\mathcal{C}}\mathrm{d}(x,y)\|v(0) - u(0)\| \\
&\quad + 16\mathcal{C}^4 e^{2\mathcal{C}}\mathrm{d}(x,y)^2 + 16\mathcal{C}^4 e^{2\mathcal{C}}\|u(0) - v(0)\|^2 + 4\mathcal{C}'\mathrm{d}(x,y)^2 \\
&=\ 2\left\langle \gamma'(0), v(0) - u(0)\right\rangle + \|v(0) - u(0)\|^2 \\
&\quad - \int_0^1 \left\langle R(\gamma'(s), (1-s)u(s) + sv(s))(1-s)u(s) + sv(s), \gamma'(s)\right\rangle ds \\
&\quad + (2\mathcal{C}^2 e^{\mathcal{C}} + 18\mathcal{C}^4 e^{2\mathcal{C}})\|v(0) - u(0)\|^2 + (18\mathcal{C}^4 e^{2\mathcal{C}} + 4\mathcal{C}')\mathrm{d}(x,y)^2 + 4\mathcal{C}^2 e^{2\mathcal{C}}\mathrm{d}(x,y)\|v(0) - u(0)\|
\end{aligned}
$$

Our conclusion follows as $\mathrm{d}(\mathrm{Exp}_x(u), \mathrm{Exp}_y(v))^2 \leq E(\gamma_1)$. $\qquad\square$

**Lemma 30.** *Let* $a_t, b_t : t \to \mathbb{R}^+$ *satisfy*

$$
\frac{d}{dt}a_t = b_t
$$
$$
\frac{d}{dt}b_t \leq Ca_t
$$

*with initial conditions* $a_0, b_0$, *then for all* $t$,

$$
a_t \leq a_0 \cosh\left(\sqrt{C}t\right) + \frac{b_0}{\sqrt{C}} \sinh\left(\sqrt{C}t\right)
$$

$$
b_t \leq \sqrt{C}\left(a_0 \sinh\left(\sqrt{C}t\right) + \frac{b_0}{\sqrt{C}} \cosh\left(\sqrt{C}t\right)\right)
$$

*Proof.* Let $x_t := a_0 \cosh\left(\sqrt{C}t\right) + \frac{b_0}{\sqrt{C}} \sinh\left(\sqrt{C}t\right)$ and $y_t := \sqrt{C}\left(a_0 \sinh\left(\sqrt{C}t\right) + \frac{b_0}{\sqrt{C}} \cosh\left(\sqrt{C}t\right)\right)$. We verify that

$$
\frac{d}{dt}x_t = \sqrt{C}\left(a_0 \sinh\left(\sqrt{C}t\right) + \frac{b_0}{\sqrt{C}} \cosh\left(\sqrt{C}t\right)\right) = y_t
$$

$$
\frac{d}{dt}y_t = C\left(a_0 \cosh\left(\sqrt{C}t\right) + \frac{b_0}{\sqrt{C}} \sinh\left(\sqrt{C}t\right)\right) = Cx_t
$$

We further verify the initial conditions. Note that $\sinh(0) = 0$ and $cosh(0) = 1$. Thus

$$
x_0 = a_0
$$
$$
y_0 = b_0
$$

Finally, we verify that $a_t \leq x_t$ and $b_t \leq y_t$ for all $t$:

$$
\frac{d}{dt}x_t - a_t = y_t - b_t
$$
$$
\frac{d}{dt}y_t - b_t \geq C(x_t - a_t)
$$

$\qquad\square$

**Lemma 31.** *Let* $a_t, b_t : t \to \mathbb{R}^+$ *satisfy*

$$
\frac{d}{dt}a_t = b_t
$$
$$
\frac{d}{dt}b_t \leq Ca_t + Dt + E
$$

*with initial conditions $a_0 = 0, b_0 = 0$, then for all $t$,*

$$a_t \leq \frac{E}{C} \cosh(\sqrt{C}t) + \frac{D}{C^{3/2}} \sinh\left(\sqrt{C}t\right) - \frac{D}{C}t - \frac{E}{C}$$

$$b_t \leq \frac{E}{\sqrt{C}} \sinh(\sqrt{C}t) + \frac{D}{C} \cosh\left(\sqrt{C}t\right) - \frac{D}{C}$$

*Proof.* Let $x_t := \frac{E}{C} \cosh(\sqrt{C}t) + \frac{D}{C^{3/2}} \sinh\left(\sqrt{C}t\right) - \frac{D}{C}t - \frac{E}{C}$ and
$y_t := \frac{E}{\sqrt{C}} \sinh(\sqrt{C}t) + \frac{D}{C} \cosh\left(\sqrt{C}t\right) - \frac{D}{C}$.

We verify that

$$\frac{d}{dt}x_t = \frac{E}{\sqrt{C}} \sinh(\sqrt{C}t) + \frac{D}{C} \cosh\left(\sqrt{C}t\right) - \frac{D}{C} = y_t$$

$$\frac{d}{dt}y_t = E \cosh(\sqrt{C}t) + \frac{D}{\sqrt{C}} \sinh\left(\sqrt{C}t\right) = Cx_t + Dt + E$$

we also verify the initial conditions that $x_0 = 0$ and $y_0 = 0$.

$\square$

**Lemma 32.** *Let $x, y \in M$, and let $E_1...E_d$ be an orthonormal basis at $T_xM$. Let $v \in T_xM$ be a random vector with $\mathbb{E}\left[\langle E_i, v\rangle \langle E_j, v\rangle\right] = \mathbb{1}\{i = j\}$. Let $\gamma : [0,1] \to M$ be any smooth path between $x$ and $y$. Let $v(t)$ be the parallel transport of $v$ along $\gamma$. Then for any basis $E'_1...E'_d$ at $T_yM$,*

$$\mathbb{E}\left[\langle v(t), E'_i\rangle \langle v(t), E'_j\rangle\right] = \mathbb{1}\{i = j\}$$

*In other words, if $v$ has identity covariance, then the parallel transport of $v$ has identity covariance.*

*Proof.* Let $E_i(t)$ be an orthonormal frame along $\gamma$ with $E_i(0) = E_i$. Under parallel transport, $\frac{d}{dt}\langle v(t), E_i(t)\rangle = 0$. Thus for all $t$,

$$\mathbb{E}\left[\langle E_i(t), v\rangle \langle E_j(t), v\rangle\right] = \mathbb{E}\left[\langle E_i(0), v\rangle \langle E_j(0), v\rangle\right] = \mathbb{1}\{i = j\}$$

Finally, consider any basis $E'_i$. Let $E'_i = \sum_k \alpha^i_k E_i(1)$, i.e. $\alpha^i_k = \langle E'_i, E_k(1)\rangle$ Then

$$\langle v, E'_i\rangle \langle v, E'_j\rangle$$
$$= \sum_{k,l} \alpha^i_j \alpha^j_l \langle v, E_k\rangle \langle v, E_l\rangle$$
$$= \sum_{k,l} \alpha^i_j \alpha^j_l \mathbb{1}\{k = l\}$$
$$= \sum_k \alpha^i_k \alpha^j_k$$
$$= \langle E'_i, E'_j\rangle$$
$$= \mathbb{1}\{i = j\}$$

$\square$

**Lemma 33.** *Let $x, y \in M$, and let $E_1...E_d$ be an orthonormal basis at $T_xM$.*

*Let $\alpha$ denote a spherically symmetric random variable in $\mathbb{R}^d$, i.e. for any orthogonal matrix $G \in \mathbb{R}^{d \times d}$*

$$\alpha \stackrel{d}{=} G\alpha$$

*Then for any $x \in M$, let $E_1...E_d$ and $E'_1...E'_d$ be two sets of orthonormal bases of $T_xM$. then*

$$\sum_{i=1}^d \alpha_i E_i \stackrel{d}{=} \sum_{i=1}^d \alpha_i E'_i$$

Consequently, let $v \in T_x M := \sum_{i=1}^{d} \alpha_i E_i$. Let $\gamma : [0, 1] \to M$ be any differentiable path between $x$ and $y$. Let $v(t)$ be the parallel transport of $v$ along $\gamma$. Then for any orthogonal basis $E'_1...E'_d$ at $T_y M$,

$$v(1) \stackrel{d}{=} \sum_{i=1}^{d} \alpha_i E'_i$$

*Proof.* First, we verify that if $\alpha$ is spherically symmetric, and $E_1..E_d$, $E'_1...E'_d$ are two sets of orthonormal basis at some point $x$, then

$$\sum_{i=1}^{d} \alpha_i E_i \stackrel{d}{=} \sum_{i=1}^{d} \alpha_i E'_i$$

To see this, notice that there exists an orthogonal matrix $G$, with $G_{i,j} = \langle E_i, E'_j \rangle$, such that

$$E_i = \sum_{j=1}^{d} G_{j,i} E'_j$$

We further verify that $G_{i,j}$ is orthogonal. It suffices to verify that $GG^T = I$.

$$\mathbb{1}\{j = k\} = \langle E'_i, E'_j \rangle = \left\langle \sum_{k=1}^{d} \langle E'_i, E_k \rangle E_j, \sum_{\ell=1}^{d} \langle E'_j, E_\ell \rangle E_\ell \right\rangle$$

$$= \sum_{k,\ell} \langle E'_i, E_k \rangle \langle E'_j, E_\ell \rangle \langle E_j, E_\ell \rangle$$

$$= \sum_{k} \langle E'_i, E_k \rangle \langle E'_j, E_k \rangle$$

$$= \langle G_{i,\cdot}, G_{j,\cdot} \rangle$$

Note that the inner product on the last line is dot product over $\mathbb{R}^d$, and the inner product on preceding lines are over $T_x M$. The above implies that

$$GG^T = I$$

i.e. $G$ is orthogonal.

Now consider any arbitrary function $f : T_x M \to \mathbb{R}$, then

$$\mathbb{E}\left[f(\sum_i \alpha_i E_i)\right] = \mathbb{E}\left[f(\sum_i \sum_j \alpha_i G_{i,j} E'_j)\right]$$

$$:= \mathbb{E}\left[f(\sum_j \beta_j E'_j)\right]$$

where we defined $\beta_j := \sum_i \alpha_i G_{i,j}$. We finally verify that $\beta \stackrel{d}{=} \alpha$. This follows from the fact that $\beta = G^T \alpha$, where $G$ is an orthogonal matrix, and the definition of spherical symmetry for $\alpha$.

Consider an arbitrary line $\gamma(t) : [0, 1] \to M$, with $x := \gamma(0)$, $y := \gamma(1)$. Let $E_i$ be an orthonormal basis at $T_x M$, and $E_i(t)$ be an orthonormal basis at $T_{\gamma(t)} M$ obtained from parallel transport of $E_i$. This proves the first claim.

To verify the second claim, let $v \in T_x M$ be a random vector, given by

$$v = \sum_{i=1}^{d} \alpha_i E_i$$

where $\alpha$ is some spherically random vector in $\mathbb{R}^d$. Let $v(t)$ be the parallel transport of $v$ along $\gamma$. Let $\alpha(t) := \langle v(t), E_i(t) \rangle$. Then by definition of parallel transport, for all $i$,

$$\frac{d}{dt} \langle v(t), E_i(t) \rangle = 0$$

so that for all $t \in [0, 1]$,

$$\alpha(t) := \alpha$$

the second claim then follows from the first claim.

$\square$

# E    Matrix ODE

In this section we provide Gronwall-style inequality for matrix ODE. The results in this section are necessary for analyzing Jacobi Equation, whose coordinates with respect to some orthonormal frame can be viewed as an ODE in $\mathbb{R}^d$. In particular, Lemma 28 and Lemma 29 rely on results in this section.

**Lemma 34** (Formal Matrix Exponent). *Given* $\mathbf{M}(t) : \mathbb{R}^+ \to \mathbb{R}^{d \times d}$, *define* $\exp_{\mathbf{mat}}(t; \mathbf{M}) : \mathbb{R}^+ \to \mathbb{R}^{d \times d}$ *as the solution to the matrix ODE*

$$\exp_{\mathbf{mat}}(0; \mathbf{M}) = I$$
$$\frac{d}{dt} \exp_{\mathbf{mat}}(t; \mathbf{M}) = \mathbf{M}(t) \exp_{\mathbf{mat}}(t; \mathbf{M})$$

*Then*

1. *Let* $\mathbf{x}(t)$ *be the solution to the ODE* $\frac{d}{dt}\mathbf{x}(t) = \mathbf{M}(t)\mathbf{x}(t)$, *for some* $\mathbf{M}$, *then*

$$\mathbf{x}(t) = \exp_{\mathbf{mat}}(t; \mathbf{M})\mathbf{x}(0)$$

2. *Let* $\mathbf{z}(t)$ *be the solution to* $\frac{d}{dt}\mathbf{z}(t) = \mathbf{M}(t)\mathbf{z}(t) + \mathbf{v}(t)$, *for some* $\mathbf{M}$, $\mathbf{v}$, *then*

$$\mathbf{z}(T) = \int_0^T \exp_{\mathbf{mat}}(T - s; \mathbf{N}_s)\mathbf{v}(s)ds + \exp_{\mathbf{mat}}(T; \mathbf{M})\mathbf{z}(0)$$

*where for any* $s, t$, $\mathbf{N}_s(t) := \mathbf{M}(s + t)$.

*Proof of Lemma 34.* Let $\mathbf{y}_t := \exp_{\mathbf{mat}}(t; \mathbf{M})\mathbf{x}(0)$. We verify that

$$\mathbf{y}(0) = 0$$
$$\frac{d}{dt}\mathbf{y}(t) = (\frac{d}{dt} \exp_{\mathbf{mat}}(t; \mathbf{M}))\mathbf{x}(0) = \mathbf{M}(t)\mathbf{y}(t)$$

Given the same dynamics and initial conditions, we conclude that $\mathbf{x}(t) = \mathbf{y}(t)$ for all $t$.

To verify the second claim, note that

$$\frac{d}{dt} \int_0^t \exp_{\mathbf{mat}}(t - s; \mathbf{N}_s)\mathbf{v}(s)ds$$

$$= \exp_{\mathbf{mat}}(0; \mathbf{N}_t)\mathbf{v}(s) + \int_0^t (\frac{d}{dt} \exp_{\mathbf{mat}}(t - s; \mathbf{N}_s))\mathbf{v}(s)ds$$

$$= \mathbf{v}(s) + \int_0^t \mathbf{N}_s(t - s) \exp_{\mathbf{mat}}(t - s; \mathbf{N}_s)\mathbf{v}(s)dt$$

$$= \mathbf{v}(s) + \mathbf{M}(t) \int_0^t \exp_{\mathbf{mat}}(t - s; \mathbf{N}_s)\mathbf{v}(s)ds$$

Additionally, $\frac{d}{dt}\mathbf{exp_{mat}}(t; \mathbf{M})\mathbf{z}(0) = \mathbf{M}(t)\mathbf{exp_{mat}}(t; \mathbf{M})\mathbf{z}(0)$, summing,

$$\frac{d}{dt}\mathbf{z}(t) = \frac{d}{dt}\int_0^t \mathbf{exp_{mat}}(t-s; \mathbf{N}_s)\mathbf{v}(s)ds + \mathbf{exp_{mat}}(t; \mathbf{M})\mathbf{z}(0)$$

$$= \mathbf{v}(s) + \mathbf{M}(t)\int_0^t \mathbf{exp_{mat}}(t-s; \mathbf{N}_s)\mathbf{v}(s)ds + \mathbf{M}(t)\mathbf{exp_{mat}}(t; \mathbf{M})\mathbf{z}(0)$$

$$= \mathbf{v}(s) + \mathbf{M}(t)\mathbf{z}(t)$$

$\square$

**Lemma 35.** *Let* $\mathbf{exp_{mat}}$ *be as defined in Lemma 34. Let*

$$\begin{bmatrix} \mathbf{A}(t) & \mathbf{B}(t) \\ \mathbf{C}(t) & \mathbf{D}(t) \end{bmatrix} := \mathbf{exp_{mat}}\left(t; \begin{bmatrix} 0 & I \\ \mathbf{M}(t) & 0 \end{bmatrix}\right)$$

*for some* $\mathbf{M}(t)$. *Assume* $\|\mathbf{M}(t)\|_2 \le L_{\mathbf{M}}$ *for all t. Then for all t,*

$$\|\mathbf{A}(t)\|_2 \le \cosh\left(\sqrt{L_{\mathbf{M}}}t\right)$$

$$\|\mathbf{B}(t)\|_2 \le \frac{1}{\sqrt{L_{\mathbf{M}}}}\sinh\left(\sqrt{L_{\mathbf{M}}}t\right)$$

$$\|\mathbf{C}(t)\|_2 \le \sqrt{L_{\mathbf{M}}}\sinh\left(\sqrt{L_{\mathbf{M}}}t\right)$$

$$\|\mathbf{D}(t)\|_2 \le \cosh\left(\sqrt{L_{\mathbf{M}}}t\right)$$

*and*

$$\|\mathbf{A}(t) - I\|_2 \le \cosh(\sqrt{L_{\mathbf{M}}}t) - 1$$

$$\|\mathbf{B}(t) - tI\|_2 \le \frac{1}{\sqrt{L_{\mathbf{M}}}}\sinh\left(\sqrt{L_{\mathbf{M}}}t\right) - t$$

$$\|\mathbf{D}(t) - I\|_2 \le \cosh\left(\sqrt{L_{\mathbf{M}}}t\right) - 1$$

$$\|\mathbf{A}(t) - I\|_2 \le \frac{1}{2}L_{\mathbf{M}}e^{L_{\mathbf{M}}}$$

$$\|\mathbf{B}(t) - tI\|_2 \le \frac{1}{6}L_{\mathbf{M}}e^{L_{\mathbf{M}}}$$

$$\|\mathbf{D}(t) - I\|_2 \le \frac{1}{2}L_{\mathbf{M}}e^{L_{\mathbf{M}}}$$

$$\|\mathbf{C}(t)\|_2 \le L_{\mathbf{M}}e^{L_{\mathbf{M}}}$$

*Proof of Lemma 35.* We first verify the first part of the lemma. Consider the ODE given by

$$\frac{d}{dt}\begin{bmatrix} \mathbf{x} \\ \mathbf{y} \end{bmatrix}(t) = \begin{bmatrix} 0 & I \\ \mathbf{M}(t) & 0 \end{bmatrix}\begin{bmatrix} \mathbf{x}(t) \\ \mathbf{y}(t) \end{bmatrix}$$

By Lemma 34, $\begin{bmatrix} \mathbf{A}(t) & \mathbf{B}(t) \\ \mathbf{C}(t) & \mathbf{D}(t) \end{bmatrix}$ satisfies

$$\begin{bmatrix} \mathbf{x}(t) \\ \mathbf{y}(t) \end{bmatrix} = \begin{bmatrix} \mathbf{A}(t) & \mathbf{B}(t) \\ \mathbf{C}(t) & \mathbf{D}(t) \end{bmatrix}\begin{bmatrix} \mathbf{x}(0) \\ \mathbf{y}(0) \end{bmatrix}$$

By Cauchy Schwarz,

$$\frac{d}{dt}\|\mathbf{x}(t)\|_2 \le \|\mathbf{y}(t)\|_2$$

$$\frac{d}{dt}\|\mathbf{y}(t)\|_2 \le L_{\mathbf{M}}\|\mathbf{x}(t)\|_2$$

We apply Lemma 30, with $a_t := \|\mathbf{x}(t)\|_2$ and $b_t := \|\mathbf{y}(t)\|_2$, $C := L_{\mathbf{M}}$. Then

$$\|\mathbf{x}_t\|_2 \le \|\mathbf{x}_0\|_2 \cosh\left(\sqrt{L_{\mathbf{M}}}t\right) + \frac{\|\mathbf{y}_0\|_2}{\sqrt{L_{\mathbf{M}}}}\sinh\left(\sqrt{L_{\mathbf{M}}}t\right)$$

$$\|\mathbf{y}_t\|_2 \le \sqrt{L_{\mathbf{M}}}\left(\|\mathbf{x}_0\|_2 \sinh\left(\sqrt{L_{\mathbf{M}}}t\right) + \frac{\|\mathbf{y}_0\|_2}{\sqrt{L_{\mathbf{M}}}}\cosh\left(\sqrt{L_{\mathbf{M}}}t\right)\right)$$

This immediately implies that

$$\|\mathbf{A}(t)\|_2 \le \cosh\left(\sqrt{L_{\mathbf{M}}}t\right)$$

$$\|\mathbf{B}(t)\|_2 \le \frac{1}{\sqrt{L_{\mathbf{M}}}}\sinh\left(\sqrt{L_{\mathbf{M}}}t\right)$$

$$\|\mathbf{C}(t)\|_2 \le \sqrt{L_{\mathbf{M}}}\sinh\left(\sqrt{L_{\mathbf{M}}}t\right)$$

$$\|\mathbf{D}(t)\|_2 \le \cosh\left(\sqrt{L_{\mathbf{M}}}t\right)$$

This proves the first claim of the Lemma.

We now prove the second claim. We verify that

$$\frac{d}{dt}\begin{bmatrix} \mathbf{x}(t) - \mathbf{x}(0) - t\mathbf{y}(0) \\ \mathbf{y}(t) - \mathbf{y}(0) \end{bmatrix} = \begin{bmatrix} \mathbf{y}(t) - \mathbf{y}(0) \\ \mathbf{M}(t)\mathbf{x}(t) \end{bmatrix}$$

$$\frac{d}{dt}\begin{bmatrix} \mathbf{x}(t) - \mathbf{x}(0) - t\mathbf{y}(0) \\ \mathbf{y}(t) - \mathbf{y}(0) \end{bmatrix} = \begin{bmatrix} \mathbf{y}(t) - \mathbf{y}(0) \\ \mathbf{M}(t)\mathbf{x}(t) \end{bmatrix} = \begin{bmatrix} \mathbf{y}(t) - \mathbf{y}(0) \\ \mathbf{M}(t)\mathbf{x}(t) \end{bmatrix}$$

$$= \begin{bmatrix} \mathbf{y}(t) - \mathbf{y}(0) \\ \mathbf{M}(t)(\mathbf{x}(t) - \mathbf{x}(0) - t\mathbf{y}(0)) \end{bmatrix} + \begin{bmatrix} 0 \\ \mathbf{M}(t)(\mathbf{x}(0) + t\mathbf{y}(0)) \end{bmatrix}$$

Thus

$$\frac{d}{dt}\|\mathbf{x}(t) - \mathbf{x}(0) - t\mathbf{y}(0)\|_2 \le \|\mathbf{y}(t) - \mathbf{y}(0)\|_2$$

$$\frac{d}{dt}\|\mathbf{y}(t) - \mathbf{y}(0)\|_2 \le L_{\mathbf{M}}\|\mathbf{x}(t) - \mathbf{x}(0) - t\mathbf{y}(0)\|_2 + L_{\mathbf{M}}(\|\mathbf{x}(0)\|_2 + t\|\mathbf{y}(0)\|_2)$$

We verify that

$$\|\mathbf{y}(t) - \mathbf{y}(0)\|_2 \le L_{\mathbf{M}}\int_0^t \|\mathbf{x}(s) - s\mathbf{y}(0)\|_2 ds + \frac{t^2}{2}L_{\mathbf{M}}$$

Let us apply Lemma 31 with $a_t = \|\mathbf{x}_t - \mathbf{x}(0) - t\mathbf{y}(0)\|_2$, $b_t = \|\mathbf{y}(t) - \mathbf{y}(0)\|_2$, $C = L_{\mathbf{M}}$, $D = L_{\mathbf{M}}\|\mathbf{y}(0)\|_2$ and $E = L_{\mathbf{M}}\|\mathbf{x}(0)\|_2$

$$\|\mathbf{x}_t - \mathbf{x}(0) - t\mathbf{y}(0)\|_2 \le \frac{L_{\mathbf{M}}\|\mathbf{x}(0)\|_2}{L_{\mathbf{M}}}\cosh(\sqrt{L_{\mathbf{M}}}t) + \frac{L_{\mathbf{M}}\|\mathbf{y}(0)\|_2}{L_{\mathbf{M}}^{3/2}}\sinh\left(\sqrt{L_{\mathbf{M}}}t\right)$$

$$- \frac{L_{\mathbf{M}}\|\mathbf{y}(0)\|_2}{L_{\mathbf{M}}}t - \frac{L_{\mathbf{M}}\|\mathbf{x}(0)\|_2}{L_{\mathbf{M}}}$$

$$= \|\mathbf{x}(0)\|_2(\cosh(\sqrt{L_{\mathbf{M}}}t) - 1) + \|\mathbf{y}(0)\|_2\left(\frac{1}{\sqrt{L_{\mathbf{M}}}}\sinh\left(\sqrt{L_{\mathbf{M}}}t\right) - t\right)$$

$$\|\mathbf{y}(t) - \mathbf{y}(0)\|_2 \le \frac{L_{\mathbf{M}}\|\mathbf{x}(0)\|_2}{\sqrt{L_{\mathbf{M}}}}\sinh(\sqrt{L_{\mathbf{M}}}t) + \frac{L_{\mathbf{M}}\|\mathbf{y}(0)\|_2}{L_{\mathbf{M}}}\cosh\left(\sqrt{L_{\mathbf{M}}}t\right) - \frac{L_{\mathbf{M}}\|\mathbf{y}(0)\|_2}{L_{\mathbf{M}}}$$

$$= \|\mathbf{x}(0)\|_2\sqrt{L_{\mathbf{M}}}\sinh(\sqrt{L_{\mathbf{M}}}t) + \|\mathbf{y}(0)\|_2(\cosh\left(\sqrt{L_{\mathbf{M}}}t\right) - 1) \qquad \text{(E.1)}$$

Again from Lemma 34, we know that

$$\begin{bmatrix} \mathbf{x}(t) \\ \mathbf{y}(t) \end{bmatrix} = \begin{bmatrix} \mathbf{A}(t) & \mathbf{B}(t) \\ \mathbf{C}(t) & \mathbf{D}(t) \end{bmatrix}\begin{bmatrix} \mathbf{x}(0) \\ \mathbf{y}(0) \end{bmatrix}$$

thus

$$\begin{bmatrix} \mathbf{x}(t) - \mathbf{x}(0) - t\mathbf{y}(0) \\ \mathbf{y}(t) - \mathbf{y}(0) \end{bmatrix} = \begin{bmatrix} \mathbf{A}(t) - I & \mathbf{B}(t) - tI \\ \mathbf{C}(t) & \mathbf{D}(t) - I \end{bmatrix} \begin{bmatrix} \mathbf{x}(0) \\ \mathbf{y}(0) \end{bmatrix}$$

combined with (E.1), and using the fact that the above hold for all $\mathbf{y}(0)$ and $\mathbf{x}(0)$, we can bound

$$\|\mathbf{A}(t) - I\|_2 \le \cosh(\sqrt{L_\mathbf{M}}t) - 1$$

$$\|\mathbf{B}(t) - tI\|_2 \le \frac{1}{\sqrt{L_\mathbf{M}}} \sinh\left(\sqrt{L_\mathbf{M}}t\right) - t$$

$$\|\mathbf{D}(t) - I\|_2 \le \cosh\left(\sqrt{L_\mathbf{M}}t\right) - 1$$

Finally, to prove the third claim,

$$\frac{d}{dt} \begin{bmatrix} \mathbf{x}(t) - t\mathbf{y}(0) - \frac{t^2}{2}\mathbf{M}(0)\mathbf{x}(0) \\ \mathbf{y}(t) - \mathbf{y}(0) - t\mathbf{M}(0)\mathbf{x}(0) \end{bmatrix}$$

$$= \begin{bmatrix} 0 & I \\ \mathbf{M}(t) & 0 \end{bmatrix} \begin{bmatrix} \mathbf{x}(t) \\ \mathbf{y}(t) \end{bmatrix} - \begin{bmatrix} 0 & I \\ \mathbf{M}(0) & 0 \end{bmatrix} \begin{bmatrix} \mathbf{y}(0) \\ \mathbf{v} \end{bmatrix}$$

$\square$

**Lemma 36.** *Let $\mathbf{exp_{mat}}$ be as defined in Lemma 34. Let*

$$\begin{bmatrix} \mathbf{A}(t) & \mathbf{B}(t) \\ \mathbf{C}(t) & \mathbf{D}(t) \end{bmatrix} := \mathbf{exp_{mat}}\left(t; \begin{bmatrix} 0 & I \\ \mathbf{M}(t) & 0 \end{bmatrix}\right)$$

*for some $\mathbf{M}(t)$. Assume $\|\mathbf{M}(t)\|_2 \le L_\mathbf{M}$ for all $t$. Then for all $t$,*

$$\|\mathbf{C}(t) - t\mathbf{M}(0)\|_2 \le \frac{(L_\mathbf{M}' + \frac{1}{2}L_\mathbf{M}^2)}{\sqrt{L_\mathbf{M}}} \sinh(\sqrt{L_\mathbf{M}}t)$$

*Proof.* The proof is similar to Lemma 35. Consider the ODE given by

$$\frac{d}{dt} \begin{bmatrix} \mathbf{x} \\ \mathbf{y} \end{bmatrix}(t) = \begin{bmatrix} 0 & I \\ \mathbf{M}(t) & 0 \end{bmatrix} \begin{bmatrix} \mathbf{x}(t) \\ \mathbf{y}(t) \end{bmatrix}$$

with initial condition $\mathbf{y}(0) = 0$.

By Lemma 34, $\begin{bmatrix} \mathbf{A}(t) & \mathbf{B}(t) \\ \mathbf{C}(t) & \mathbf{D}(t) \end{bmatrix}$ satisfies

$$\begin{bmatrix} \mathbf{x}(t) \\ \mathbf{y}(t) \end{bmatrix} = \begin{bmatrix} \mathbf{A}(t) & \mathbf{B}(t) \\ \mathbf{C}(t) & \mathbf{D}(t) \end{bmatrix} \begin{bmatrix} \mathbf{x}(0) \\ \mathbf{y}(0) \end{bmatrix}$$

$$\frac{d}{dt} \begin{bmatrix} \mathbf{x}(t) - \mathbf{x}(0) - \frac{t^2}{2}\mathbf{M}(0)\mathbf{x}(0) \\ \mathbf{y}(t) - t\mathbf{M}(0)\mathbf{x}(0) \end{bmatrix}$$

$$= \begin{bmatrix} \mathbf{y}(t) - t\mathbf{M}(0)\mathbf{x}(0) \\ \mathbf{M}(t)\mathbf{x}(t) - \mathbf{M}(0)\mathbf{x}(0) \end{bmatrix}$$

$$= \begin{bmatrix} \mathbf{y}(t) - t\mathbf{M}(0)\mathbf{x}(0) \\ \mathbf{M}(t)\mathbf{x}(t) - \mathbf{M}(0)\mathbf{x}(0) \end{bmatrix}$$

$$= \begin{bmatrix} \mathbf{y}(t) - t\mathbf{M}(0)\mathbf{x}(0) \\ \mathbf{M}(t)(\mathbf{x}(t) - \mathbf{x}(0) - \frac{t^2}{2}\mathbf{M}(0)) \end{bmatrix} + \begin{bmatrix} 0 \\ (\mathbf{M}(t) - \mathbf{M}(0))\mathbf{x}(0) \end{bmatrix} + \begin{bmatrix} 0 \\ \frac{t^2}{2}\mathbf{M}(t)\mathbf{M}(0)\mathbf{x}(0) \end{bmatrix}$$

By Cauchy Schwarz, for all $t \le 1$,

$$\frac{d}{dt} \left\| \mathbf{x}(t) - \mathbf{x}(0) - \frac{t^2}{2}\mathbf{M}(0)\mathbf{x}(0) \right\|_2 \le \|\mathbf{y}(t) - t\mathbf{M}(0)\mathbf{x}(0)\|_2$$

$$\frac{d}{dt} \|\mathbf{y}(t) - t\mathbf{M}(0)\mathbf{x}(0)\|_2 \le L_\mathbf{M} \left\| \mathbf{x}(t) - \mathbf{x}(0) - \frac{t^2}{2}\mathbf{M}(0)\mathbf{x}(0) \right\|_2 + (L_\mathbf{M}' + \frac{1}{2}L_\mathbf{M}^2)\|\mathbf{x}(0)\|_2$$

Apply Lemma 31 with $a_t = \left\|\mathbf{x}(t) - \mathbf{x}(0) - \frac{t^2}{2}\mathbf{M}(0)\mathbf{x}(0)\right\|_2$, $b_t = \|\mathbf{y}(t) - t\mathbf{M}(0)\mathbf{x}(0)\|_2$, $C = L_{\mathbf{M}}$, $D = 0$ and $E = (L'_{\mathbf{M}} + \frac{1}{2}L_{\mathbf{M}}^2)\|\mathbf{x}(0)\|_2$ to get

$$a_t \leq \frac{(L'_{\mathbf{M}} + \frac{1}{2}L_{\mathbf{M}}^2)\|\mathbf{x}(0)\|_2}{L_{\mathbf{M}}}(\cosh(\sqrt{L_{\mathbf{M}}}t) - 1)$$

$$b_t \leq \frac{(L'_{\mathbf{M}} + \frac{1}{2}L_{\mathbf{M}}^2)\|\mathbf{x}(0)\|_2}{\sqrt{L_{\mathbf{M}}}}\sinh(\sqrt{L_{\mathbf{M}}}t)$$

Finally, recall that

$$\mathbf{y}(t) - t\mathbf{M}(0)\mathbf{x}(0) = (\mathbf{C}(t) - t\mathbf{M}(0))\mathbf{x}(0)$$

Since we have shown that $\|\mathbf{y}(t) - t\mathbf{M}(0)\mathbf{x}(0)\|_2 \leq \frac{(L'_{\mathbf{M}} + \frac{1}{2}L_{\mathbf{M}}^2)\|\mathbf{x}(0)\|_2}{\sqrt{L_{\mathbf{M}}}}\sinh(\sqrt{L_{\mathbf{M}}}t)$ for all $\mathbf{x}(0)$, it follows that

$$\|\mathbf{C}(t) - t\mathbf{M}(0)\|_2 \leq \frac{(L'_{\mathbf{M}} + \frac{1}{2}L_{\mathbf{M}}^2)}{\sqrt{L_{\mathbf{M}}}}\sinh(\sqrt{L_{\mathbf{M}}}t)$$

$\square$

## F   Miscellaneous Lemmas

**Lemma 37.** *Let $c \in \mathbb{R}^+$ be such that $c \geq 3$. For any $x$ satisfying $x \geq 3c\log c$, we have that*

$$\frac{x}{\log x} \geq c$$

The following Lemma is taken from Sun et al. [2019]:

**Lemma 38.** *For any $x \in M$, $a, y \in T_x M$*

$$\mathrm{d}(\mathrm{Exp}_x(y + a), \mathrm{Exp}_{\mathrm{Exp}_x(a)}(\Gamma_x^{\mathrm{Exp}_x(a)}y)) \leq L_R\|a\|\|y\|(\|a\| + \|y\|)e^{\sqrt{L_R}(\|a\| + \|y\|)}$$

*Proof.* From the proof of Lemma 3 from Sun et al. [2019] (which is in turn a refinement of the proof from Karcher [1977])

$$\mathrm{d}(\mathrm{Exp}_x(y + a), \mathrm{Exp}_{\mathrm{Exp}_x(a)}(\Gamma_x^{\mathrm{Exp}_x(a)}y))$$

$$\leq \int_0^1 \frac{\cosh(\sqrt{L_R}\|y + (1-t)a\|) - \frac{\sinh(\sqrt{L_R}\|y + (1-t)a\|)}{\sqrt{L_R}\|y + (1-t)a\|}}{\|y + (1-t)a\|}dt \cdot \|a\|\|y\|$$

$$\leq \sqrt{L_R}\int_0^1 \sqrt{L_R}\|y + (1-t)a\|e^{\sqrt{L_R}\|y + (1-t)a\|}dt \cdot \|a\|\|y\|$$

$$\leq L_R\|a\|\|y\|(\|a\| + \|y\|)e^{\sqrt{L_R}(\|a\| + \|y\|)}$$

where we use the fact from Lemma 39 that for all $r \geq 0$,

$$\frac{\cosh(r)}{r} - \frac{\sinh(r)}{r^2} \leq re^r$$

$\square$

**Lemma 39.** *For all $r \geq 0$,*

$$\sinh(r) \leq re^r$$

$$\cosh(r) - 1 \leq \frac{r^2}{2}e^r$$

$$\frac{\cosh(r)}{r} - \frac{\sinh(r)}{r^2} \leq re^r$$

$$\cosh(r) \leq e^r$$

$$\frac{\sinh(r)}{r} - 1 \leq r^2 e^3$$

*Proof.* Elementary computation from power series. $\square$