# OpenReview forum: "Efficient Sampling on Riemannian Manifolds via Langevin MCMC"
_NeurIPS.cc/2022/Conference — NeurIPS 2022 Accept_

### Official Review · Reviewer_GcMX · 2022-07-07

**Rating:** 7
**Confidence:** 4
**Soundness:** 3 good
**Presentation:** 3 good
**Contribution:** 2 fair

**Summary:**

The paper under review studies the Langevin diffusion on Riemannian manifold. The SDE (overdamped Langevin dynamic) is defined as the the intrinsic diffusion process on the manifold with the drift term given as the gradient (Levi-Civita sense) of the potential function. The discrete scheme is given as the exponential map at each iteration. The error are studied in the Geometric Euler-Murayama setting. The convergence analysis is conducted in various cases under different assumptions of the curvature tensor and the drift vector fields.

**Questions:**

1. The general setting of the paper is to generalize the over-damped Langevin to Riemannian manifold. Although the foundation and the setup is clear, it is not clear how the non-convex problem could be solved in this case. In the Euclidean setting, there are various methods to tackle the non-convex problem. Is it possible to say anything how the Riemannian manifold could contribute in the non-convex setting?

2. The negative Ricci curvature and convex/ non-convex of the vector field should be discussed together, in the sense that which type of combination could be solved and which combinations are not.

3. In the Euclidean setting, the target distribution is just $e^{-h(x)}$. In the Riemannian setting, one need to consider the extra $d vol_g(x)$ term. One should make it clear, in application why $e^{-h}d vol_g(x)$ is the interesting target distribution, can the author provide any examples ?

4. Could the author explain briefly how is the two-sided curvature bound and the negative lower Ricci curvature bound used explicitly in the results and proof? This could make the contribution straightforward without looking into the long appendix.


**Strengths And Weaknesses:**

The presentation is clean and neat. The requirement of allowing the Ricci curvature tensor to be negative is quite impressive if there is no extra Lyapunov functions are used.
The appendix seems to be well organized, but it is too long. (I have to say that I cannot read all of the appendix).

The results seems to be interesting, however the application is not thoroughly discussed and there are no experiments in the work.  The key idea of the main results (proof) should be made clear, like why the negative Ricci curvature lower bound is Okay. (From the reviewer's experience, Lyapunov function shall play a role in such situations, if this is not the case in the current work, this definitely deserve some space.)

---

> ### Author Response · Authors · 2022-08-02
> **Response to Reviewer GcMX  (1/2)**
>
> We thank the reviewer for their review and helpful comments. We will first address question 3 since it is also related to the reviewer's comment about applications of this work.
>
> > In the Euclidean setting, the target distribution is just $e^{-h(x)}$. In the Riemannian setting, one need to consider the extra $dvol_g(x)$ term . One should make it clear, in application why $e^{-h} dvol_g(x)$ is the interesting target distribution, can the author provide any examples?
>
> > ... the application is not thoroughly discussed and there are no experiments in the work.
>
> One practical example where $dvol_g$ significantly improves sampling is using diffusion wrt Fisher-Rao metric for sampling from a posterior distribution. (see Girolami and Calderhead: https://rss.onlinelibrary.wiley.com/doi/10.1111/j.1467-9868.2010.00765.x). In certain experiments, manifold Langevin can be up to a **few hundred times faster (in runtime)** than Euclidean Langevin.
>
> On the theory side, authors such as Ahn and Chewi, Gatmiry and Vempala have shown that mirror Langevin (which is related to diffusion wrt the Hessian metric of log-density) for log-concave densities have very good convergence rates. Results such as Brascascamp Lieb imply mixing rates **independent of the condition number of the log-density** for Hessian manifold diffusion.
>
> A third interesting setting involves recent attempts to analyze the generalization properties of SGD on neural network training via its approximation by a Langevin diffusion wrt the gradient-covariance manifold. See e.g. Pesme, Pillaud-Vivien, Flammarion, and references therein. We have also listed other cases of interesting $dvol_g$ in section 2 (where the above mentioned papers are also discussed).
>
> > The general setting of the paper is to generalize the over-damped Langevin to Riemannian manifold. Although the foundation and the setup is clear, it is not clear how the non-convex problem could be solved in this case. In the Euclidean setting, there are various methods to tackle the non-convex problem. Is it possible to say anything how the Riemannian manifold could contribute in the non-convex setting?
>
> In addition to our response to the previous question, we list below some specific advantages of manifold Langevin for nonconvex potentials:
> - In our analysis, mixing rate depends on **parameterization independent** quantities, which could be particularly important since for non-convex distributions, mixing rates can be exponential in quantities such as the dissipativity parameter. In the aforementioned examples, these quantities are more benign on manifolds.
> - Certain problems are naturally occuring on manifolds (e.g. statistical manifold), and Euclidean approaches would need to **additionally handle projection onto a manifold, which is often a non-convex body**.
> - Results such as Ahn and Chewi/Gatmiry Vempala work for non-convex potentials via techniques such as the Holley-Stroock theorem, possibly guaranteeing better mixing rates for the manifold diffusion compared to Euclidean diffusion.
>
> (I am not certain that I understood your question, please let me know if you were asking about something else)
>
> > The negative Ricci curvature and convex/ non-convex of the vector field should be discussed together, in the sense that which type of combination could be solved and which combinations are not.
>
> When Ricci curvature + convexity (contractivity) of drift is positive (i.e. Assumption 2 with $R=0$), the iteration complexity is polynomial in app problem parameters. When Ricci curvature + drift contractivity is negative (either due to large negative Ricci curvature, or due to strongly diverging drift), the Langevin MCMC is still guaranteed to well-approximate the Langevin Diffusion, but the Langevin Diffusion can mix very slowly leading to a large number of steps. See line 261-284 for a quantitative discussion.
>
> As reviewer QBd7 pointed out, our condition on Ricci curvature is similar to the Bakry Emery condition, which also allows for negative Ricci curvature. If you are familiar with that result, it may help provide intuition for what cases are easy and what cases are hard (in terms of iteration complexity).

---

> > ### Author Response · Authors · 2022-08-02
> > **Response to Reviewer GcMX (2/2)**
> >
> > > ... why the negative Ricci curvature lower bound is Okay. (From the reviewer's experience, Lyapunov function shall play a role in such situations, if this is not the case in the current work, this definitely deserve some space.)
> >
> > When Ricci curvature + drift contractivity is negative, we indeed use a Lyapunov function (defined in Appendix B.2) in our convergence analysis. However, when the sum is positive, we can show contraction in the distance directly, even if, say, Ricci curvature is negative. We will add a more detailed discussion of this to the paper. Thank you for the suggestion.
> >
> > > Could the author explain briefly how is the two-sided curvature bound and the negative lower Ricci curvature bound used explicitly in the results and proof? This could make the contribution straightforward without looking into the long appendix.
> >
> > The two-sided sectional curvature bound is used to control **discretization error**, most notably, the error in approximating a Brownian motion by a single Gaussian step (see proof sketch of Lemma 1). It is mainly used in Lemmas 28 and 38.
> >
> > The negative Ricci curvature bound is mainly for **continuous time convergence**. It ensures that two Brownian motions do not diverge too quickly -- if the Ricci curvature is very negative, two Brownian motions, regardless of coupling, will in expectation drift far away from each other very quickly, and if the drift does not contract enough to counteract this, there will be no stationary distribution. It is mainly used in Lemma 3.

---

> > > ### Comment · Reviewer_GcMX · 2022-08-10
> > > **Response to author**
> > >
> > > Thanks for your detailed response. I have raised my score to 7.

---

### Official Review · Reviewer_cfEV · 2022-07-10

**Rating:** 6
**Confidence:** 3
**Soundness:** 3 good
**Presentation:** 3 good
**Contribution:** 3 good

**Summary:**

The authors show theoretically that under some assumptions, sampling from a distribution on a Riemannian manifold using the associated discretized SDE provides a good approximate solution. In particular, it is shown that the error of the discretization can be bounded and that the final density is close to the true density under the Wasserstein metric.

**Questions:**

- The SDE that is analysed is used to generate samples from a stationary distribution. In practice, this means generating a "long enough" chain such that the collected samples to represent the associated density. I think when such approaches are used, due to the discretization there is an implicit bias which is tackled using an acceptance ratio scheme. Is this necessary in your sampling scheme? How the geometry influences the behaviour due to the discretization?
- In your analysis the expmap operations seem to take place in the ambient space, so I assume that the parameters $\beta$ and $\zeta$ are tangent vectors in the ambient space (Eq. 1.2)?
- The results related to the Wasserstein distances imply that empirical distribution of the collected samples with the discretized SDE, is close to the true distribution? And the closeness is measured with the Wasserstein distance using the geodesic distance as the distance measure in the associated integral?
- In Lemma 1 the $\zeta$ that is used for the Expmap is used as the initial condition for the true solution $x(t)$ of the SDE?
- For the Theorem 1 to hold, it seems that the starting point $x_0$ must be close enough to a mode $x^*$. If this is the case, then I suppose that the generated samples from the discretized SDE are a good approximation of the density but only locally near the mode? In contrast, the Lemma 1 holds even if the $x_0$ is not near a mode $x^*$, is this correct?
- Are there any simple synthetic examples that satisfy the assumptions?

**Limitations:**

Some limitations of the analysis are discussed.

**Strengths And Weaknesses:**

- Originality: I am not an expert in the topic of sampling, but it seems to me that the current work is novel and differs from the related works, which seem to be properly discussed.

- Quality: The technical part of the paper seems to be correct and sound. Of course, I could not read carefully all the theory, but the analysis seems reasonable.

- Clarity: I think that the clarity of the paper can be improved such that to help non-expert readers. I acknowledge that the authors tried to simplify parts of their analysis to make it accessible, but some parts perhaps is possible to be improved. For example:
	1. Some toy examples of the method can be provided to allow the reader become familiar with the topic. For example, a distribution on an appropriate Riemannian manifold together with the associated algorithm/results.
	2. Some additional images could help the reader understand parts of the theory and the proofs. Figure 1 helps a lot in understanding, and I think more figures in the same spirit would have been very beneficial.
	3. In a related manner, I am not sure if it is possible to simplify some technical parts even further. Maybe some parts of the sketch-proofs can be moved in the appendix and instead provide the gist with graphical examples?

- Significance: The theoretical results seem to be interesting and useful, and as it is claimed by the authors their analysis improves upon previous approaches. I think that the provided guarantees can be potentially taken into account when practical models are developed. However, the downside is that it is not easy to verify if and when the assumptions holds, while the authors do not seem to provide some examples. Also, even if the main target group of the paper seems to be readers very familiar with the associated topics, I believe non-expert readers could be benefited if parts of the paper become more accessible for them.

---

> ### Author Response · Authors · 2022-08-02
> **Response to Reviewer cfEV (1/2)**
>
> We thank the reviewer for their review and feedback. We address the questions below.
>
> > ...the downside is that it is not easy to verify if and when the assumptions holds, while the authors do not seem to provide some examples...Are there any simple synthetic examples that satisfy the assumptions?
>
> Below, we provide two examples where our assumptions are easy to verify:
>
> One rather well known motivating example is the setting of posterior sampling of parameters of a normal distribution, using Langevin diffusion wrt Fisher-Rao metric. This is a constant-negative-curvature space, and details+empirical simulations of manifold Langevin can be found in Section 5.1, Girolami and Calderhead: https://rss.onlinelibrary.wiley.com/doi/10.1111/j.1467-9868.2010.00765.x. More generally, the Fisher Rao metric is suitable for many other exponential family distributions as well.
>
> Another simple example is any L-gradient-Lipschitz function defined on a unit sphere. Since the manifold is compact without boundary, the first part of Assumption 2 is satisfied automatically by taking $R=\pi$. The sectional of the sphere is 1, and smoothness properties follow from smoothness of the potential. (The sphere example is also nice for illustrating why Brownian motion mixes very fast under positive Ricci curvature.)
>
> _Specifically regarding Assumption 2_: we refer the reviewer to our response to question #3 of reviewer 9bhF above, where we state the Euclidean version of Assumption 2. This is a somewhat common assumption when studying non-convex sampling (see e.g. Gorham, Vollham, Mackey), and we provide some further quantitative discussion on Lines 261-284.
>
> We will add a motivating example to illustrate the assumptions, thank you for your suggestion.
>
> > The SDE that is analysed is used to generate samples from a stationary distribution. In practice, this means generating a "long enough" chain such that the collected samples to represent the associated density. I think when such approaches are used, due to the discretization there is an implicit bias which is tackled using an acceptance ratio scheme. Is this necessary in your sampling scheme? How the geometry influences the behaviour due to the discretization?
>
> There is indeed an implicit bias in our case, though we do not need to tackle it explicitly using acceptance ratio. This bias shows up in the form of discretization error in Theorem 1. Specifically, the $\tilde{O}(\delta^{1/2})$ term below line 247. By taking the stepsize $\delta$ sufficiently small, this bias goes to 0. Using an acceptance ratio may indeed allow us to take larger stepsize $\delta$ and lead to faster algorithms, but that would be for future work.
>
> The bias depends on the geometry of the manifold via polynomial dependence on sectional curvature, as well as smoothness/contractivity of the drift which are defined wrt the manifold distance.
>
> > In your analysis the expmap operations seem to take place in the ambient space, so  I assume that the parameters $\beta$ and $\zeta$ are tangent vectors in the ambient space (Eq. 1.2)?
>
> The expmap operator $Exp_x(v)$ is the point in $M$ obtained from starting at $x \in M$ and following the geodesic in the $v$ direction for a unit time. $\beta$ and $\zeta$ are explicitly vectors in the tangent space at some point $x$ in $M$. However, any expmap is equivalent to following some curve in embedding coordinates, so if you prefer you can view $\beta$ and $\zeta$ as representing geodesic curves in the embedding coordinates, which are uniquely parameterized by the initial velocity.

---

> > ### Author Response · Authors · 2022-08-02
> > **Response to Reviewer cfEV (2/2)**
> >
> >
> > >The results related to the Wasserstein distances imply that empirical distribution of the collected samples with the discretized SDE, is close to the true distribution? And the closeness is measured with the Wasserstein distance using the geodesic distance as the distance measure in the associated integral?
> >
> > Yes, both statements are correct.
> >
> > > In Lemma 1 the $\zeta$ that is used for the Expmap is used as the initial condition for the true solution $x(t)$ of the SDE?
> >
> > $\zeta$ is not related to the initial condition of $x(t)$.
> >
> > The initial condition for the true solution $x(t)$ is $x(0)$, which is arbitrary. Lemma 1 considers the discretization $x^0(t)$, which is the Expmap in some $\beta + \zeta$ direction, starting at $x(0)$. The statement of Lemma 1 does not say how $\zeta$ is related to $x(t)$. However, in the proof of Lemma 1 (and in the proof sketch), one can see that $\sqrt{t}\zeta$ is in fact the endpoint of a Brownian motion $B(t)$, which is related in a complex way to the manifold Brownian motion $B^g_t$ that drives $x(t)$ in (1.1).
> >
> >
> > > For the Theorem 1 to hold, it seems that the starting point $x_0$ must be close enough to a mode $x^*$. If this is the case, then I suppose that the generated samples from the discretized SDE are a good approximation of the density but only locally near the mode? In contrast, the Lemma 1 holds even if the $x_0$ is not near a mode $x^*$?
> >
> > The initial condition on $x_0$ in Theorem 1 is _only for convenience_ (under Assumption 2, we will contract to $x^*$ exponentially fast from any starting position). In fact, it is entirely possible that at some step $k$, $dist(x_k,x^*) > 2R > dist(x_0,x^*)$, which is fine. Our guarantee is only that the total error in distribution is $\tilde{O}(\delta^{1/2})$. We do not show whether this error is "mostly due to far away points", but that is a reasonable guess. You are correct that Lemma 1 does not assume anything about starting distance to $x^*$.
> >
> > We sincerely hope that the response to your concerns, as well as the overall response to other reviewers’ concerns, helps assuage your concerns, and view this paper in a more favorable light.

---

> > > ### Comment · Reviewer_cfEV · 2022-08-08
> > > **After rebuttal**
> > >
> > > First of all thank you for your replies.
> > >
> > > I still have a follow-up question regarding Lemma 1, maybe the answer is straightforward but I am a bit puzzled.
> > > - At the point $x(0)$ the $\beta(x(0))$ specifies the tangent vector (initial velocity). The random sample $\zeta$ introduces some stochasticity and the $x^0(t) = \text{Expmap}(t \beta(x(0)) + \sqrt{t}\zeta)$ induces a trajectory.
> > > - On the other hand, a "true trajectory" $x(t)$ is initialized again at $x(0)$, has the same initial velocity but from Eq. 1.1 we know that this trajectory is a bit more complicated due to the Brownian motion term $B_t^g$. As you already replied the $\zeta$ is not (directly) related to the "true trajectory" $x(t)$, and also, we can not compute the actual trajectory $x(t)$.
> > > - Does the Lemma 1 shows that for a small time interval, a long sequence of Expmaps with stochasticity (the end of the previous expmap specifies the initial velocity of the next expmap via $\beta(\cdot)$ plus some noise $\zeta$), converges in expectation to a "true trajectory" $x(t)$? And in addition, replacing this long sequence of Expmaps with only one is still a reasonable choice?
> > > - So the expectation in Lemma 1 is taken with respect to the random trajectories taken by (theoretically) solving Eq. 1.1 and the random Expmaps implied by $\zeta$?
> > >
> > > In general, I like the paper and the effort to make it accessible even if the content is purely technical. However, I still believe that some images and/or some extra discussion can help the (non-expert) reader to understand better the gist of your theoretical results. I will increase my score influenced by the rest of the reviews, and I hope that the accessibility will be improved in the final version.

---

> > > > ### Author Response · Authors · 2022-08-08
> > > > **Follow-up Response to Reviewer cfEV**
> > > >
> > > > Thank you for your reconsideration, as well as for clarifying your question and for your suggestions on presentation. We will keep that in mind when preparing the final draft.
> > > >
> > > > You are correct in your description of the relation between $x^0(t)$ and $x(t)$. The expectation in Lemma 1 is taken with respect to the random trajectories, where randomness is entirely due to 1. $\zeta$ and 2. $B^g_t$ (which are not independent).
> > > >
> > > > We will elaborate a little more on your question
> > > > > Does the Lemma 1 shows that for a small time interval, a long sequence of Expmaps with stochasticity (the end of the previous expmap specifies the initial velocity of the next expmap via $\beta(\cdot)$ plus some noise $\zeta$), converges in expectation to a "true trajectory" $x(t)$? And in addition, replacing this long sequence of Expmaps with only one is still a reasonable choice?
> > > >
> > > > For each integer $i$, $x^i(t)$ represents a "discrete sequence of steps, with stepsize $\delta^i = 2^{-i} T$" (see line 173). You are correct that as $i\to \infty$ (i.e. stepsize $\to$ 0), $x^i(t)$ converges to $x(t)$ (the true trajectory). This convergence is in expected-manifold-distance, with expectation taken wrt the randomness in $\zeta$ and $B^g_t$.
> > > >
> > > > In application to Theorem 1, $x^0(T)$ from Lemma 1 represents **a single Euler Murayama step**. I.e. $(T \text{ in Lemma 1}) = (\delta \text{ in Theorem 1})$. The guarantee in Theorem 1, for Langevin MCMC, uses Lemma 1 many times, one time for each single Euler Murayama step. In short, **in Lemma 1, $T$ is supposed to be a rather small value, corresponding to a single Euler Murayama step**, unlike in Theorem 1, where $T$ can be arbitrarily large.
> > > >
> > > > It is for this reason that "replacing this long sequence of Expmaps with only one is still a reasonable choice" -- because even the single step, $x^0(t)$, is only over a short time $T$. On the other hand, the $x^i(t)$, which consist of large numbers of of Exp maps, are only for analysis purposes.

---

### Official Review · Reviewer_9bhF · 2022-07-13

**Rating:** 8
**Confidence:** 3
**Soundness:** 4 excellent
**Presentation:** 4 excellent
**Contribution:** 3 good

**Summary:**

The paper is concerned with convergence analysis of the Langevin MCMC algorithm on a Riemannian manifold. This is extension of the results on the Euclidean space to the manifold setting. It provides error rates between the continuous-time stochastic process that solves the Langevin eq. on the manifold and its Euler-Murayama discretized version (which represents the algorithm). The discretization error combined with convergence rate of the continuous process  provides error bounds for the output of the algorithm: number of iterations scales with $\tille O(\epsilon^{-2})$ for $\epsilon$ error. The convergence analysis is also extended to the case where only unbiased random estimate of the gradient are available. The error bounds hold on the dissipativity assumption on the drift outside a ball (which includes gradient of non-convex functions) and lower bounds on the Ricci curvature of the manifold. The second result for stochastic gradients require dissipativity everywhere.

**Questions:**

Minor:

1- Line 28: by $g(x)$ do you mean the Riemannian metric?

2- Line 97: item ii): do you also mean almost surely?

3- What is the Euclidean counterpart of assumption 2? I think including it would make the assumption easier to understand.

4- Sketch proof of Lemma 1: Equation after 172: I think it would be better to clarify the relation between $\zeta^i_k$ for different $i$ by including their definitions there, rather than after 4.1. Then it is easier for the reader to understand Figure 1 and the following two steps in connection with the definition.

5- I did not understand the remark following line 276: "under assumption 3, $\beta$ satisfies assumption 2"? Do you mean $q\leq L'_\beta$

6- I think $\mathcal R$ should not appear in the result of Theorem 2.




**Limitations:**

Yes

**Strengths And Weaknesses:**

Strength:
- Well-written paper given its technical nature
- Well discussion of related literature and connection to previous results
- Important and useful theoretical results and their discussion
- Explicit convergence rate in terms of  constants
- Generality of the assumptions


Weakness:
- minor clarifications on details

---

> ### Author Response · Authors · 2022-08-02
> **Response to Reviewer 9bhF**
>
> We thank the reviewer for their review as well as for their careful reading of the paper and valuable suggestions. We address the questions below.
>
> > 1- Line 28: by $g(x)$ do you mean the Riemannian metric?
>
> Yes, $g$ is the Riemannian metric.
>
> > 2- Line 97: item ii): do you also mean almost surely?
>
> Yes, we mean almost surely, we will update the draft to be more precise.
>
> > 3- What is the Euclidean counterpart of assumption 2? I think including it would make the assumption easier to understand.
>
> Thank you for your suggestion. We agree that it would help clarify the assumption (and in addition relate to earlier Euclidean sampling papers which make similar assumptions). The Euclidean version of Assumption 2 is as follows:
>
> Replace $\Gamma(\beta(y), y\to x)$ by $\beta(y)$, and replace $\gamma'(0)$ by $y-x$, so Assumption 2 would read:
>
> A. for $||x-y||^2 >R$, $\langle\beta(y)-\beta(x), y-x\rangle < -m ||x-y||^2$, e.g. this holds if $\beta$ is the negative gradient of a m-strongly-convex function
>
> B. for $||x-y||^2 <R$, $\langle \beta(y)-\beta(x), y-x\rangle < q ||x-y||^2$, e.g. this holds if $\beta$ is the negative gradient of a $q$-gradient-lipschitz function.
>
> You may recognize this as the distant-dissipativity assumption that is often considered in non-convex Euclidean sampling.
>
> > 4- Sketch proof of Lemma 1: Equation after 172: I think it would be better to clarify the relation between $\zeta^i_k$ for different $i$ by including their definitions there, rather than after 4.1. Then it is easier for the reader to understand Figure 1 and the following two steps in connection with the definition.
>
> Thank you for the suggestion. This does seem to make the sketch easier to follow, we will move the part up or add a better description of $\zeta^i$ around line 172.
>
> > 5- I did not understand the remark following line 276: "under assumption 3, satisfies assumption 2"? Do you mean $q<L_\beta'$
>
> Yes, we mean $\beta$ satisfies the second statement of Assumption 2 with $q < L_{\beta}'$
>
> > 6- I think $R$ should not appear in the result of Theorem 2.
>
> Thank you for catching that, $R$ is indeed 0 so it should not appear.

---

### Official Review · Reviewer_QBd7 · 2022-07-13

**Rating:** 8
**Confidence:** 4
**Soundness:** 3 good
**Presentation:** 4 excellent
**Contribution:** 4 excellent

**Summary:**

The authors analyzed an Euler type discretization of diffusions, in particular the Langevin diffusion, on a large class of Riemannian manifolds. The authors provided a one-step discretization error bound, which leads to a sampling guarantee when discretizing the Langevin diffusion. The authors also extended these results to the stochastic gradient setting.

Given the challenging nature of studying discretizations of diffusions on general manifolds, I believe any results should be welcomed, even if it is not ultimately the best error and sampling bounds possible. I would encourage the area chair and the other reviewers to not penalize the paper for not achieving the best possible sampling bound, as it is still a significant step towards understanding sampling on manifolds.

**Questions:**

I have several technical questions that I would like to clarify with the authors. I believe these are due to the technical nature of the paper, and therefore should not affect my score, but I would still prefer to understand them in detail since we have a discussion period.

# On Lemma 1 - Discretization Error Bound

On a separate note, I just want to say the diagram is actually really helpful to understanding the steps of the proof, as the notation is quite heavy.

1. On line 204, summing the bounds give an exponential factor of $\exp( O( L_R d \delta^i ) )$ and there a recursion over $k$ as well, which would compound this to the exponent $2^i$. My question is, does $\delta$ then need to be chosen in a way to remove the exponential factor here?

2. On a higher level, I'm surprised that the authors did not have to work with stochastic parallel transport along a Brownian path, which I personally found quite difficult. Can the authors comment on which part of the construction that allowed the authors to avoid this technical difficulty? I'm speculating here, but is it due to the construction of $x^i(t)$ such that the realized paths are always geodesics and not Brownian?

# On Theorem 1 - Langevin Sampling Bound

1. Can I interpret the assumption of $m - L_{Ric}/2 > 0$ as morally a Bakry--Emery type criterion (especially if $\mathcal{R}=0$)?

2. In a similar spirit, can I also interpret the factor $exp( -(q+L_{Ric}/2) \mathcal{R}^2 )$ in the convergence rate $\alpha$ as a Holley--Stroock type perturbation?

**Strengths And Weaknesses:**

# Strengths
1. The authors provided the first (to the best of my knowledge) discretization error bounds for diffusions on a large class of manifolds in Lemma 1. In particular, the discretization is computable (i.e. not a Brownian motion increment on the manifold), and the assumptions on the manifold is significantly weaker than existing work.

2. The authors provided one of the first sampling complexity bounds on a large class of manifolds in Theorem 1, with Gatmiry and Vempala (2022) - the only other work I know of - taking a different approach and therefore should not be directly compared.

# Weaknesses
1. It is likely that the sampling bound in Theorem 1 can still be improved in terms of other factors such as $d, m, \mathcal{R}, L_{\text{Ric}}$ etc.

---

> ### Author Response · Authors · 2022-08-02
> **Response to Reviewer QBd7**
>
> We thank the reviewer for their review and insightful questions, and for identifying the strengths of this paper. We address the questions below.
>
> > On line 204, summing the bounds give an exponential factor of $e^{L_R d \delta^i}$ and there a recursion over $k$ as well, which would compound this to the exponent $2^i$. My question is, does then need to be chosen in a way to remove the exponential factor here?
>
> Yes. Note that $\delta^i = T \cdot 2^{-i} = \delta^0 \cdot 2^{-i}$ (see line 173). For any $i$, recursing over $=0...2^i$ gives a total exponent of $e^{L_R d 2^i \delta^i} = e^{L_R d \delta^0}$. We pick $\delta^0$ to be smaller than $O(1/(L_R d))$, which ensures that $e^{L_R d \delta^0}$ is a constant, thus controlling the $e^{L_R d 2^i \delta^i}$ error for all $i$. Note also that $\delta^0$ is also the actual stepsize in the Langevin MCMC algorithm (and $\delta^i$ are only for analytical purposes).
>
> > On a higher level, I'm surprised that the authors did not have to work with stochastic parallel transport along a Brownian path, which I personally found quite difficult. Can the authors comment on which part of the construction that allowed the authors to avoid this technical difficulty? I'm speculating here, but is it due to the construction of $x^i(t)$ such that the realized paths are always geodesics and not Brownian?
>
> You are exactly correct -- the construction of $x^i(t)$ being sequences of discrete geodesic steps in Gaussian directions (instead of Brownian motion) is what lets us sidestep the stochastic manifold analysis. The bounds we prove uniformly for $x^i(t)$ extend to its limit as $i\to \infty$, which is a Brownian motion. There is also one place where we crucially rely on parallel transport along the discretized-Brownian-approximation-sequences: see the third line of expression (4.1), where $E^{i+1}\_{2k+1} = \Gamma (E^i\_{2k}; x^{i+1}\_{2k} \to x^{i+1}\_{2k+1}))$, thus we are essentially performing parallel transport along each discrete geodesic step. (Pictorially, this corresponds to the part in Figure 1 where we break down the black arrow into the blue and red arrows.)
>
>
> > Can I interpret the assumption of $m-L_{Ric}/2$ as morally a Bakry--Emery type criterion (especially if $R=0$)?
>
> Yes, we believe that this is very similar to Bakry Emery. In our case, $L_{Ric}$ comes up when analyzing the distance evolution of synchronously-coupled Brownian motions, which is related to the eigenvalue of the Laplace Beltrami operator, which is in turn related to why $L_{Ric}$ appears in Bakry Emery.
>
> > In a similar spirit, can I also interpret the factor $e^{-(q+L_{Ric}/2)R^2}$ in the convergence rate as a Holley--Stroock type perturbation?
>
> Thank you for pointing this out, this is an excellent observation. This is indeed similar to a Holley-Strook type perturbation, and we will add a note about this to the draft. For simplicity consider a compact manifold without boundary with diameter $R$, with Ricci curvature $> -L_{Ric}$. Let $U$ be a $q$-gradient-Lipschitz potential. We can decompose $U$ into a $-L_{Ric}/2$-convex function (denote by $U_1$) and a $q-L_{Ric}/2$ non-convex perturbation with magnitude $(q+L_{Ric}/2)R^2$ (denote by $U_2$). Bakry-Emery implies that $e^{-U_1}$ has a nice log-Sobolev constant, and Holly-Strook's implies that the log-Sobolev constant for $e^{-U} = e^{-U_1 - U_2}$ contains an additional factor of $e^{-(q+L_{Ric}/2)R^2}$.

---

> > ### Comment · Reviewer_QBd7 · 2022-08-08
> > **Response to the Authors**
> >
> > Thank you for the detailed response. I believe all my questions are adequately addressed, and I will raise my score to 8.

---

> > > ### Author Response · Authors · 2022-08-08
> > > **Follow-up Response to Reviewer QBd7**
> > >
> > > Thank you!

---

### Meta-Review · Area_Chair_g1mr · 2022-08-27

**Recommendation:** Accept
**Confidence:** Certain

**Metareview:**

This paper focus on the problem of sampling from a distribution on a large class of Riemannian manifolds. Authors study an Euler-type discretization in this context, for the Langevin diffusion. The paper establishes a control over the one-step discretization error, which is then iterated to obtain a sampling guarantee for the discrete algorithm. Several extensions are also discussed, e.g. the stochastic gradient setting.

Sampling on Riemannian manifolds is a challenging and active area of research. Given that the paper can cover a wide range of manifolds, I also agree with the reviewers, that the contributions of this paper are solid. I strongly recommend accepting this paper.

**Award:**

No

---

### Decision · Program_Chairs · 2022-09-14

Accept